

# Adult life strategy affects distribution patterns in abyssal isopods – implications for conservation in Pacific nodule areas

Saskia Brix[1], Karen J. Osborn[2], Stefanie Kaiser[1,3], Sarit B. Truskey[2], Sarah M. Schnurr[1,4], Nils Brenke[1], Marina Malyutina[5] & Pedro M. Martinez[1,4]

[1] *Senckenberg am Meer, German Centre for Marine Biodiversity Research (DZMB) c/o Biocenter Grindel, Center of Natural History, Universität Hamburg, Martin-Luther-King-Platz 3, 20146 Hamburg, Germany*
[2] *Smithsonian National Museum of Natural History, 10th and Constitution Ave NW, Washington, DC, 20013 USA*
[3] *CeNak, Center of Natural History, Universität Hamburg, Martin-Luther-King-Platz 3, 20146 Hamburg, Germany*
[4] *University of Oldenburg, FK V, IBU, AG Marine Biodiversitätsforschung, Ammerländer Heerstraße 114-118, 26129 Oldenburg*
[5] *A.V. Zhirmunsky National Scientific Center of Marine Biology, Far Eastern Branch, Russian Academy of Sciences, Palchevsky St, 17, Vladivostok 690041, Russia*

*Correspondence to*: Saskia Brix (sbrix@senckenberg.de)

**Abstract**

Aim of our study is to gain a better knowledge about the isopod crustacean fauna of the abyssal Clarion Clipperton Fracture Zone (CCZ) located in the central Pacific Ocean. In total, we examined 22 EBS samples taken at 6 abyssal areas in the central pacific manganese nodule area (CCZ and DISCOL). The dataset comprised 619 specimens belonging to 187 species of four different isopod families: 91 species (48.6 % of total) belonging to Munnopsidae, 63 (33.6 %) to Desmosomatidae, 24 (12.8%) to Haploniscidae and 9 (4.8 %) to Macrostylidae. The total number of species found was relatively similar between sites ranging from 38 (German Contractor area) to 50 species (French contractor area). 68 species were represented by singeletons. The ranges of distribution differ between families. In total 77% of the species were recorded in a single area (and thus being unique for this specific area), 13.9% in 2 areas, 5.3% in 3 areas, 2.6% in 4 areas and 1% in 5 areas. The proportion of species present in a single area increased in this sequence: Munnopsidae (75.8%), Desmosomatidae (77.7%) and Haploniscidae (83%).

A total of 6 (66.6%) out of 9 species of Macrostylidae was recorded in a single area contrasted by the most common species being from this family, Macrostylidae_*Macrostylis*_M05 with 46 specimens (present in all areas besides DISCOL) followed by several species of Munnopsidae with 10 or more specimens in the dataset. The CCZ areas show the highest number of shared species. Generally, the high diversity in each area is reflected by a low similarity between sampling areas. The rarefraction curves indicate that species richness is similar between areas, but the real number of species is still not sampled.

The most distant areas from the central CCZ, the APEI3 and DISCOL, are the most different.



# 1 Introduction

Spanning 60% of the Earth's surface, deep-sea areas (below 200 m water depth) harbour an immense diversity of habitats and species, but also large deposits of metal-rich seafloor minerals (e.g., polymetallic sulphides, cobalt-rich ferromangansese crusts, phosphorite- and polymetallic (Mn-) nodules). Despite initial endeavours to explore these resources starting in the 1960s, growing economic interests coupled with advancing technologies to extract minerals from the seafloor have now made deep-sea mining becoming quite realistic (Wedding et al. 2015; Jones et al. 2017).

The abyssal Clarion Clipperton Fracture Zone (CCZ, Fig. 1) located in the tropical north-eastern Pacific is commercially the most important area of proposed Mn-nodule mining. Extraction of these mineral resources will inevitably lead to habitat loss and changes at the directly mined sites primarily through removal, blanketing and compaction of the upper sediment layer (5-20 cm) (Miljutin et al. 2011; Ramirez-Llodra et al. 2011; Jones et al. 2017; Gollner et al. 2017). Furthermore, areas beyond the actual mining block may be indirectly affected through the generation of a potentially toxic sediment cloud, as well as discharge water from dewatering processes at the sea surface (Oebius et al. 2001; Hauton et al. 2017). Thus, the scale and magnitude of its ecological footprint and how it is mitigated will determine whether mining operations will be feasible in the long-term (Petersen et al. 2016).

In order to make predictions on the recolonization potential of the deep-sea fauna, an understanding of the modes and drivers of species' geographic distributions is required. That is, species with a broader distribution and better dispersal ability likely have a greater potential to recolonize impacted areas than those with a limited dispersal capacity. In turn, this understanding would contribute to the design and establishment of ecological reserve areas in the CCZ (Baco et al. 2016; Vanreusel et al. 2016; De Smet et al. 2017).

Presumed low levels of environmental variability and absence of obvious dispersal barriers, led to the assumption that deep-sea species have wider horizontal distributions compared to shallow-water representatives (McClain & Hardy 2010). However, molecular studies have shown that morphologically similar, but genetically distinct (cryptic) species are common among deep-sea lineages, fundamentally changing our understandings of deep-sea species distributions (e.g., Vrijenhoek et al. 1994; Pfenninger & Schwenk 2007; Raupach et al. 2007; Havermans et al. 2013; Brix et al. 2014, 2015; Jennings et al. 2018, in press). Conversely, for some species there is morphological and genetic support for wide geographic distributions even across major topographic barriers (Brix et al. 2011; Menzel et al. 2011; Riehl & Kaiser 2012; Janssen et al. 2015; Easton & Thistle 2016; Bober et al. 2018; Brix et al.2018). However, biological data on dispersal distances of deep-sea species is still fragmentary due to overall low sampling effort compared to the sheer size of deep-sea floor, and the scant knowledge on species' taxonomy.

Marine benthic invertebrates exhibit a range of reproductive strategies, all these strategies are directly linked to their dispersal potential, with decreasing potential dispersal distance from those with pelagic development to those with direct development (brooding). Yet, some species with true wide geographic ranges have with direct development, and some putatively good dispersers have more limited distributions (Johannesson 1988; Shank 2010; Packmor et al. 2015; Janssen et al. 2015). While



there are a number of ecological and evolutionary factors used to explain differences in range size, including evolutionary history, physiological tolerance and food availability, early life history, nonetheless, range size seems to represent a relatively good proxy of dispersal capacity (Grantham et al 2003; Sherman et al. 2008; Hilario et al. 2015; Janssen et al. 2015; Baco et al. 2016; but see Johannesson 1988; Lester et al. 2007). For benthic taxa with a pelagic larval phase, the time larvae spend in the water column (planktonic larval duration, PLD), is often used to predict dispersal distances (Hilario et al. 2015). For direct

developers that lack planktonic stages, dispersal is limited to active migration and passive drift or floating (rafting) of the adult stage, making estimation of dispersal distances arguably more complicated (Thiel & Haye 2006).

In this study, we assess the role of the adults' lifestyle in determining the large-scale distribution of asellote isopods across the CCZ. Asellote isopods of superfamily Janiroidea are the most numerous and diverse crustacean taxon encountered within abyssal benthic samples (Brandt et al., 2007). With a few exceptions, isopods lack planktonic larvae, and thus levels of gene

flow result from the active and/or passive migration of adults (Brandt, 1992). Asellotes are principally detritivores and foraminiferivores, but different groups show different lifestyles. The Munnopsidae Lilljeborg 1864 are the most diverse and abundant janiroids in the deep sea and their diversity is reflected in numerous morphological and ecological adaptations, most important of which is their paddle-like posterior legs that are highly specialized for swimming or digging. In the Desmosomatidae Sars 1897, usually referred to as an epifaunal family, swimming adaptations are only poorly expressed

compared to the Munnopsidae Lilljeborg 1864, (Hessler 1981; Hessler and Stromberg, 1989). Yet, still desmosomatids bear long natatory setae on their posterior pereopods and are thus considered to be moderate swimmers (Hessler 1981; Svarvasson 1984; Hessler and Strömberg, 1989; Brix et al., 2015, Bober et al. 2018). The Haploniscidae Hansen, 1916 have no modifications for swimming or burrowing. While in situ observations are lacking, information from epibenthic sledge and core sampling suggests haploniscids live at or near the sediment surface (Harrison 1989). Finally, the Macrostylidae Hansen, 1916,

due to their infaunal tubicolous mode of life, likely have the least dispersal potential and thus distributional ranges may be very limited. Their sexual dimorphism may allow males of some species to be more mobile on the suprabenthos compared to the females however (Harrison 1989; Hessler and Strömberg, 1989; Riehl & Kaiser, 2012; Bober et al. 2018).

In a previous molecular assessment of wide-spread isopod species across the Mid-Atlantic Ridge (MAR), Bober et al. (2018) found lifestyle to have a profound effect on dispersal distances, with munnopsid species maintaining gene flow across the

MAR, while distributional ranges in desmosomatids, nannoniscids (Brix et al. 2018) and macrostylids were much more restricted (but see Riehl et al. 2017). Thus, we expect munnopsid species to exhibit the widest geographic distributions compared to other families. Furthermore, we expect to find the correlation between geographic and genetic distance to be more pronounced in lineages with limited dispersal ability (Haye et al. 2012; Janssen et al. 2015; Riehl et al. 2018). In the absence of detailed information on species' distributional ranges in the CCZ, and the abyss in general, using lifestyle as a dispersal

ability proxy would be highly beneficial to forecasting faunal recolonization potential following disturbance events and related environmental changes. This information would be essential for conservation planning.

The objective of this study is to identify the distributional ranges of four different deep-sea janiroid families (Munnopsidae, Desmosomatidae, Haploniscidae, and Macrostylidae) with varying lifestyles and to determine if these can be used as a proxy



to estimate dispersal distances. Samples were collected during two expeditions in the course of the JPI Oceans Pilot Action
"Ecological Aspects of Deep-Sea Mining" (JPIO) to the CCZ and DISCOL Experimental Area (DEA) area in the northeastern and southeastern Pacific respectively. We carried out a molecular analysis of two mitochondrial DNA markers (COI and 16S) backed up by morphological means, to delineate species in an integrative approach. Based on this species delimitation, we test several statistical parameters to gain more knowledge about the species richness and similarity of the different areas.

## 1.1 Material and Methods

Isopod specimens were collected with an epibenthic sledge (EBS) on the CCZ (SO239 cruise, 13 EBS deployments, Table 1) and the Peru Basin (SO242-1 cruise, 9 EBS deployments, Table 1) from the RV Sonne in 2015. In the CCZ the samples were taken in 4 contractor areas, from east to west: BGR (German contractor), IOM (Interoceanmetal Joint Organization), GSR (Belgian contractor), IFREMER (French contractor). In addition, the APEI3 (Area of Particular Environmental Interest number 3) was sampled. The sediment samples were immediately fixed on deck in pre-chilled 96% non-denatured ethanol and kept
cool throughout the sorting process according to Riehl et al. (2014). One to three posterior legs (natapods) of each isopod specimen were dissected and used for DNA extraction. Before DNA extraction all isopod specimens were morphologically determined to family level and given individual voucher numbers. All voucher specimens are stored at the Center of Natural History, Hamburg (CeNak) or the crustacean collection Senckenberg, Frankfurt (Table 1). After DNA extraction, all isopod specimens were identified morphologically to species level using a LEICA MZ 12.5 stereomicroscope by SB, NB, MM. All
determinations were entered into the excel spreadsheet (table 1) using this as baseline for creating maps in QGIS, as well as for statistical analysis. All specimen information and molecular data are managed via the Barcode of Life database (BoLD) in the projects "CCZ - Clarion and Clipperton Fracture Zones biodiversity" and "DISCO - DISturbance and reCOLonization experiment in a manganese nodule area of the SE Pacific Ocean". For this publication we created a dataset "Dataset - DS-LOCOM Locomotion of adult isopods influences distribution" holding a subset of 619 specimens for GenBank submission
and making the sequences visible after publication. All data are stored in the BoLD along with a project OECID, which contains all available data and is made publicly available via GenBank submission. The BIN system in BoLD compares newly submitted sequences with all already available sequences in BoLD clustering them according to their molecular divergence using clustering algorithms. Each cluster receives a unique and specific BIN (barcode identity number as stated for each specimen with COI sequence in table1).

Outgroups for each family tree consisted of the following: Macrostylidae = *Thaumastosoma diva* KY951731, *Thaumastosoma platycarpus* IDesm10, *Ketosoma vemae* VTDes013 (16S only), KM14-Iso261 *Ketosoma* sp. 2, KY951731, and *Ketosoma hessleri* KY951729. Haploniscidae = *Ianiropsis epilittoralis* AF260835, AF260836, AF260858, and AF260859. Desmosomatids = *Betamorpha fusiformis* EF116524, EF116525, EF116527, EF116528, and *Betamorpha africana* EF682292. Munnopsidae = *Thaumastosoma platycarpus* IDesm10, *Ketosoma vemae* VTDes013, *Ketosoma werneri* D3D60 (COI only),



and *Thaumastosoma diva* D3D64 (16S only). Outgroups were chosen based on the most recent evidence for likely sister groups and available sequences.

*Molecular Methods*

A fragment of the mitochondrial gene Cytochrome Oxidase Subunit 1 (COI) was amplified and sequenced using the primers
jgHCO2198 and jgLCO1490 (Geller et al., 2013) following the protocol of Riehl et al. (2014). Ribosomal 16S sequences were amplified and sequenced using the primers 16Sar and 16Sbr (Palumbi, 1992). The sequences were processed using Geneious 11.1.3 and compared against the GenBank nucleotide database. Sequences were aligned using MAAFT 7.388 (Katoh and Standley, 2013) implemented within Geneious v. 10.1.3. COI sequences were translated into amino-acid sequences within Geneious and checked for stop codons to prevent the inclusion of pseudogenes (Buhay, 2009). COI and 16S datasets were
used individually for VSearch and ABGD species delimitation analyses and both individually and concatenated as a singled mitochondrial dataset for phylogenetic tree reconstruction and PTP/mPTP species delimitation analyses. Tree estimations for each family were run in RAxML (Katoh and Standley, 2013) using the GTRGAMMA model and 1000 bootstrap replicates. Multiple species delimitation methods were applied to the four datasets and results varied based on the amount of within clade sampling, occurrence of singletons, and within and between clade variation.  VSearch (Rognes et al., 2016) applies a pairwise
identity threshold and generates clusters of sequences that fall within a specified percent identity, thus assuming a barcode gap, though these can be hard to identify in some cases. VSearch was performed on individual genes without an outgroup. ABGD was performed through the online ABGD webserver (http://wwwabi.snv.jussieu.fr/public/abgd/abgdweb.html, 08/18/2018; X = 0.5) on COI and 16S alignments by family. ABGD was performed on uncorrected p-distances using entire datasets under the assumption that the smallest gap in the pairwise distance histogram reflected the boundary between
intraspecific variation (smaller values) and interspecific variation (larger values). Poisson tree processes (PTP) and multi-rate PTP were run using the stand alone mPTP software implementing -single and –multi switch commands on the fully bifurcated trees generated above. Our data contained multiple individuals with the sample haplotypes but the replicate haplotypes can confound delimitation analyses and lead to over-splitting (Marki et al., 2018) so we calculated the minimum branch length for each sequence and used the minimum branch threshold option in order to ignore these replicate branches in subsequent
PTP/mPTP analyses. MCMC analyses were run for 100 million generations, sampling every 10,000 and discarding the first 2 million generations as burn-in. Analyses were initiated using a random delimitation as starting point. We ran 3 MCMC chains for each analysis and assessed chain convergence by checking average standard deviation of delimitation support values (ASDDSV) across the 3 independent MCMC runs, accepting values near zero and below 0.05 as individual MCMC chains appearing to converge on the same distribution of delimitations (Ronquist et al., 2012, Kapli et al., 2017). We inspected the
MCMC output trees and collapsed all putative species clades that had support below 0.70, which resulted in number of supported clades being within the credible range of delimited species (CCI) and the range across CCI where probability is 0.95 (HPD). The ML estimate, on the other hand, was not always within these intervals, meaning that this ML point estimate delimitation was not supported by MCMC analyses (the estimate may instead represent a local maximum or random solution



derived across the ML likelihood surface) and demonstrates the importance of running MCMC analyses. Singletons greatly
affected mPTP analyses but not PTP or ABGD so were removed from mPTP and retained for PTP and ABGD.

*Isopod Communities and diversity analyses*

Analysis of community similarity between areas and their diversity was performed in R using package 'vegan' (Oksanen et al
2008). The sampling effort, expressed as the number of Epibenthic Sledge (EBS) deployments per area was uneven ranging
from 2 to 8 deployments, therefore the similarity between communities was done using relative abundance (Chord distance,
see Legendre & Gallagher 2001) and using 'presence-absence' to explore faunistic differences. Ordination was done using
nMDS. The community table (Appendix_supplement1) shows the number of specimens from each species found adding up
all EBS samples for a given area. As the number of specimens found differs between areas, diversity comparison was achieved
using rarefaction curves, together with standard diversity indices Shannon, Simpson and Jaccard's Evenness. The expected
number of species per area was inferred using extrapolation methods. Chao1 (Chao 1994, Colwell & Coddington 1994) uses
the proportions of singletons and doubletons in the sample to estimate expected species richness, while ACE (Chazdon et al
1998) is an abundance-base coverage estimator. For the analysis of beta (regional) diversity, the total multiple-site beta
diversity $\beta_{SOR}$ was calculated using the modified Sørensen Index (Sørensen 1948, Balseaga & Orme 2012), and $\beta_{SOR}$ was
decomposed into its additive components "multiple-site species turnover" $\beta_{SIM}$ (Simpson Index, Simpson 1943) and "multiple-
site nestedness" $\beta_{SNE}$ using R package 'betapart' (Balseaga 2010, Balseaga & Orme 2012). In order to explore the relative
contribution of every area to species turnover and nestedness, these values were calculated taking one area out each time in a
jackkniffe approach. Changes in turnover and nestedness are attributable to the area each time excluded from the analysis.

**1.2 Results**

In total, we examined 22 EBS samples taken at six abyssal areas. The dataset comprised 619 specimens belonging to 168
putative species (Table 1). The Munnopsidae accounted for 51 % of the species (congruent) and 48 % of the specimens,
Desmosomatidae accounted for 30 % of the species and 23 % of the specimens, Haploniscidae accounted for 23 % of the
species and 14 % of the specimens, while the Macrostylidae accounted for only 5 % of the species and 15 % of the specimens
(Table 4). Desmosomatids were the most diverse group with 0.36 species per specimen (congruent). Haploniscids and
munnopsids were nearly as diverse as the desmosomatids with 0.26 and 0.29 species per specimen, while macrostylids were
the least diverse with only 0.09 species per specimen. If you remove species represented by a single specimen (singletons)
from the species counts and compare total species numbers to number of species represented in more than one collection
location, you see that 52 % of desmosomatid species were found in more than one collection location, while only 38 and 37 %
of macrostylid and munnopsid species, respectively, and only 26 % of haploniscid species were found in more than one
collection location.

*Molecular data*



All isopod families were reciprocally monophyletic (Figs. 2-5). As expected with fast evolving genes such as COI and 16S, good resolution is given at the tips of the tree and most recent relationships such as species and sometimes even generic level. However, no resolution of relationships deeper in the trees was obtained. Given that the research question here is one of species delimitation, we did not attempt to find markers that would resolve deeper nodes in the trees. It is notable that the percentage of species new to science is quite high and reaches more than 87 % in our dataset. In total, as many as seven of the 187

delimited species were described either previously from other deep-sea locations or even based on CCZ material (Malyutina et al. submitted; two new spp. within the new genus *Pirinectes* Malyutina & Brix gen. nov., Riehl & De Smet in press for *Macrostylis* cf. *metallicola*, Brix et al. 2018 for *Eugerdella* cf. *egoni*). Emphasis is put here on "may be", because the assigned species names are indicated with a "cf." and need more detailed taxonomic verification.

The congruent species delimitation resulted in 86 munnopsid species OTUs (Table 1, Figure 2). Putative species clade definition based on genetic data suggests there is substantial cryptic diversity within the Munnopsidae. Specimens identified

as belonging to *Disconectes* belonged to 14 different putative species, of which those putative species formed 7 clades. Only the singleton and clades with fewer samples came from a single collection region. Specimens identified as belonging to the "catch-all" genus *Eurycope* belonged to 22 different putative species, of which those putative species formed nine higher level clades. One putative *Paramunnopsis* species was collected from three different region while another was collected from two

different regions and was found to be within the same putative species clade as a specimen identified as *Munnopsis abyssalis*. Of the six putative *Betamorpha* species, four were singletons and one contained specimens collected from three different regions. All collected *Bellibos*, belonging to two putative species, were collected from a single region.

The congruent species delimitation resulted in 51 desmosomatid species OTUs (Table 1, Figure 3). The genera *Chelator* Hessler, 1970 (6 spp.), *Oecidibranchus* Hessler, 1970 (1 sp.), *Mirabilicoxa* Hessler, 1970 (12 spp.), *Eugerdella* (kussakin,

1965) (18 spp.), *Disparella* Hessler, 1970 (5 spp.), *Prochelator* Hessler, 1970 (4 spp.) and *Eugerda* Meinert, 1895 (3 spp.) were present in our dataset. Genetically defined clade composition closely mirrored the morphological identification (Figure 3).

The congruent species delimitation resulted in 23 haploniscid species OTUs (Table 1, Fig. 4). The clades represent the genera *Mastigoniscus* (9 spp.), *Haploniscus* (9 spp.) and *Chauliodoniscus* (5 spp.). In Haploniscidae, 100 % of species collected are

new to science.

The congruent species delimitation resulted in eight macrostylid species in this mongeneric family (Table 1, Fig. 5). Putative species "*Macrostylis* sp. 1", collected from both GC area and adjacent to IOM area, was strongly supported as sister to the rest of the available macrostylids. The remainder of the macrostylids formed a single clade that was differentiated into seven individual putative species clades (Fig. 5). Only two of these putative species clades can be easily distinguished from the others

based on morphology, while the rest have yet to have morphological apomorphies identified for them. All eight putative species clades were supported by a minimum bootstrap value of 97 % in the maximum likelihood-based phylogenetic estimations. These eight species are the same that were stable across both COI/16S species delimitation analyses (Osborn et al. in prep. for detailed species delimitation analyses comparing methods and challenges with each family's dataset). It may be possible with





additional sampling to separate the putative species further but based on this dataset, there was not consistent evidence for
further splitting so we chose to be conservative with regard to splitting putative species. Four species clades were
geographically isolated within a single CCZ region (Fig X1, clades 4, 6, 7 and 8), the rest contained members from two to five
regions. There was genetic signal that suggested genetic differentiation between regions within the largest putative species
clade with representatives collected from five regions, but this differentiation, or perhaps our sample size, was not sufficient
to support further species level splits.

*Community and Diversity comparison by area*

The community table (supplement 1) shows the counts of each species by area. The diversity values are summarized in Table
2. A total of 22 sites (EBS deployments) were sampled at 6 areas. Sampling effort was uneven, with most samples taken in the
DISCOL area in the Peru Basin (8). For all other areas 2-4 sites were sampled. A total of 619 specimens could be assigned to
168 species. None of the species was recorded in all 6 areas, while the most common species was
240 Macrostylidae_*Macrostylis*_M05 with 46 specimens (present in all areas besides DISCOL). Other species (see Appendix
supplement 1) with 10 or more specimens were the munnopsids *Disconectes*_Mu11 (22 specimens), *Eurycope*_Mu37,
*Disconectes*_Mu08 (both with 18 specimens), *Munneurycope*_Mu67, the haploniscids *Haploniscus*_H10 (13 specimens each)
*Mastigoniscus*_H22 (with 12 specimens) and with 10 specimens the desmosomatid *Eugerdella*_D39, the macrostylids
*Macrostylis*_M03 and *Macrostylis*_M04. The reminding 177 species had less than 10 records, 68 species were represented by
245 singletons.

The total number of species found was relatively similar between sites ranging from 38 (GSR) to 50 species (IFREMER).
Remarkably the number of species neither correlates with number of specimens (pearson correlation 0.34, p=0.49), nor with
number of sites per area (pearson correlation -0.02, p=0.95). IOM presented the highest number of unique species (species
recorded only in one area) with 36 species (90 % of the species present in the area were unique), followed by DISCOL (31
species, 76 %) and FC (Ifremer; 34 species 68 %). All other areas had less unique species. The extrapolated number of species
present per area ranged between 49 (GC: BGR) and 80 (BC: GSR) according to Chao1, and 53 (GC: BGR) and 80 (FC:
Ifremer) according to ACE. Between 50 % and 12 % of the species remained unrecorded as predicted by Chao1 and ACE.
Diversity values (Shannon, Simpson and Jaccard) are high at all areas. Nevertheless, lowest diversity values were recorded at
BGR area (evenness 0.88, Simpson 0.94, Shannon 3.34) whiles all other area show similar higher values.

Half of the EBS deployments (11) were in the core CCZ area (all areas excluding APEI3 and DISCOL), but these accounted
for (not half, but) 2/3 of the specimens (425) and 2/3 of the species (117) recorded. A total of 99 species (84% of all species)
were found exclusively in the CCZ area. Chao1 and ACE predicted 137-146 species for the CCZ and 235-252 species for all
areas together.

*CCZ areas show highest number of shared species*

Table 3 shows the faunistic similarity between areas. The greatest number of shared species are between CCZ areas. For
instance, GSR shares 16 species with each of BGR and IOM areas, and 11 species with IFREMER. While GSR shares only 4
species with DISCOL and 2 with APEI3. The highest numbers of non-shared species (mean 80.4 ±4.3) are found between





APEI3 and any other area, followed by DISCOL (mean 78.4 ±6.5) although t-test shows no significant difference between them (p=0.58).

*High diversity in each area is reflected by low similarity between sampling areas*

Total multi-site beta diversity was high (total βSOR 0.885, Table 2), meaning that the similarity between areas was low. The beta diversity between CCZ only areas was lower (total βSOR 0.767) revealing slightly higher congruence between areas in the CCZ. In both cases the highest proportion of beta diversity is due to species turnover (βSIM) with only a small proportion accounting for nestedness (βSNE), but the nestedness proportion is 3 times greater within CCZ areas (βSNE = 0.021) than

270 when considering all areas together (βSNE = 0.007). This is also evidenced by removing the areas one by one and calculating beta diversity with the reminding areas only. Removal of APEI3 and DISCOL results in the highest increase in nestedness (βSNE goes from 0.007 to 0.011), while the removal of any of the CCZ areas either does not change βSNE or it decreases up to βSNE = 0.004.

*The known unknown: real number of species still not sampled*

Rarefaction analysis (Figs. 6, 7) shows that all areas are similar in terms of species richness. The lowest curve being BGR (slightly lower diversity) and the highest being IOM. Neither curves show signs of having reached an asymptote.

*Distance matters: APEI and DISCOL more different than central CCZ claims*

Community analysis using Chord distance was ordinated in an nMDS diagram (Fig. 8), showing the more similar CCZ areas clustering together and the more different DISCOL and APEI3 distinctly apart from each other and from the CCZ areas. Not

so evident is the pattern in the presence/absence ordination (Fig. 9) because of the high dissimilarity between areas. The ordination is highly influenced by the number of unique species, highest at IOM, lowest at BGR along the y-axis and other areas spread along the x-axis. The box-plot shows highest median presence/absence dissimilarity to other areas at APEI3, DISCOL and IFREMER areas. The boxplot (Fig. 10) shows that the median Chord distance of the area to any other areas is greater at APEI3 and DISCOL and smaller at any of the CCZ areas.

*Comparison by family, species ranges and beta-diversity*

The species abundance diversity greatly differs between families (Table 4). Munnopsidae was the most abundant and diverse family with 294 specimens (199 in CCZ) belonging to 91 species (55 in CCZ), followed by Desmosomatidae with 143 specimens (193 in CCZ) belonging to 63 species (43 in CCZ). These latter families have a similar diversity as evidenced by the by-family rarefaction curve (Fig. 6). Differences in diversity between these families are due to differences in abundance

rather than species richness.

The family Haploniscidae is less diverse and was present with 88 specimens (53 in CCZ) belonging to 24 species (14 in CCZ). The family Macrostylidae, although relatively common, is much less diverse, 94 specimens (70 in CCZ) belonged to only 9 species (5 in CCZ). The rarefaction curves of these two families show signs of saturation. This is also indicated by the predicted number of species by Chao1. No additional (unseen) species of Haploniscidae and Macrostylidae are expected in the present

dataset, while the expected number of Munnopsidae and Desmosomatidae is 110 and 98 respectively.



Total beta diversity (βSOR) and species turnover (βSIM) increases in this sequence Munnopsidae (βSOR = 0.873; βSIM = 0.860), Desmosomatidae (βSOR = 0.904; βSIM = 0.895) and Haploniscidae (βSOR = 0.916; βSIM = 0.898). This pattern is not evident when comparing within the CCZ (see Table 4). Macrostylidae has lower beta diversity and species turnover (βSOR = 0.809; βSIM = 0.777) mainly due to a single species that shows a large distribution range (see discussion).

The ranges of distribution differ between families (Table 5). No species of either family was present in all six studies areas. While only one species belonging to Munnopsidae was present in five areas (absent in DISCOL). The most widely distributed species of Desmosomatidae was present in all four CCZ areas (no desmosomatid was present in five or six areas). The proportion of species present in a single area increased in this sequence Munnopsidae (75.8%), Desmosomatidae (77.7%) and Haploniscidae (83%). A total of 6 (66.6%) out of 9 species of Macrostylidae was recorded in a single area. In total 77% of the
species were recorded in a single area, 13.9% in 2 areas, 5.3% in 3 areas, 2.6% in 4 areas and 1% in 5 areas.

### 1.3 Discussion

The most common biological unit is the "species". A general public understanding of a healthy ecosystem is to have many species living in it. The definition of "what is a species" is tricky, and often discussed, more than 20 species definitions exist (summary in Fišer et al. 2017). With our SD analysis, we provide a stable system to define a species in the deep sea as baseline
for a more closer ecological view on the samples.

Our results indicate, that life-style, and more precisely the locomotion (dispersal) capabilities of deep-sea asellotes are structuring their biodiversity patterns at medium and large scales. The family Munnopsidae is the most mobile of the four families studied here. They possess large swimming legs and can be observe swimming in the deep-sea water layers on ROV videos (citation here). The second most mobile family is the Desmosomatidae. They live on the surface of the sediments, but
have appendages modified for swimming (not as pronounced as Munnopsidae). On contrary there is no evidence that Haplonoscidae can swim. These asellotes live on or in the sediments and have short legs that they use for crawling. Macrostyllidae live most probably in tubes into the sediment, although some males of this family are good swimmers.

The diversity patterns seems to correspond with these differences in locomotory capabilities. Munnopsidae is the most abundant and diverse family. We do not believe that this perception is biased by the sampling gear, because Munnopsidae are
nevertheless most of the time sitting on the sediments were they feed on foraminifera and only swim occasionally (citation). The enhanced locomotory capabilities of Munnopsidae will result in an enhance connectivity between areas in the CCZ. It may well be that the species have large distribution ranges and therefore the probability of finding them in the EBS samples is higher. Desmosomatidae are less abundant and therefore also diversity is a bit lower. But species richness of Desmosomatidae is as large as Munnopsidae as shown in Figure 6. Finally, the families with reduced dispersal capability have remarkably less
species diversity in the area and the species have much more restricted distributions. Haploniscidae have 83% of the species present in a single area, while this percentage is lower for Desmosomatidae (77%) and Munnopsidae (75%). Macrostylidae deserves as special mention. This sediment dwelling family displayed an unsusal low diversity. Only nine species were





recorded although 94 specimens were analysed (more specimens than Haploniscidae). And these species had a remarkably small range of distribution, as 6 out of 9 species were found in a single area and two species in two contiguous areas. This
pattern would have reinforced our hypothesis. But one of the species is present in as much as five areas, the so called "sp. M" (OTU M04-M07, see Table 1, Fig. 5). In our study, low morphological variation is contrasted by genetic differentiation in *Macrostylis* sp. M (cf. *metallicola*), which belongs to at minimum three different species according to our SD. If a real affiliation to *metallicola* can be provided for clade 5 this would follow the wide distribution of this species across the CCZ according to Riehl and De Smet (in review). The authors state that also in their distribution data, they find molecular hints of
*metallicola* being a complex of more (cryptic) species and thus, the morphological uniform appearance leads to underestimating biodiversity or we may observe ongoing radiation processes. However, as a result any mining impact on the populations of this species would disturb this process or limit the genetic potential of the population and thus, cause changes in the radiation and distribution pattern of this species.

Genetic differentiation in Macrostylidae: Riehl & Kühne (in review) state that two species from the North Pacific Ocean are
indicated to be one in reality: *Macrostylis ovata* and *M. grandis* specimens were genetically not distinct but identical or highly similar. Differences of below 1 % uncorrected p distance and single mutational differences in the 16S marker provided a clear indication that *M. ovata* is a junior synonym of *M. grandis*. This range of intraspecific variability is supported by previous studies on Macrostylidae, which reported up to 8 % p- distance of intraspecific variation (Bober et al., 2018b, 2018a; Kniesz et al., 2018; Riehl and De Smet, under review; Riehl and Kaiser, 2012).

It is important to remark that our result cannot be extrapolated to understanding the global diversity of the families. Even if the global species pool of all four families would be the same, it would be easier to collect more species of Munnopsidae and Desmosomatidae because they have larger distributional ranges than species with smaller distribution ranges like Haploniscidae and Macrostylidae, just because our sampling is limited in space. We only visited a few areas in the abyssal Eastern Pacific.

Little is known about the behaviour of deep-sea asellotes (Hessler and Strömberg, 1989) and most observations come from the morphological descriptions of dead specimens. Especially Macrostylidae, a change to a more epifaunal lifestyle with sexually mature males reproducing with probably stationary females was discussed in species descriptions of strongly sexual dimorphic species (Bober et al., 2017; Kniesz et al., 2017). Sex-specific differences in dispersal capacities are known more from Macrostylidae than from Munnopsidae for which, to our knowledge, no dispersal effecting sexual dimorphisms are apparent.

In desmosomatids and nannoniscids, sexual dimorphism is more pronounced than in munnopsids, for example, males show more adaptations to swimming than females in various species. The species delimitation done on the KuramBIO II dataset for desmosomatids and nannoniscids (Jennings et al. in press) revealed that a strong sexual dimorphism, especially the genus *Mirabilicoxa,* limits morphological species determination and only the integrated approach made a clear assignment to species possible. Thus, determination based on morphological features may underestimate true species richness, which became evident
for macrostylids in our data set. The wide distribution of M05 would have been easily explained it the males would have showed a strong sexual dimorphism, but this was not the case in the individuals available in our dataset.



Haploniscidae show a sexual dimorphism, which is strongly visible in males, while females of different species may have a similar morphological appearance. Thus, species determination sometimes depends on the male specimens and is also not possible in juvenile stages (Brökeland 2010a ,b, Brix et al. 2011).

Brandt et al. (2011) considered the distribution of isopod families sorted by mobility types: walking, swimming, burrowing, walking-swimming, walking-burrowing. There is a low similarity of the Isopoda found on the Maud Rise seamount compared to the other deep-sea stations in the Weddell Sea: especially the comparably "lower mobile" families Macrostylidae and Haplomunnidae were highly abundant. To find two isopod families with comparably restricted active distribution abilities in such high abundances on a seamount top was regarded by Brandt et al. (2011) as unusual.

Bober et al. (2018) used only one munnopsid species with a known pan-ocean distribution, *Acanthocope galatheae* Wolff, 1962 (Malyutina et al. 2018) while for desmosomatids, nannoniscids and macrostylids the complete amount of available species and specimens (>400 specimens for both families, nannonicids and desmosomatids, resulted in 72 species for COI and 45 for 16S by species delimitation according to Brix et al. 2018) was used. Our dataset used the complete set of specimens available in the family Munnopsidae and revealed that this swimming family is the only one showing potential species with

atlantic-pacific distribution (in case of *Acanthocope* cf. *galathea*) and also showing the distribution over the largest distances. On genus level, pacific-atlantic distribution has been reported also within the nannoniscids for species of two genera possessing swimming legs in a strong sexual dimorphism (Kaiser et al. 2017). Due to their prevailing reproduction mode (brooding) coupled with putatively poor swimming abilities most of species within the sister-family of Desmosomatidae, the Nannoniscidae (most species of this family have walking legs), we expected to find strong population divergence or even

presence of cryptic lineages in relation to distance. Kaiser et al. (unpublished, personal communication) show that two *Nannoniscus* lineages show wide geographic distribution (>1400 km apart), but there is also evidence for cryptic lineages in close vicinity (same licence area); some evidence that geographic distance is important, but also heteeogeneity and oceanographic currents (Taboada et al. 2018).

Although we are dealing with a brooding taxon here and may not discuss larval distribution, adult forms of these small faunal

species (2 – 10 mm average size), will be influenced by currents when moving actively in the water column. Etter & Bower (2015) tested the distance of distribution during the PLD in the North Atlantic Ocean using physical particals as models. This experiment resulted in a possible distribution over hundreds of kilometres and even through current systems with a strong temperature gradient. Thus, actively swimming taxa are more likely not depending on any watermass or current system as already indicated by Schnurr et al. (2014, 2018) for the subarctic region around Iceland. For other assellote families in the

present dataset, water masses did play a major role shaping distribution patterns, more than benthic surface structure (Brix & Svavarsson 2010) while for other, sediment types are most important as outlined by Stransky & Svavarsson (2010).

Based on two separately treated genetic datasets of Macrostylidae and Desmosomatidae/Nannoniscidae from the central Atlantic Ocean, Bober et al. (2018) found most species at only one side of the Mid Atlantic Ridge (MAR). The MAR seems to be a dispersal barrier for the non-swimming Macrostylidae and weakly-swimming Desmosomatidae and Nannoniscidae.

However, four species of Macrostylidae and Desmosomatidae did cross the MAR, but evidence for regular unrestricted gene



flow is lacking. Brix et al. (2018) observed from SD data for desomomatids and nannoniscids of the VEMA fracture zone in the North Atlantic Ocean that even robustly-sampled species exhibit "small" ranges of around 500 km, and three species were distributed on the order of 1000–2500 km. Interestingly Wilson (2017), for the pacific abyss, measured the rate of species turnover. Isopods change at a rate of 0.012 species per km, this gives an approximate linear species range of 84 km. Assuming

circular distribution this gives an isopod species range of 2,228 km$^2$.

Some deep-sea taxa are reported to have broader ranges compared to the shallow-water taxa (Costello and Chaudhary, 2017). Either this might be an artefact of species misidentification or a result of the evolutionary history of these deep-sea species. Another hypothesis to explain the broad horizontal ranges of some deep-sea species was the "thought to be homogeneity" of seafloor habitats and stable abiotic conditions in temperature, salinity and pressure (McClain & Hardy 2010). The suggestion

of Carney (2005) that abiotic and biotic factors vary greatly with depth and this restricts vertical ranges of many species despite the potential for broad horizontal distribution ranges as also discussed for isopods along the Kuril -Kamchatka Trench (Bober et al. in review, Jennings et al. in press) as well as in polar regions (Brix et al. 2014).

Nearly half of the deep-sea bivalve and gastropod that have a larval stage in the North Atlantic Ocean have wide distribution ranges along an entire basin (Rex 1981) or even show a pan-Atlantic distribution (Jennings & Etter 2014). The same pattern

is observed for cirriped crustaceans distributing along currents in the South Pacific/Indian Ocean along the hydrothermal vent chains (Suzuki et al 2018) or along currents in the North Atlantic underwater mountain chains (like the acorn barnacle *Bathylasma* cf. *hirsutum*, Brix pers. observation). On the other hand, these unique deep-sea habitats such as vent sites, seamounts, hard rocks or cold-water coral reefs may limit the distribution ranges of species because their geochemical cycles and biological activity promotes restricted ranges and isolation, generating highly endemic faunas (McClain & Hardy 2010).

Endemism in the deep-sea habitats is known and describes as "rare" species (Brandt et al. 2007) those species occurring at only one sampling point with only one individual. This phenomenon is also observed in our dataset in each of the four families. It has been discussed as sampling bias due to patchiness of distribution by Kaiser et al. (2009) for EBS samples from the Southern Ocean. Therefore, it is not clear whether the high number of so called "singletons" in our dataset is true endemicity or a result of sampling bias. Pelagic species and pelagic life stages of many benthic species can drift and swim across and/or

between oceans during their lifetime. Benthic species, however, spend most of their life on the seabed, and thus may be dispersing shorter distances (Costello et al., 2017). However, it has to be noted that there is – even more than in the VEMA dataset - a large distance between the sampling locations (inside CCZ/DISCOL) and the likely patchiness (Kaiser et al., 2009) cannot be sufficiently inferred based on our analysis – especially because in the CCZ dataset not every specimen was sequenced (in the DISCOL dataset yes, but with a lower success rate than in CCZ). Nevertheless, our dataset represents the most

comprehensive dataset for the deep sea so far. Nevertheless, we are still facing the problem of undersampling the real number of species (Fig. 6, 7), may be except for Macrostylidae (Fig. 6).

Compared to all other asellotan isopod families, munnopsids are highly specialized for swimming and accordingly, some species have moved towards a benthopelagic (e.g., in *Munnopsoides* Tattersall, 1905) or even holobenthic (e.g., in *Paramunnopsis* Hansen, 1916) mode, while others follow a burrowing (e.g., in *Ilyarachna* Sars 1869, or *Bellibos* Haugsness





& Hessler, 1979), or epibenthic (e.g., in *Rectisura* Malyutina, 2003 or *Vanhoeffenura* Malyutina, 2004) life style (reviewed in Osborn 2009).

For the swimming Munnopsidae Bober et al. (2018) were able to detect persistent gene flow across the MAR in the example species *Acanthocope galathea* Wolff, 1962. Specimens were collected along a latitudinal transect crossing the tropical abyssal North Atlantic during the Vema-TRANSIT expedition (Malyutina et al. 2018). For *Acanthocope galathea* a persistent gene

flow over a vast geographic distance of 1,843 km is assumed in the VEMA fracture zone. This species is also available in the Pacific dataset and may be regarded as world-wide distributed (as indicated from the genetic data in our pacific dataset) or alternatively as putative cryptic species due to the large genetic distances in the Atlantic and Pacific datasets.

Malyutina et al. (submitted) described a new genus and two new species of the munnopsid subfamily Eurycopinae from the CCZ material. The new genus was revealed by the molecular SD independent from the taxonomic investigation and

440 morphological analysis and was independently confirmed by the molecular SD approach. In previous SDs for Desmosomatidae (Brix et al. 2018, Jennings et al. in press), the genera clustered well together, representing most probable relationships (Hessler 1970) as well as showing the taxonomic problem in the case of *Eugerdella* (Brix et al. 2018) and *Mirabilicoxa* (Jennings et al. in press). In the case of desmosomatids and nannoniscids, most comparable to the present dataset is the VEMA dataset in a horizontal distribution calculating species ranges, while the KuramBIO II dataset is limited by a vertical distribution of species

showing a strong bathymetric influence (factor depth) on species distribution as already stated for several peracarid taxa in other regions of the world (Brix et al. 2014 for *Chelator insignis*, Havermann et al. 2013 for *Eurythenes gryllus*).

**Conclusion**

Cardaso et al. (2011) list seven reasons why invertebrates are rarely included in present-day conservation. We focus on the most common and most fundamental drawback: taxonomic incompleteness. Our study of a community wehre over 87 % of

450 the isopod species are new to science or described within the last two years, indicates the need for quick assessment tools like species delimitation in the deep-sea environment. Additionally, taxonomic expertise is needed, which can lead to a description of the key species even though it is not possible to describe every species due to time constraints (Brix et al. 2018). If no SD or other rapid assessment method is possible due to constraints in the sampling method or fixation of the samples, the taxonomic incompleteness leads to incomplete knowledge of species distributions, ecology, population dynamics, but also

lower public interest in those species. Even though taxonomic incompleteness is an old and well-known problem in conservation, molecular taxonomy (e.g. Fujita et al. 2012, Fontaneto et al. 2015) has unveiled that the taxonomic impediment may be much deeper than previously thought. Our dataset shows the known "unknown" living in 5000 m depth on the deep-sea floor in an area just awaiting more human impact when extraction of the metal resources here begins.

Geographic distance and locomotion type is most important for connectivity of populations. Exceptions like "sp. M" seem to

460 underline a rule that natatory capability allows only the munnopsids to occur in five of the six areas samples. Long-distance populations are more diverse than patchy/local populations. Janssen et al. (in press) stated that in the case of polychaetes with



long- and short-distance dispersal capabilities, large populations are continuously distributed over large geographic scales. Although their analyses (Janssen et al. in press) suggest a similar pattern in isopods, spatial genetic structuring of isopod populations do imply weak barriers to gene flow. They conclude that mining-related habitat destruction will most likely impact

the continuity of isopod populations. This is based on the assumption that ecosystem recovery after major impacts is predicted to occur slowly at evolutionary time scales. As already stated in Blaczewicz et al. (2019), studies on species richness and distribution patterns of small specimen like peracarid crustaceans are indispensable for the conservation of the abyssal ecosystem and for the development of management strategies for sustained commercial activities in the future.

**Author contributions**

Saskia Brix: Manuscript writing, coordination and management of sequence data (in BoLD), quality check, morphological identification, discussing the species delimitations, figures and manuscript writing, preparing the voucher specimens for museum storage.

Karen J. Osborn: Data quality assessment/control, alignments, supervision of species delimitation analyses, tree/species delimitation figures, portions to the manuscript, editing.

Sarah Schnurr: preparing specimens for genetics and sampling on board as well as lab work at the Smithsonian producing the raw data and providing preliminary trees.

Sarit B. Truskey: performing the species delimitation and phylogenetic analyses and preliminary trees.

Stefanie Kaiser: sorting and preparing specimens on board, helping with the morphological species delimitation, discussing the idea and providing ideas in manuscript writing, adding important parts to the text.

Nils Brenke: species determination of DISCOL Isopoda and morphological comparison to the JPIO dataset together with Saskia Brix.

Marina Malyutina: Identification of the Munnopsidae and linking the manuscript to the description of a new genus.

Pedro Martinez: Paper idea and statistical analyses, manuscript writing and statistical figures.

**Sample availability**

No geoscientific samples which are registered as International Geo Sample Number (IGSN) have been used for the manuscript.

**Competing interests**

The authors declare that they have no conflict of interest.



**Acknowledgements**

We appreciate the teamwork for our paper, which was greatly supported by the working groups at the DZMB in Wilhelmshaven
and Hamburg (Germany) as well as at the Smithsonian Institution, Washington D.C. (USA). Amy Driskell and the Smithsonian
lab staff kindly supported us in the very beginning with access to the Smithsonian LAB for Sarah Schnurr, Saskia Brix and
Karen Jeskulke. Amy continued helping with tricky BoLD uploads of sequence data. Magdalini Christoudolu is thanked for
helping with the BoLD management of the whole pacific project from which we extracted our dataset. Nele Johanssen and
Torben Riehl gave us a first impression on the morphological determination of JPIO haplonicids and macrostylids. Karen
Jeskulke not only helped with the lab work at the Smithsonian and the DZMB in Hamburg, she contributed largely to the
databank entries at DZMB HH. We thank Sven Petersen (GEOMAR) for enabling to Saskia the QGIS entries of the shape file
for the contractor areas. The cruises SO239 and SO242 were financed by the German Ministry of Education and Science
(BMBF) as a contribution to the European project JPI Oceans ''Ecological Aspects of Deep-Sea Mining''. The authors
acknowledge funding from BMBF under Contract 03F0707E.



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



**Figures**

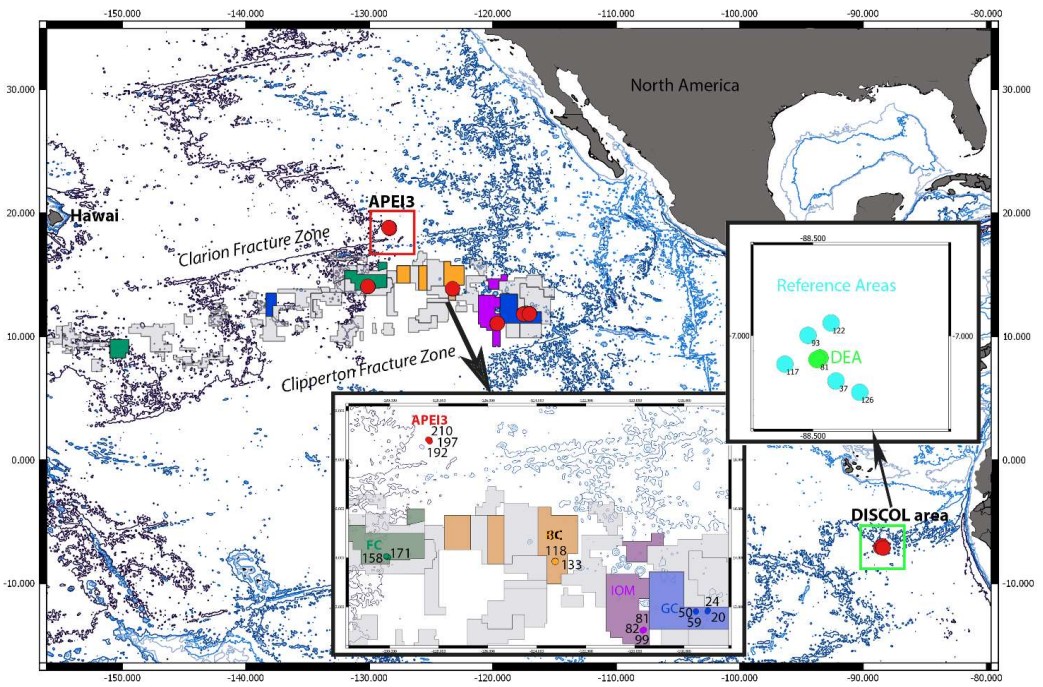

**Fig. 1: Map of the locations of the EBS sampling sites (red dots) within the manganese nodule contractor and the DISCOL Experimental Area (DEA) areas in the north- and south-eastern Pacific. The colourcode in this map reflects the colourcode given in the circle trees (Fig.s 2 – 5), but is not reflected in the statistical graphs (Fig.s 6 – 10). In the CCZ the samples were taken in four contractor areas, from east to west: GC (dark blue - German contractor: BGR), IOM (violet - Interoceanmetal Joint Organization), BC (orange - Belgian contractor: GSR), FC (dark green - French contractor: IFREMER). In addition, the APEI3 (red - Area of**
**Particular Environmental Interest number 3) and DISCOL Experimental Area (light green/blue – DEA and Reference Areas).**





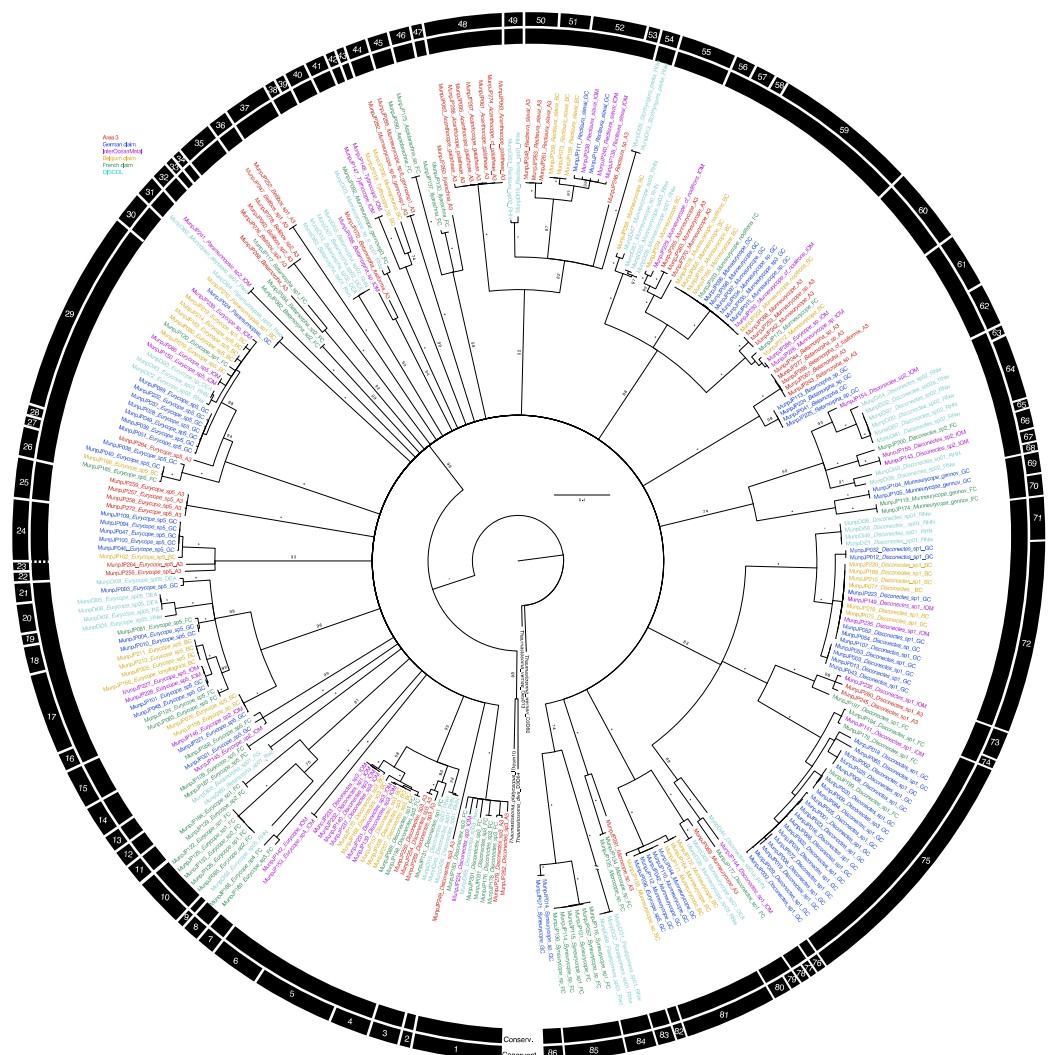

**Figure 2. Phylogenetic tree of all munnopsid samples based on 16S and COI sequences for 294 specimens. Colours indicate collection location, with black indicating outgroups. All unsupported branches were collapsed and bootstrap support indicated with asterisks indicating 100 % bootstrap support. The outer two bars summarize the results of the species delimitation analyses which included morphological determination, Vsearch for individual genes, ABGD for individual genes, PTP and mPTP for both individual genes and the concatenated datasets. The conservative bar indicates that all SD analyses supported that split, while the congruent bar indicates that the majority of SD analyses indicated that split. Numbers on congruent bars are arbitrary and provided to allow a way to refer to specific supported clades.**



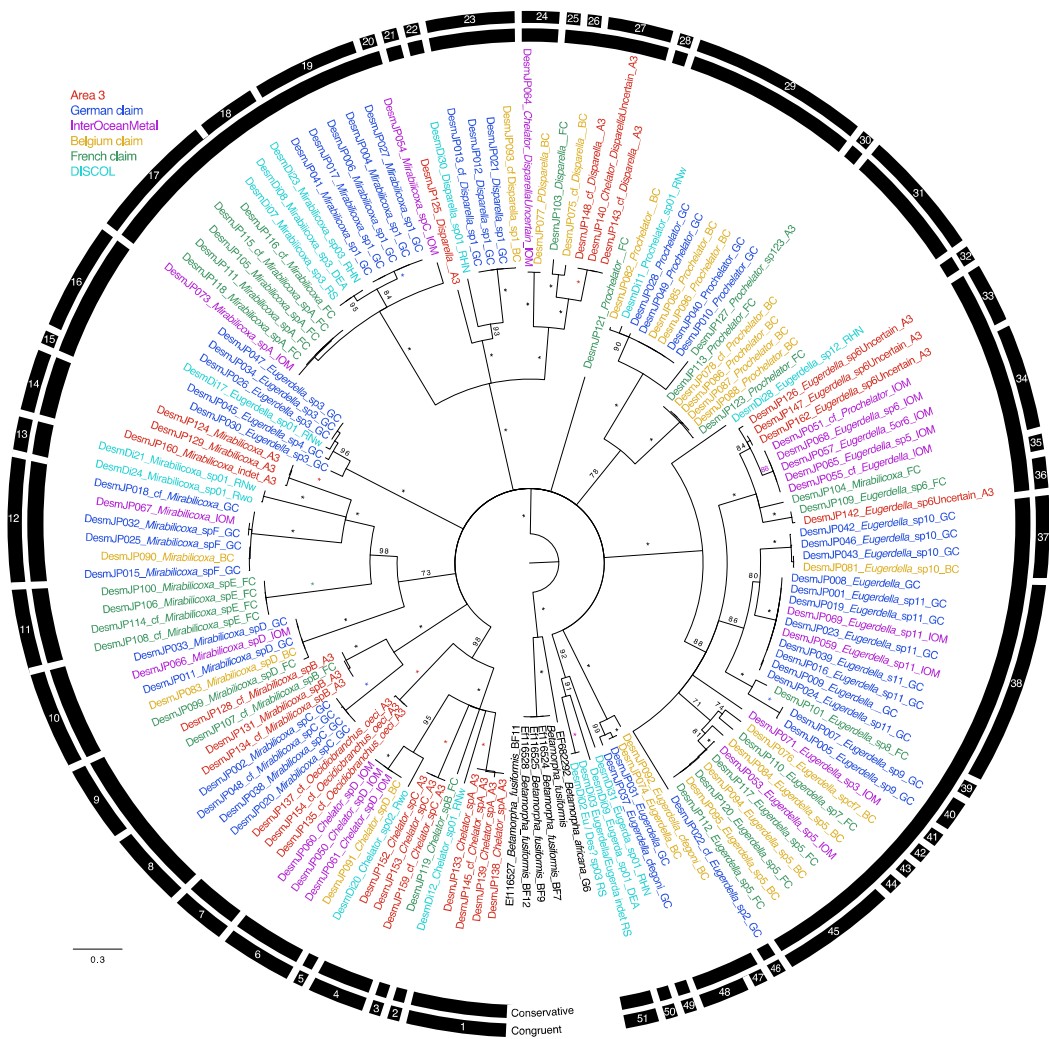

**Figure 3:** Phylogenetic tree of all desmosomatid samples based on 16S and COI sequences for 143 specimens. Colors indicate collection location, with black indicating outgroups. All unsupported branches were collapsed and bootstrap support indicated with asterisks indicating 100 % bootstrap support. The outer two bars summarize the results of the species delimitation analyses which included morphological determination, Vsearch for individual genes, ABGD for individual genes, PTP and mPTP for both individual genes and the concatenated datasets. The conservative bar indicates that all SD analyses supported that split, while the congruent bar indicates that the majority of SD analyses indicated that split. Numbers on congruent bars are arbitrary and provided to allow a way to refer to specific supported clades.





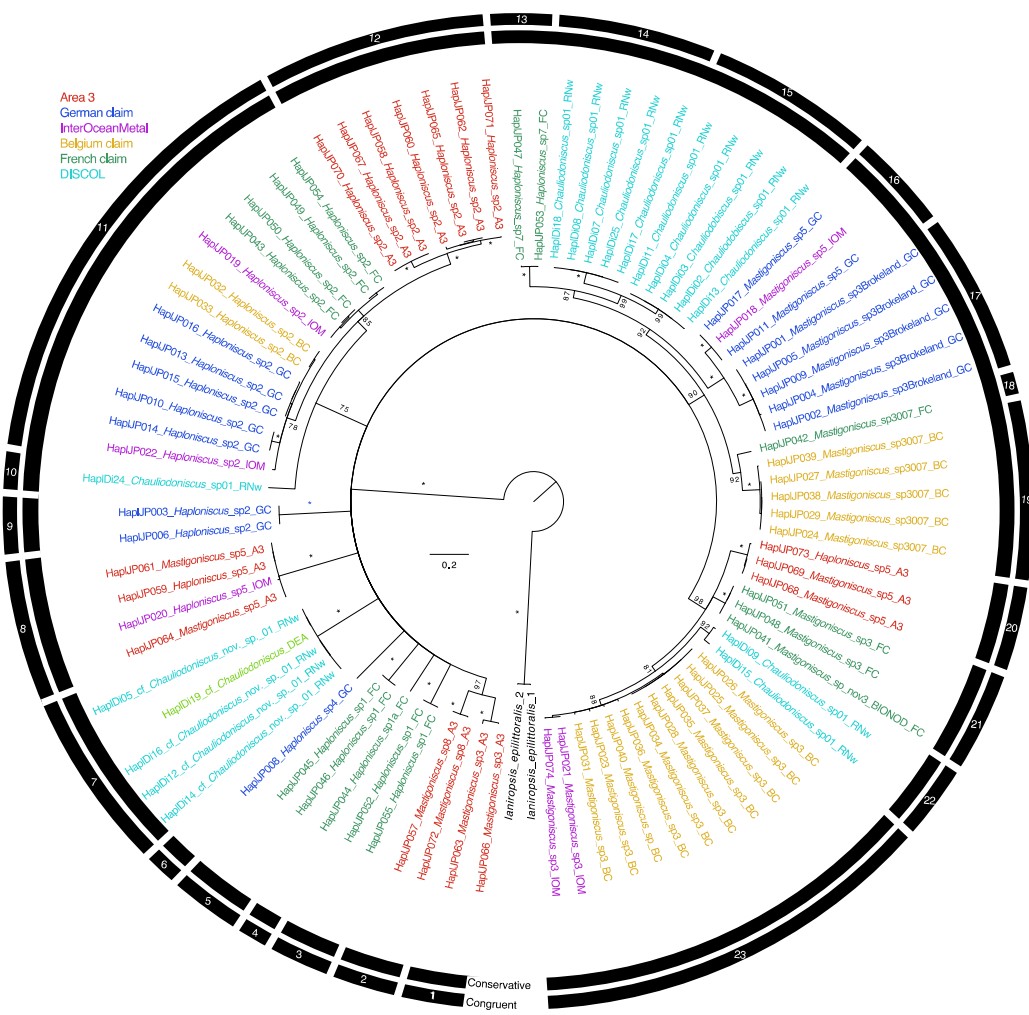

**Figure 4: Phylogenetic tree of all haploniscid samples based on 16S and COI sequences for 88 specimens. Colors indicate collection location, with black indicating outgroups. All unsupported branches were collapsed and bootstrap support indicated with asterisks indicating 100 % bootstrap support. The outer two bars summarize the results of the species delimitation analyses which included morphological determination, Vsearch for individual genes, ABGD for individual genes, PTP and mPTP for both individual genes and the concatenated datasets. The conservative bar indicates that all SD analyses supported that split, while the congruent bar indicates that the majority of SD analyses indicated that split. Numbers on congruent bars are arbitrary and provided to allow a way to refer to specific supported clades.**



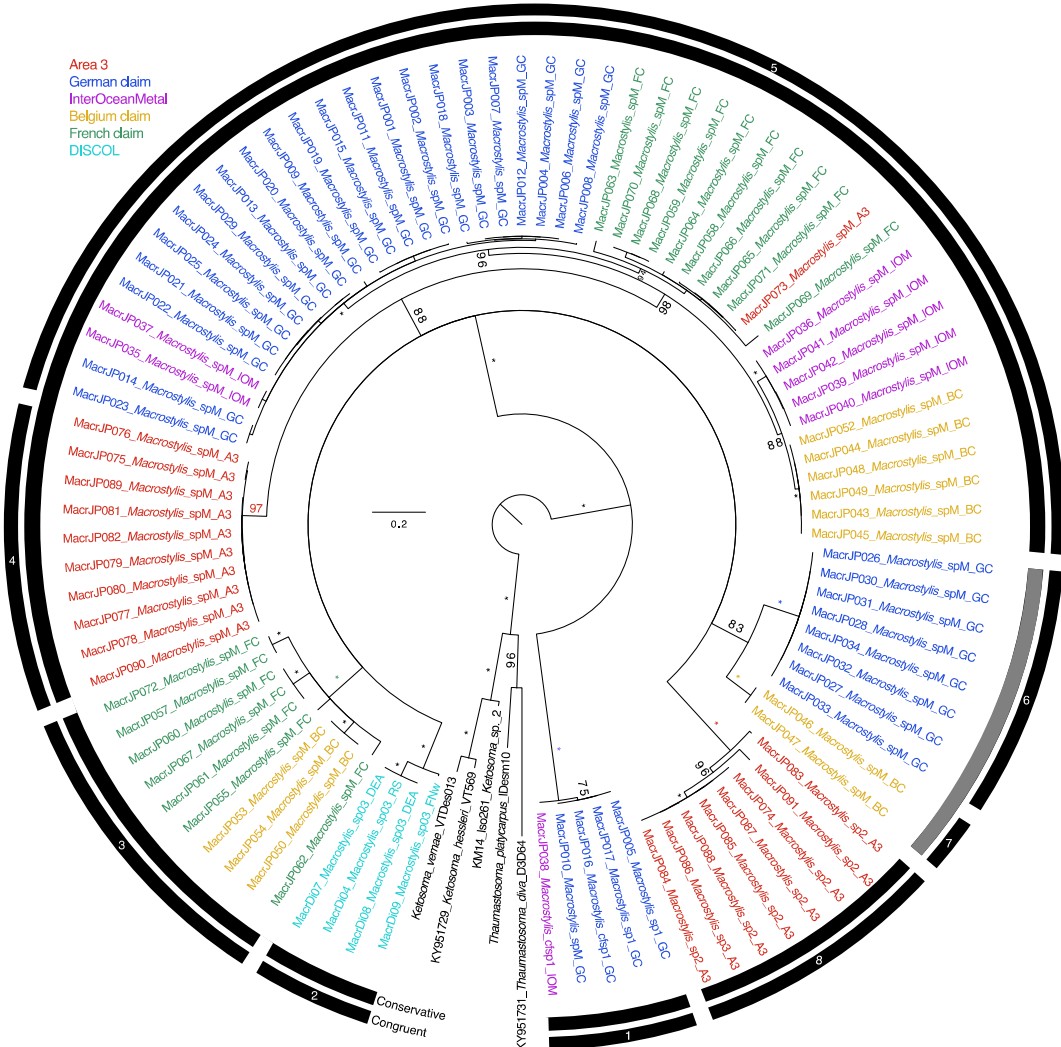

**Figure 5:** Phylogenetic tree of all macrostylid samples based on 16S and COI sequences for 94 specimens. Colors indicate collection location, with black indicating outgroups. All unsupported branches were collapsed and bootstrap support indicated with asterisks indicating 100 % bootstrap support. The outer two bars summarize the results of the species delimitation analyses which included morphological determination, Vsearch for individual genes, ABGD for individual genes, PTP and mPTP for both individual genes and the concatenated datasets. The conservative bar indicates that all SD analyses supported that split, while the congruent bar indicates that the majority of SD analyses indicated that split. Numbers on congruent bars are arbitrary and provided to allow a way to refer to specific supported clades.





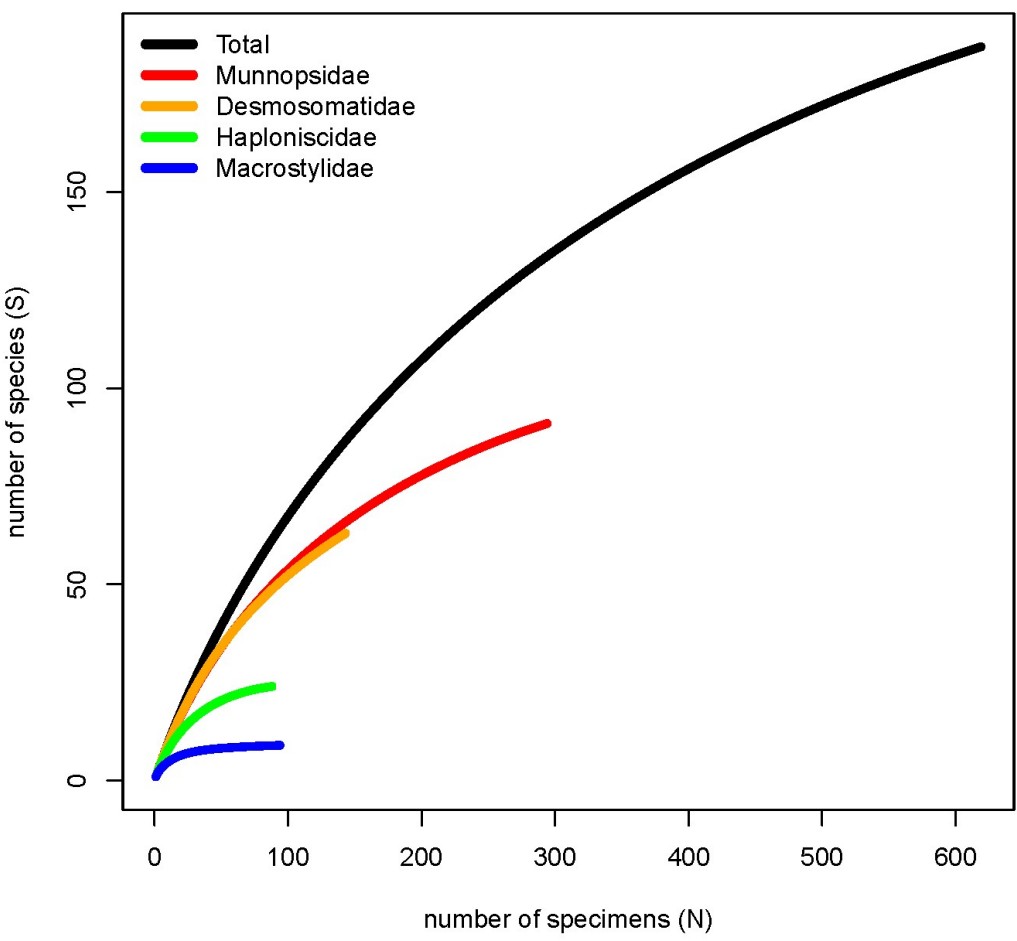

Fig. 6: Rarefaction analysis by isopod family, considering all areas together.





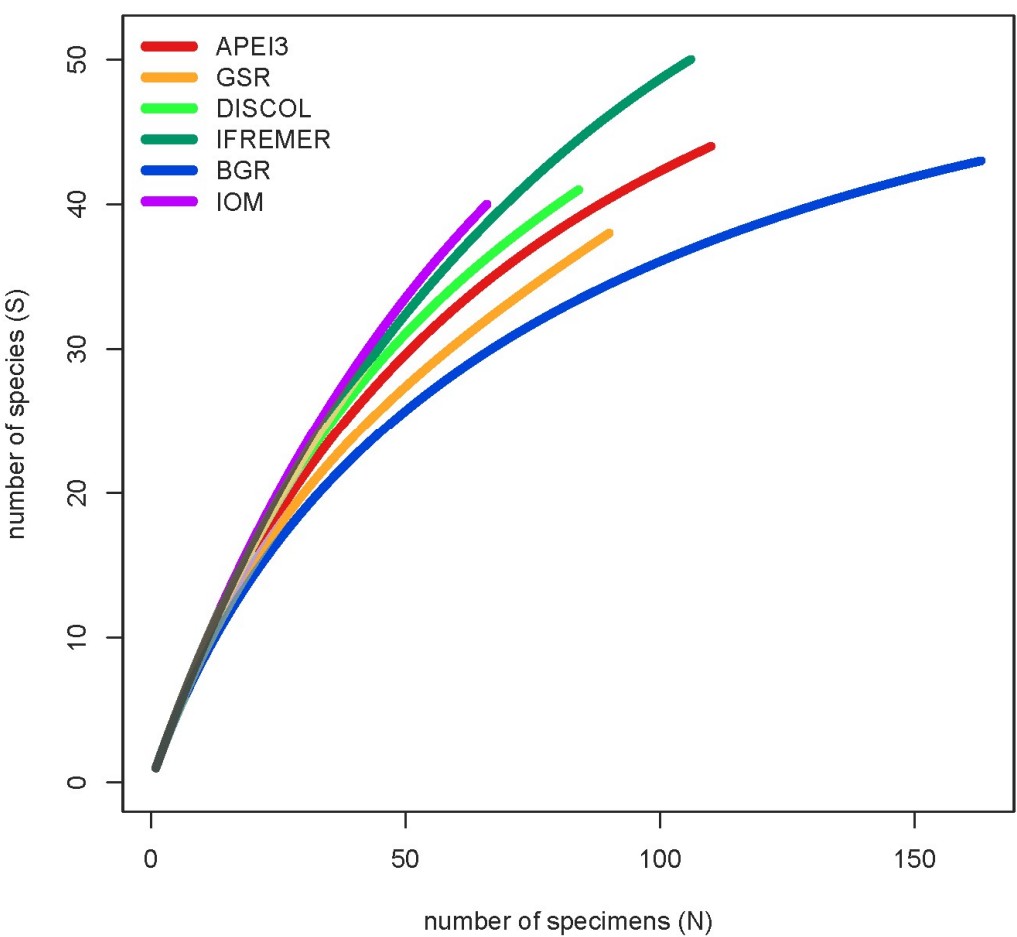

**Fig. 7: Rarefaction analysis by area, considering all families together.**





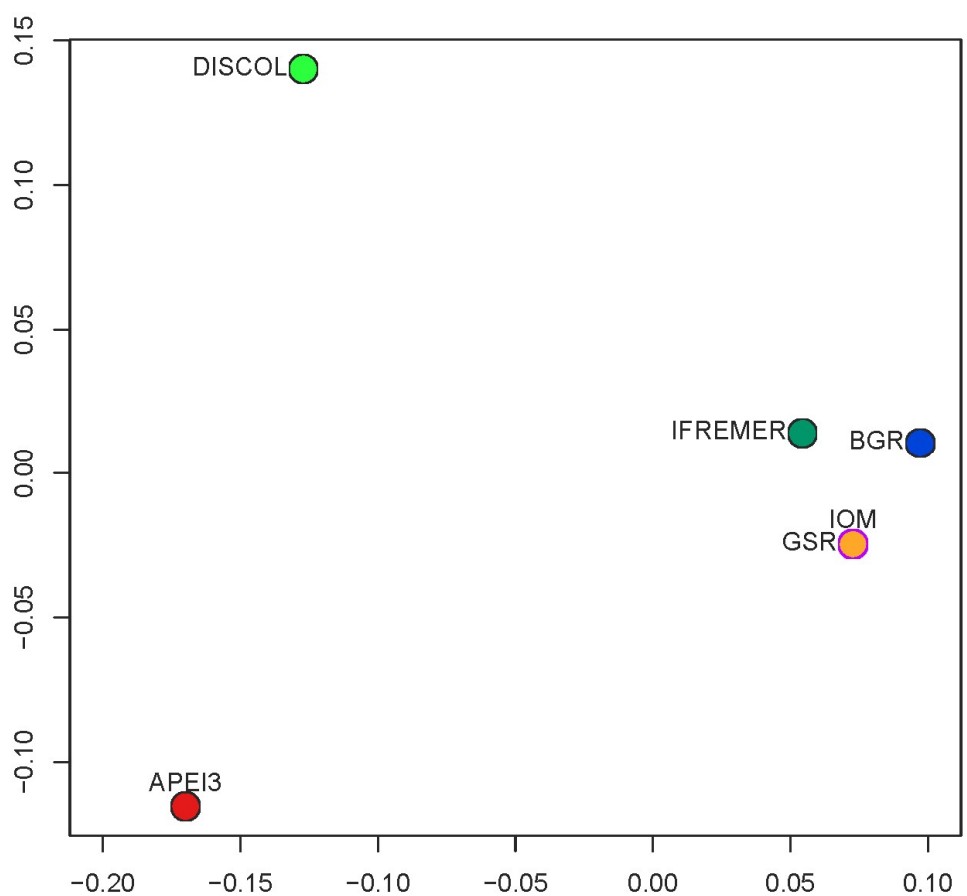

**Fig. 8: nMDS ordination plot of Chord-distance between areas.**



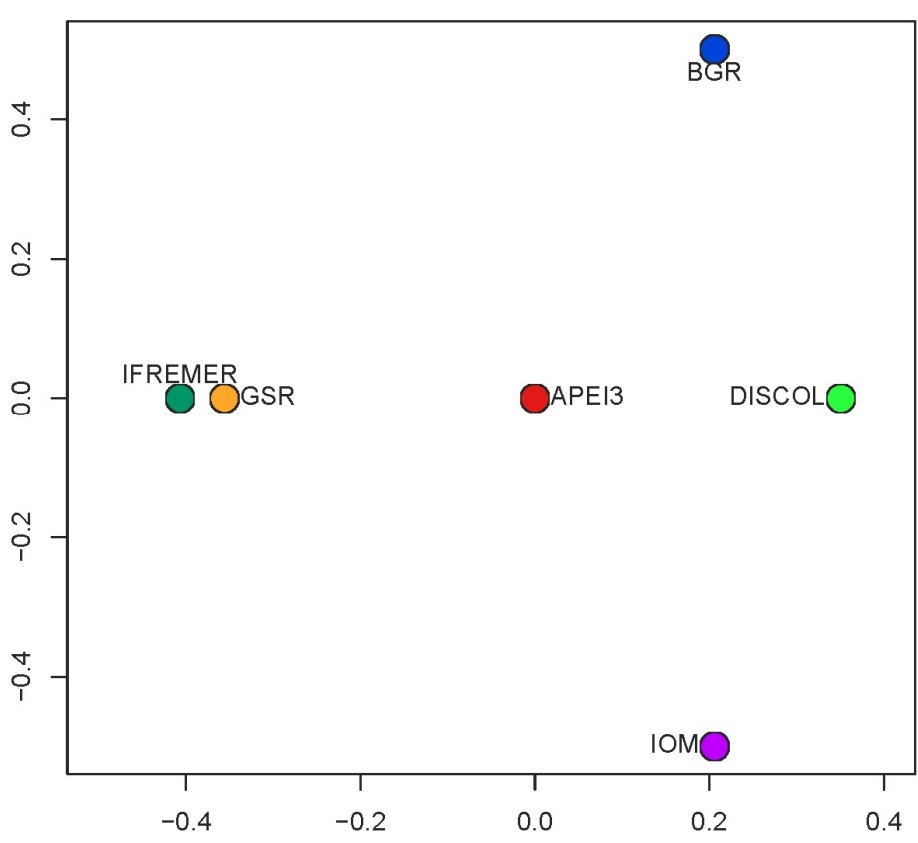

**Fig. 9: nMDS ordination plot of Euclidean-distance between areas of presence-absence transformed data.**





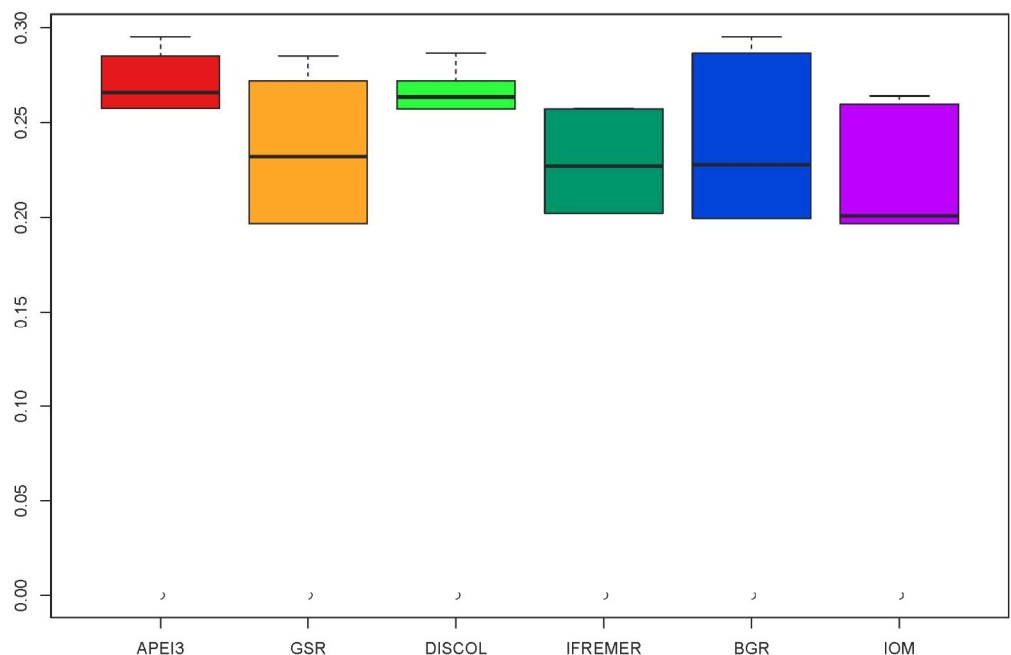

**Fig. 10: Box and whiskers plot showing the median and range of the Chord distance of every area to other areas.**





**Tables**

**Table 1: List of specimens used for this study including all information about station, species identification from morphology and molecular species delimitation (OTUs), museum storage and associated database numbers in BoLD and GenBank.**





| original Field_ID (used in circle trees) | BoLD sample ID | genus | morphospecies | OTU (congruent) | BIN (BoLD) | database_ID DZMB HH | museum catalogue number | family | loco type | RegionCode | station | GenBank no COI | GenBank no 16S |
|---|---|---|---|---|---|---|---|---|---|---|---|---|---|
| DesmDi02 | DSB_1214 | cf. Eugerda | sp. 03 | D52 | BOLD:ADL4889 | 50277 | ZMH K-56556 | Desmosomatidae | WS | DISCOL | 37 | X | |
| DesmDi03 | DSB_1215 | cf. Eugerda | indet | D52 | BOLD:ADL4889 | 50278 | ZMH K-57490 | Desmosomatidae | WS | DISCOL | 45 | X | |
| DesmDi07 | DSB_1219 | Mirabilicoxa | sp. 3 | D19 | BOLD:ADL2574 | 50282 | ZMH K-57491 | Desmosomatidae | WS | DISCOL | 37 | X | X |
| DesmDi08 | DSB_1292 | Mirabilicoxa | sp. 3 | D19 | BOLD:ADL2574 | 50357 | ZMH K-57492 | Desmosomatidae | WS | DISCOL | 81 | X | X |
| DesmDi09 | DSB_1293 | Eugerda | sp. 01 | D51 | BOLD:ADL2893 | 50358 | ZMH K-57493 | Desmosomatidae | WS | DISCOL | 85 | X | X |
| DesmDi11 | DSB_1295 | Prochelator | sp. 01 | D30 | BOLD:ADL2576 | 50360 | ZMH K-57494 | Desmosomatidae | WS | DISCOL | 93 | X | |
| DesmDi12 | DSB_1296 | Chelator | sp. 01 | D03 | BOLD:ADM1410 | 50361 | ZMH K-57495 | Desmosomatidae | WS | DISCOL | 93 | X | X |
| DesmDi17 | DSB_1301 | Eugerdella | sp. 01 | D17 | BOLD:ADL5038 | 50366 | ZMH K-57496 | Desmosomatidae | WS | DISCOL | 93 | X | |
| DesmDi20 | DSB_1345 | Chelator | sp. 02 | D06 | BOLD:ADL9249 | 50410 | ZMH K-57497 | Desmosomatidae | WS | DISCOL | 126 | X | X |
| DesmDi21 | DSB_1346 | Mirabilicoxa | sp. 01 | D15 | BOLD:ADL5368 | 50411 | ZMH K-57498 | Desmosomatidae | WS | DISCOL | 117 | X | X |
| DesmDi23 | DSB_1348 | Mirabilicoxa | sp. 03 | D19 | BOLD:ADL2574 | 50413 | ZMH K-57499 | Desmosomatidae | WS | DISCOL | 122 | X | X |
| DesmDi24 | DSB_1349 | Mirabilicoxa | sp. 01 | D15 | BOLD:ADL5368 | 50414 | ZMH K-57500 | Desmosomatidae | WS | DISCOL | 126 | X | X |
| DesmDi28 | DSB_1353 | Eugerdella | sp. 12 | D33 | BOLD:ADL4885 | 50418 | ZMH K-57501 | Desmosomatidae | WS | DISCOL | 122 | X | X |
| DesmDi30 | DSB_1355 | Disparella | sp. 01 | D23 | BOLD:ADL5538 | 50420 | ZMH K-57502 | Desmosomatidae | WS | DISCOL | 122 | X | X |
| DesmDi31 | DSB_1356 | Eugerda | sp. 01 | D50 | | 50421 | ZMH K-57503 | Desmosomatidae | WS | DISCOL | 122 | | X |
| DesmJP001 | DSB_1762 | Eugerdella | sp.11 | D39 | BOLD:ADW6372 | 50827 | ZMH K-57504 | Desmosomatidae | WS | GC | 20 | X | X |
| DesmJP002 | DSB_1763 | Mirabilicoxa | sp.C | D09 | BOLD:ADL5035 | 50828 | ZMH K-57505 | Desmosomatidae | WS | GC | 20 | X | X |
| DesmJP004 | DSB_1765 | Mirabilicoxa | sp.1 | D20 | BOLD:ADL5037 | 50830 | ZMH K-57506 | Desmosomatidae | WS | GC | 20 | X | X |
| DesmJP005 | DSB_1766 | Eugerdella | sp.9 | D41 | BOLD:ADG0134 | 50831 | ZMH K-57507 | Desmosomatidae | WS | GC | 20 | X | |
| DesmJP006 | DSB_1767 | Mirabilicoxa | sp.1 | D20 | BOLD:ADL5037 | 50832 | ZMH K-57508 | Desmosomatidae | WS | GC | 20 | X | X |
| DesmJP007 | DSB_1768 | Eugerdella | sp.9 | D41 | BOLD:ADG0134 | 50833 | ZMH K-57509 | Desmosomatidae | WS | GC | 20 | X | |
| DesmJP008 | DSB_1769 | Eugerdella | indet | D39 | | 50834 | ZMH K-57510 | Desmosomatidae | WS | GC | 20 | | X |
| DesmJP009 | DSB_1770 | Eugerdella | indet | D39 | BOLD:ADW6372 | 50835 | ZMH K-57511 | Desmosomatidae | WS | GC | 20 | X | X |
| DesmJP010 | DSB_1771 | Prochelator | indet | D30 | BOLD:ADF9936 | 50836 | ZMH K-57512 | Desmosomatidae | WS | GC | 20 | X | |
| DesmJP011 | DSB_1772 | Mirabilicoxa | sp.D | D11 | BOLD:ADL5366 | 50837 | ZMH K-57513 | Desmosomatidae | WS | GC | 20 | X | |
| DesmJP012 | DSB_1773 | Disparella | sp.1 | D24 | BOLD:ADL4887 | 50838 | ZMH K-57514 | Desmosomatidae | WS | GC | 20 | X | X |
| DesmJP013 | DSB_1774 | cf. Disparella | sp.1 | D24 | BOLD:ADL4887 | 50839 | ZMH K-57515 | Desmosomatidae | WS | GC | 20 | X | X |
| DesmJP015 | DSB_1776 | Mirabilicoxa | sp.F | D13 | BOLD:ACY9838 | 50841 | ZMH K-57516 | Desmosomatidae | WS | GC | 24 | X | X |
| DesmJP016 | DSB_1777 | Eugerdella | sp.11 | D39 | BOLD:ADW6372 | 50842 | ZMH K-57517 | Desmosomatidae | WS | GC | 24 | X | X |
| DesmJP017 | DSB_1778 | Mirabilicoxa | sp.1 | D20 | BOLD:ADL5037 | 50843 | ZMH K-57518 | Desmosomatidae | WS | GC | 24 | X | X |
| DesmJP018 | DSB_1779 | cf. Mirabilicoxa | indet | D13 | BOLD:ACY9838 | 50844 | ZMH K-57519 | Desmosomatidae | WS | GC | 24 | X | |
| DesmJP019 | DSB_1780 | Eugerdella | sp.11 | D39 | BOLD:ADW6372 | 50845 | ZMH K-57520 | Desmosomatidae | WS | GC | 24 | X | X |
| DesmJP020 | DSB_1781 | Mirabilicoxa | sp.C | D09 | BOLD:ADL5035 | 50846 | ZMH K-57521 | Desmosomatidae | WS | GC | 24 | X | X |
| DesmJP021 | DSB_1782 | Disparella | sp.1 | D24 | BOLD:ADL4887 | 50847 | ZMH K-57522 | Desmosomatidae | WS | GC | 24 | X | X |
| DesmJP022 | DSB_1783 | cf. Eugerdella | sp.2 | D47 | BOLD:ADL4886 | 50848 | ZMH K-57523 | Desmosomatidae | WS | GC | 24 | X | |
| DesmJP023 | DSB_1784 | Eugerdella | sp.11 | D39 | BOLD:ADW6372 | 50849 | ZMH K-57524 | Desmosomatidae | WS | GC | 24 | X | X |
| DesmJP024 | DSB_1785 | Eugerdella | sp.11 | D39 | BOLD:ADW6372 | 50850 | ZMH K-57525 | Desmosomatidae | WS | GC | 24 | X | X |
| DesmJP025 | DSB_1786 | Mirabilicoxa | sp.F | D13 | BOLD:ACY9838 | 50851 | ZMH K-57526 | Desmosomatidae | WS | GC | 24 | X | X |
| DesmJP026 | DSB_1787 | Eugerdella | sp. 3 | D17 | BOLD:ADL5370 | 50852 | ZMH K-57527 | Desmosomatidae | WS | GC | 50 | X | |
| DesmJP027 | DSB_1788 | Mirabilicoxa | sp.1 | D20 | BOLD:ADL5037 | 50853 | ZMH K-57528 | Desmosomatidae | WS | GC | 50 | X | X |
| DesmJP028 | DSB_1789 | Prochelator | indet | D30 | BOLD:ADF9936 | 50854 | ZMH K-57529 | Desmosomatidae | WS | GC | 50 | X | |
| DesmJP030 | DSB_1791 | Eugerdella | sp. 3 | D16 | BOLD:ADM0377 | 50856 | ZMH K-57530 | Desmosomatidae | WS | GC | 50 | X | |
| DesmJP031 | DSB_1792 | Eugerdella | indet | D49 | BOLD:ADL2572 | 50857 | ZMH K-57531 | Desmosomatidae | WS | GC | 50 | X | |
| DesmJP032 | DSB_1793 | Mirabilicoxa | sp.F | D13 | BOLD:ACY9838 | 50858 | ZMH K-57532 | Desmosomatidae | WS | GC | 50 | X | X |
| DesmJP033 | DSB_1794 | Mirabilicoxa | sp.D | D11 | BOLD:ADL5366 | 50859 | ZMH K-57533 | Desmosomatidae | WS | GC | 50 | X | X |
| DesmJP034 | DSB_1795 | Eugerdella | sp. 3 | D17 | BOLD:ADL5370 | 50860 | ZMH K-57534 | Desmosomatidae | WS | GC | 50 | X | |
| DesmJP037 | DSB_1798 | Eugerdella | cf. egoni | D49 | BOLD:ADL5536 | 50863 | ZMH K-57535 | Desmosomatidae | WS | GC | 50 | X | |
| DesmJP038 | DSB_1799 | Mirabilicoxa | sp.C | D09 | BOLD:ADL5035 | 50864 | ZMH K-57536 | Desmosomatidae | WS | GC | 59 | X | X |
| DesmJP039 | DSB_1800 | Eugerdella | sp.11 | D39 | BOLD:ADW6372 | 50865 | ZMH K-57537 | Desmosomatidae | WS | GC | 59 | X | X |
| DesmJP040 | DSB_1801 | Prochelator | indet | D30 | BOLD:ADF9936 | 50866 | ZMH K-57538 | Desmosomatidae | WS | GC | 59 | X | |
| DesmJP041 | DSB_1802 | Mirabilicoxa | sp.1 | D20 | BOLD:ADL5037 | 50867 | ZMH K-57539 | Desmosomatidae | WS | GC | 59 | X | X |
| DesmJP042 | DSB_1803 | Eugerdella | sp.10 | D38 | BOLD:ADL5031 | 50868 | ZMH K-57540 | Desmosomatidae | WS | GC | 59 | X | X |
| DesmJP043 | DSB_1804 | Eugerdella | sp.10 | D38 | BOLD:ADL5031 | 50869 | ZMH K-57541 | Desmosomatidae | WS | GC | 59 | X | |
| DesmJP045 | DSB_1806 | Eugerdella | sp. 4 | D17 | BOLD:ACY6387 | 50871 | ZMH K-57542 | Desmosomatidae | WS | GC | 59 | X | |
| DesmJP046 | DSB_1807 | Eugerdella | sp.10 | D38 | BOLD:ADL5031 | 50872 | ZMH K-57543 | Desmosomatidae | WS | GC | 59 | X | X |
| DesmJP047 | DSB_1808 | Eugerdella | sp. 3 | D17 | BOLD:ADL5370 | 50873 | ZMH K-57544 | Desmosomatidae | WS | GC | 59 | X | |
| DesmJP048 | DSB_1809 | cf. Mirabilicoxa | sp.C | D09 | BOLD:ADL5035 | 50874 | ZMH K-57545 | Desmosomatidae | WS | GC | 59 | X | X |
| DesmJP049 | DSB_1810 | Prochelator | indet | D30 | BOLD:ADF9936 | 50875 | ZMH K-57546 | Desmosomatidae | WS | GC | 59 | X | |
| DesmJP050 | DSB_1811 | Chelator | sp.D | D07 | BOLD:ADL5034 | 50876 | ZMH K-57547 | Desmosomatidae | WS | IOM | 81 | X | X |
| DesmJP051 | DSB_1812 | cf. Eugerdella | indet | D36 | BOLD:ADG1796 | 50877 | ZMH K-57548 | Desmosomatidae | WS | IOM | 81 | X | |



| | | | | | | | | | | | | | |
|---|---|---|---|---|---|---|---|---|---|---|---|---|---|
| DesmJP053 | DSB_1814 | Eugerdella | sp.5 | D44 | BOLD:ADL5216 | 50879 | ZMH K-57549 | Desmosomatidae | WS | IOM | 81 | X | X |
| DesmJP054 | DSB_1815 | Mirabilicoxa | sp.C | D21 | BOLD:ADL2890 | 50880 | ZMH K-57550 | Desmosomatidae | WS | IOM | 81 | X | X |
| DesmJP055 | DSB_1816 | cf. Eugerdella | indet | D36 | BOLD:ADG1796 | 50881 | ZMH K-57551 | Desmosomatidae | WS | IOM | 81 | X | |
| DesmJP057 | DSB_1818 | Eugerdella | 5 or 6 | D36 | BOLD:ADG1796 | 50883 | ZMH K-57552 | Desmosomatidae | WS | IOM | 81 | X | |
| DesmJP059 | DSB_1820 | Eugerdella | sp.11 | D39 | BOLD:ADW6372 | 50885 | ZMH K-57553 | Desmosomatidae | WS | IOM | 81 | X | X |
| DesmJP060 | DSB_1821 | Chelator | sp.D | D07 | BOLD:ADL5034 | 50886 | ZMH K-57554 | Desmosomatidae | WS | IOM | 81 | X | X |
| DesmJP061 | DSB_1822 | Chelator | sp.D | D07 | BOLD:ADL5034 | 50887 | ZMH K-57555 | Desmosomatidae | WS | IOM | 81 | X | X |
| DesmJP064 | DSB_1825 | cf. Disparella | indet | D25 | BOLD:ADL5032 | 50890 | ZMH K-57556 | Desmosomatidae | WS | IOM | 99 | X | X |
| DesmJP065 | DSB_1826 | Eugerdella | sp. 5 | D36 | BOLD:ADG1796 | 50891 | ZMH K-57557 | Desmosomatidae | WS | IOM | 99 | X | |
| DesmJP066 | DSB_1827 | Mirabilicoxa | sp.D | D11 | BOLD:ADL5366 | 50892 | ZMH K-57558 | Desmosomatidae | WS | IOM | 99 | X | X |
| DesmJP067 | DSB_1828 | Mirabilicoxa | indet | D13 | BOLD:ACY9838 | 50893 | ZMH K-57559 | Desmosomatidae | WS | IOM | 99 | X | |
| DesmJP068 | DSB_1829 | Eugerdella | sp. 6 | D36 | BOLD:ADG1796 | 50894 | ZMH K-57560 | Desmosomatidae | WS | IOM | 99 | X | |
| DesmJP069 | DSB_1830 | Eugerdella | sp.11 | D39 | BOLD:ADW6372 | 50895 | ZMH K-57561 | Desmosomatidae | WS | IOM | 99 | X | X |
| DesmJP071 | DSB_1832 | Eugerdella | sp. 3 | D42 | BOLD:ADM1411 | 50897 | ZMH K-57562 | Desmosomatidae | WS | IOM | 99 | X | |
| DesmJP073 | DSB_1834 | Mirabilicoxa | sp.A | D18 | BOLD:ACY5985 | 50899 | ZMH K-57563 | Desmosomatidae | WS | IOM | 99 | X | X |
| DesmJP074 | DSB_1835 | Eugerdella | indet | D49 | BOLD:ADL2572 | 50900 | ZMH K-57564 | Desmosomatidae | WS | BC | 118 | X | |
| DesmJP075 | DSB_1836 | cf. Disparella | indet | D27 | BOLD:ADL5535 | 50901 | ZMH K-57565 | Desmosomatidae | WS | BC | 118 | X | X |
| DesmJP076 | DSB_1837 | Eugerdella | sp. cf.7 | D45 | BOLD:ADL2895 | 50902 | ZMH K-57566 | Desmosomatidae | WS | BC | 118 | X | X |
| DesmJP077 | DSB_1838 | Disparella | indet | D25 | BOLD:ADL5032 | 50903 | ZMH K-57567 | Desmosomatidae | WS | BC | 118 | X | X |
| DesmJP078 | DSB_1839 | cf. Prochelator | sp.1 | D32 | BOLD:ADL3300 | 50904 | ZMH K-57568 | Desmosomatidae | WS | BC | 118 | X | |
| DesmJP081 | DSB_1842 | Eugerdella | sp.10 | D38 | BOLD:ADL5031 | 50907 | ZMH K-57569 | Desmosomatidae | WS | BC | 118 | X | X |
| DesmJP082 | DSB_1843 | Prochelator | indet | D30 | BOLD:ADF9936 | 50908 | ZMH K-57570 | Desmosomatidae | WS | BC | 118 | X | |
| DesmJP083 | DSB_1844 | Mirabilicoxa | sp.D | D11 | BOLD:ADL5366 | 50909 | ZMH K-57571 | Desmosomatidae | WS | BC | 118 | X | |
| DesmJP084 | DSB_1845 | Eugerdella | sp. 5 | D43 | BOLD:ADL5372 | 50910 | ZMH K-57572 | Desmosomatidae | WS | BC | 118 | X | X |
| DesmJP085 | DSB_1846 | Prochelator | indet | D30 | BOLD:ADF9936 | 50911 | ZMH K-57573 | Desmosomatidae | WS | BC | 118 | X | |
| DesmJP086 | DSB_1847 | Prochelator | sp.1 | D32 | BOLD:ADL3300 | 50912 | ZMH K-57574 | Desmosomatidae | WS | BC | 118 | X | |
| DesmJP087 | DSB_1848 | Prochelator | sp.1 | D32 | BOLD:ADL3300 | 50913 | ZMH K-57575 | Desmosomatidae | WS | BC | 133 | X | |
| DesmJP088 | DSB_1849 | Prochelator | sp.1 | D32 | BOLD:ADL3300 | 50914 | ZMH K-57576 | Desmosomatidae | WS | BC | 133 | X | |
| DesmJP090 | DSB_1851 | Mirabilicoxa | indet | D13 | BOLD:ACY9838 | 50916 | ZMH K-57577 | Desmosomatidae | WS | BC | 133 | X | |
| DesmJP091 | DSB_1852 | Chelator | sp.D | D07 | BOLD:ADL5034 | 50917 | ZMH K-57578 | Desmosomatidae | WS | BC | 133 | X | X |
| DesmJP092 | DSB_1853 | Eugerdella | cf. egoni | D48 | BOLD:ADL5371 | 50918 | ZMH K-57579 | Desmosomatidae | WS | BC | 133 | X | |
| DesmJP093 | DSB_1854 | cf. Disparella | sp.1 | D24 | BOLD:ADL4887 | 50919 | ZMH K-57580 | Desmosomatidae | WS | BC | 133 | X | X |
| DesmJP094 | DSB_1855 | Eugerdella | sp. 5 | D44 | BOLD:ADL5216 | 50920 | ZMH K-57581 | Desmosomatidae | WS | BC | 133 | X | X |
| DesmJP095 | DSB_1856 | Eugerdella | sp.5 | D44 | BOLD:ADL5216 | 50921 | ZMH K-57582 | Desmosomatidae | WS | BC | 133 | X | X |
| DesmJP096 | DSB_1857 | Prochelator | indet | D30 | BOLD:ADF9936 | 50922 | ZMH K-57583 | Desmosomatidae | WS | BC | 133 | X | |
| DesmJP099 | DSB_1860 | Mirabilicoxa | sp.D | D11 | BOLD:ADL5215 | 50925 | ZMH K-57584 | Desmosomatidae | WS | FC | 158 | X | X |
| DesmJP100 | DSB_1861 | Mirabilicoxa | sp.E | D12 | BOLD:ADL5213 | 50926 | ZMH K-57585 | Desmosomatidae | WS | FC | 158 | X | X |
| DesmJP101 | DSB_1862 | Eugerdella | sp. 8 | D40 | BOLD:ADL2751 | 50927 | ZMH K-57586 | Desmosomatidae | WS | FC | 158 | X | X |
| DesmJP103 | DSB_1864 | Disparella | indet | D26 | BOLD:ADL2577 | 50929 | ZMH K-57587 | Desmosomatidae | WS | FC | 158 | X | X |
| DesmJP104 | DSB_1865 | Mirabilicoxa | indet | D37 | BOLD:ADL5211 | 50930 | ZMH K-57588 | Desmosomatidae | WS | FC | 158 | X | |
| DesmJP105 | DSB_1866 | Mirabilicoxa | sp.A | D18 | BOLD:ACY5985 | 50931 | ZMH K-57589 | Desmosomatidae | WS | FC | 158 | X | X |
| DesmJP106 | DSB_1867 | Mirabilicoxa | sp.E | D12 | BOLD:ADL5213 | 50932 | ZMH K-57590 | Desmosomatidae | WS | FC | 158 | X | X |
| DesmJP107 | DSB_1868 | cf. Mirabilicoxa | sp.B | D10 | BOLD:ADL4888 | 50933 | ZMH K-57591 | Desmosomatidae | WS | FC | 158 | X | X |
| DesmJP108 | DSB_1869 | cf. Mirabilicoxa | sp.E | D12 | BOLD:ADL5213 | 50934 | ZMH K-57592 | Desmosomatidae | WS | FC | 158 | X | X |
| DesmJP109 | DSB_1870 | Eugerdella | sp. 6 | D34 | BOLD:ADL5033 | 50935 | ZMH K-57593 | Desmosomatidae | WS | FC | 158 | X | |
| DesmJP110 | DSB_1871 | Eugerdella | sp. 7 | D46 | BOLD:ADL2578 | 50936 | ZMH K-57594 | Desmosomatidae | WS | FC | 158 | X | X |
| DesmJP111 | DSB_1872 | Mirabilicoxa | sp.A | D18 | BOLD:ACY5985 | 50937 | ZMH K-57595 | Desmosomatidae | WS | FC | 171 | X | X |
| DesmJP112 | DSB_1873 | Eugerdella | sp.5 | D44 | BOLD:ADL5216 | 50938 | ZMH K-57596 | Desmosomatidae | WS | FC | 171 | X | |
| DesmJP113 | DSB_1874 | Prochelator | indet | D32 | BOLD:ADL3300 | 50939 | ZMH K-57597 | Desmosomatidae | WS | FC | 171 | X | |
| DesmJP114 | DSB_1875 | cf. Mirabilicoxa | sp.E | D12 | BOLD:ADL5213 | 50940 | ZMH K-57598 | Desmosomatidae | WS | FC | 171 | X | X |
| DesmJP115 | DSB_1876 | cf. Mirabilicoxa | sp.A | D18 | BOLD:ACY5985 | 50941 | ZMH K-57599 | Desmosomatidae | WS | FC | 171 | X | X |
| DesmJP116 | DSB_1877 | cf. Mirabilicoxa | indet | D18 | BOLD:ACY5985 | 50942 | ZMH K-57600 | Desmosomatidae | WS | FC | 171 | X | |
| DesmJP117 | DSB_1878 | Eugerdella | sp.5 | D44 | BOLD:ADL5216 | 50943 | ZMH K-57601 | Desmosomatidae | WS | FC | 171 | X | |
| DesmJP118 | DSB_1879 | Mirabilicoxa | sp.A | D18 | BOLD:ACY5985 | 50944 | ZMH K-57602 | Desmosomatidae | WS | FC | 171 | X | X |
| DesmJP119 | DSB_1880 | Chelator | sp.B | D04 | BOLD:ADL5212 | 50945 | ZMH K-57603 | Desmosomatidae | WS | FC | 171 | X | X |
| DesmJP120 | DSB_1881 | Eugerdella | sp.5 | D44 | BOLD:ADL5216 | 50946 | ZMH K-57604 | Desmosomatidae | WS | FC | 171 | X | |
| DesmJP121 | DSB_1882 | Prochelator | indet | D29 | BOLD:ADL2573 | 50947 | ZMH K-57605 | Desmosomatidae | WS | FC | 171 | X | |
| DesmJP123 | DSB_1884 | Prochelator | sp.1 | D32 | BOLD:ADL3300 | 50949 | ZMH K-57606 | Desmosomatidae | WS | FC | 171 | X | |
| DesmJP124 | DSB_1885 | Mirabilicoxa | indet | D14 | BOLD:ADL5365 | 50950 | ZMH K-57607 | Desmosomatidae | WS | A3 | 192 | X | |
| DesmJP125 | DSB_1886 | Disparella | indet | D22 | BOLD:ADL2753 | 50951 | ZMH K-57608 | Desmosomatidae | WS | A3 | 192 | X | X |
| DesmJP126 | DSB_1887 | Eugerdella | sp. 6 ? | D35 | BOLD:ADL5217 | 50952 | ZMH K-57609 | Desmosomatidae | WS | A3 | 192 | X | |
| DesmJP127 | DSB_1888 | Prochelator | sp. 123 | D31 | BOLD:ADL2579 | 50953 | ZMH K-57610 | Desmosomatidae | WS | A3 | 192 | X | |
| DesmJP128 | DSB_1889 | cf. Mirabilicoxa | sp.B | D10 | BOLD:ADL4888 | 50954 | ZMH K-57611 | Desmosomatidae | WS | A3 | 192 | X | X |
| DesmJP129 | DSB_1890 | Mirabilicoxa | indet | D14 | BOLD:ADL5365 | 50955 | ZMH K-57612 | Desmosomatidae | WS | A3 | 192 | X | |



| | | | | | | | | | | | | | |
|---|---|---|---|---|---|---|---|---|---|---|---|---|---|
| DesmJP131 | DSB_1892 | *Mirabilicoxa* | sp.B | D10 | BOLD:ADL4888 | 50957 | ZMH K-57613 | Desmosomatidae | WS | A3 | 192 | X | X |
| DesmJP133 | DSB_1894 | *Chelator* | sp.A | D01 | BOLD:ADL5030 | 50959 | ZMH K-57614 | Desmosomatidae | WS | A3 | 192 | X | X |
| DesmJP134 | DSB_1895 | cf. *Mirabilicoxa* | sp.B | D10 | BOLD:ADL4888 | 50960 | ZMH K-57615 | Desmosomatidae | WS | A3 | 192 | X | X |
| DesmJP135 | DSB_1896 | *Oecidiobranchus* | oeci | D08 | BOLD:ADL5367 | 50961 | ZMH K-57616 | Desmosomatidae | WS | A3 | 192 | X | X |
| DesmJP137 | DSB_1898 | *Oecidiobranchus* | oeci | D08 | BOLD:ADL5367 | 50963 | ZMH K-57617 | Desmosomatidae | WS | A3 | 197 | X | X |
| DesmJP138 | DSB_1899 | *Chelator* | sp.A | D01 | BOLD:ADL5030 | 50964 | ZMH K-57618 | Desmosomatidae | WS | A3 | 197 | X | X |
| DesmJP139 | DSB_1900 | *Chelator* | sp.A | D01 | | 50965 | ZMH K-57619 | Desmosomatidae | WS | A3 | 197 | | |
| DesmJP140 | DSB_1901 | cf. *Disparella* | indet | D28 | BOLD:ADL3347 | 50966 | ZMH K-57620 | Desmosomatidae | WS | A3 | 197 | X | X |
| DesmJP142 | DSB_1903 | *Eugerdella* | sp. 6 ? | D34 | BOLD:ADL5033 | 50968 | ZMH K-57621 | Desmosomatidae | WS | A3 | 197 | X | |
| DesmJP143 | DSB_1904 | cf. *Disparella* | indet | D28 | BOLD:ADL3347 | 50969 | ZMH K-57622 | Desmosomatidae | WS | A3 | 197 | X | |
| DesmJP145 | DSB_1906 | cf. *Chelator* | sp.A | D01 | BOLD:ADL5030 | 50971 | ZMH K-57623 | Desmosomatidae | WS | A3 | 197 | X | X |
| DesmJP147 | DSB_1908 | *Eugerdella* | sp. 6 ? | D35 | BOLD:ADL5217 | 50973 | ZMH K-57624 | Desmosomatidae | WS | A3 | 197 | X | |
| DesmJP148 | DSB_1909 | cf. *Disparella* | indet | D28 | BOLD:ADL3347 | 50974 | ZMH K-57625 | Desmosomatidae | WS | A3 | 210 | X | X |
| DesmJP152 | DSB_1913 | *Chelator* | sp.C | D05 | BOLD:ADL2581 | 50978 | ZMH K-57626 | Desmosomatidae | WS | A3 | 210 | X | X |
| DesmJP153 | DSB_1914 | *Chelator* | sp.C | D05 | BOLD:ADL2581 | 50979 | ZMH K-57627 | Desmosomatidae | WS | A3 | 210 | X | X |
| DesmJP154 | DSB_1915 | *Oecidiobranchus* | oeci | D08 | BOLD:ADL5367 | 50980 | ZMH K-57628 | Desmosomatidae | WS | A3 | 210 | X | X |
| DesmJP159 | DSB_1920 | cf. *Chelator* | sp.C | D05 | BOLD:ADL2581 | 50985 | ZMH K-57629 | Desmosomatidae | WS | A3 | 210 | X | X |
| DesmJP160 | DSB_1921 | *Mirabilicoxa* | indet | D14 | BOLD:ADL5365 | 50986 | ZMH K-57630 | Desmosomatidae | WS | A3 | 210 | X | X |
| DesmJP162 | DSB_1949 | *Eugerdella* | sp. 6 ? | D35 | BOLD:ADL5217 | 51014 | ZMH K-57631 | Desmosomatidae | WS | A3 | 197 | X | |
| HaplDi02 | DSB_1320 | *Chauliodobiscus* | sp. 01 | H15 | BOLD:ADL2819 | 50385 | ZMH K-57632 | Haploniscidae | W | DISCOL | 93 | X | X |
| HaplDi03 | DSB_1321 | *Chauliodoniscus* | sp. 01 | H15 | BOLD:ADL2819 | 50386 | ZMH K-57633 | Haploniscidae | W | DISCOL | 93 | X | X |
| HaplDi04 | DSB_1322 | *Chauliodoniscus* | sp. 01 | H15 | BOLD:ADL2819 | 50387 | ZMH K-57634 | Haploniscidae | W | DISCOL | 93 | X | X |
| HaplDi05 | DSB_1323 | cf. *Chauliodoniscus* | indet | H07 | BOLD:ADL3283 | 50388 | ZMH K-57635 | Haploniscidae | W | DISCOL | 93 | X | |
| HaplDi07 | DSB_1325 | *Chauliodoniscus* | sp. 01 | H14 | BOLD:ADL2983 | 50390 | ZMH K-57636 | Haploniscidae | W | DISCOL | 93 | X | X |
| HaplDi08 | DSB_1326 | *Chauliodoniscus* | sp. 01 | H14 | BOLD:ADL2983 | 50391 | ZMH K-57637 | Haploniscidae | W | DISCOL | 93 | X | X |
| HaplDi09 | DSB_1327 | *Chauliodoniscus* | sp. 01 | H23 | BOLD:ADL3281 | 50392 | ZMH K-57638 | Haploniscidae | W | DISCOL | 93 | X | |
| HaplDi11 | DSB_1329 | *Chauliodoniscus* | sp. 01 | H15 | BOLD:ADL2819 | 50394 | ZMH K-57639 | Haploniscidae | W | DISCOL | 93 | X | X |
| HaplDi12 | DSB_1330 | cf. *Chauliodoniscus* | indet | H07 | BOLD:ADL3283 | 50395 | ZMH K-57640 | Haploniscidae | W | DISCOL | 93 | X | |
| HaplDi13 | DSB_1331 | *Chauliodoniscus* | sp. 01 | H15 | BOLD:ADL2819 | 50396 | ZMH K-57641 | Haploniscidae | W | DISCOL | 93 | X | X |
| HaplDi14 | DSB_1332 | cf. *Chauliodoniscus* | indet | H07 | BOLD:ADL3283 | 50397 | ZMH K-57642 | Haploniscidae | W | DISCOL | 93 | X | |
| HaplDi15 | DSB_1333 | *Chauliodoniscus* | sp. 01 | H23 | BOLD:ADL2981 | 50398 | ZMH K-57643 | Haploniscidae | W | DISCOL | 93 | X | |
| HaplDi16 | DSB_1334 | cf. *Chauliodoniscus* | indet | H07 | BOLD:ADL3283 | 50399 | ZMH K-57644 | Haploniscidae | W | DISCOL | 93 | X | |
| HaplDi17 | DSB_1335 | *Chauliodoniscus* | sp. 01 | H14 | BOLD:ADL2983 | 50400 | ZMH K-57645 | Haploniscidae | W | DISCOL | 93 | X | X |
| HaplDi18 | DSB_1336 | *Chauliodoniscus* | sp. 01 | H14 | BOLD:ADL2983 | 50401 | ZMH K-57646 | Haploniscidae | W | DISCOL | 93 | X | X |
| HaplDi19 | DSB_1337 | cf. *Chauliodoniscus* | indet | H07 | BOLD:ADL3283 | 50402 | ZMH K-57647 | Haploniscidae | W | DISCOL | 85 | X | |
| HaplDi24 | DSB_1342 | *Chauliodoniscus* | sp. 01 | H09 | BOLD:ADL6396 | 50407 | ZMH K-57648 | Haploniscidae | W | DISCOL | 117 | X | |
| HaplDi25 | DSB_1343 | *Chauliodoniscus* | sp. 01 | H14 | BOLD:ADL2983 | 50408 | ZMH K-57649 | Haploniscidae | W | DISCOL | 117 | X | X |
| HaplJP001 | DSB_2044 | *Mastigoniscus* | sp. # 3 | H17 | BOLD:ADL3127 | 51109 | ZMH K-57650 | Haploniscidae | W | GC | 20 | X | X |
| HaplJP002 | DSB_2045 | *Mastigoniscus* | sp. # 3 | H17 | BOLD:ADL3127 | 51110 | ZMH K-57651 | Haploniscidae | W | GC | 20 | X | X |
| HaplJP003 | DSB_2046 | *Haploniscus* | sp. # 2 | H12 | BOLD:ADL3122 | 51111 | ZMH K-57652 | Haploniscidae | W | GC | 20 | X | X |
| HaplJP004 | DSB_2047 | *Mastigoniscus* | sp. # 3 | H17 | BOLD:ADL3127 | 51112 | ZMH K-57653 | Haploniscidae | W | GC | 20 | X | X |
| HaplJP005 | DSB_2048 | *Mastigoniscus* | sp. # 3 | H17 | BOLD:ADL3127 | 51113 | ZMH K-57654 | Haploniscidae | W | GC | 20 | X | X |
| HaplJP006 | DSB_2049 | *Haploniscus* | sp. # 2 | H12 | BOLD:ADL3122 | 51114 | ZMH K-57655 | Haploniscidae | W | GC | 24 | X | |
| HaplJP008 | DSB_2051 | *Haploniscus* | sp. # 4 | H06 | BOLD:ADL6761 | 51116 | ZMH K-57656 | Haploniscidae | W | GC | 24 | X | |
| HaplJP009 | DSB_2052 | *Mastigoniscus* | sp. # 3 | H17 | BOLD:ADL3127 | 51117 | ZMH K-57657 | Haploniscidae | W | GC | 24 | X | X |
| HaplJP010 | DSB_2053 | *Haploniscus* | sp. # 2 | H10 | BOLD:ADL3282 | 51118 | ZMH K-57658 | Haploniscidae | W | GC | 50 | X | |
| HaplJP011 | DSB_2054 | *Mastigoniscus* | sp. # 5 | H16 | BOLD:ADL2988 | 51119 | ZMH K-57659 | Haploniscidae | W | GC | 50 | X | X |
| HaplJP013 | DSB_2056 | *Haploniscus* | sp. # 2 | H10 | BOLD:ADL4714 | 51121 | ZMH K-57660 | Haploniscidae | W | GC | 50 | X | |
| HaplJP014 | DSB_2057 | *Haploniscus* | sp. # 2 | H10 | BOLD:ADL3282 | 51122 | ZMH K-57661 | Haploniscidae | W | GC | 50 | X | |
| HaplJP015 | DSB_2058 | *Haploniscus* | sp. # 2 | H10 | BOLD:ADL4714 | 51123 | ZMH K-57662 | Haploniscidae | W | GC | 50 | X | X |
| HaplJP016 | DSB_2059 | *Haploniscus* | sp. # 2 | H10 | BOLD:ADL4714 | 51124 | ZMH K-57663 | Haploniscidae | W | GC | 50 | X | X |
| HaplJP017 | DSB_2060 | *Mastigoniscus* | sp. # 5 | H16 | BOLD:ADL2988 | 51125 | ZMH K-57664 | Haploniscidae | W | GC | 59 | X | X |
| HaplJP018 | DSB_2061 | *Mastigoniscus* | sp. # 5 | H16 | BOLD:ADL2988 | 51126 | ZMH K-57665 | Haploniscidae | W | IOM | 81 | X | X |
| HaplJP019 | DSB_2062 | *Haploniscus* | sp. # 2 | H10 | BOLD:ADL3123 | 51127 | ZMH K-57666 | Haploniscidae | W | IOM | 81 | X | |
| HaplJP020 | DSB_2063 | *Mastigoniscus* | sp. # 5 | H08 | BOLD:ADL2984 | 51128 | ZMH K-57667 | Haploniscidae | W | IOM | 81 | X | |
| HaplJP021 | DSB_2064 | *Mastigoniscus* | sp. # 3 | H22 | BOLD:ADL6395 | 51129 | ZMH K-57668 | Haploniscidae | W | IOM | 81 | X | |
| HaplJP022 | DSB_2065 | *Haploniscus* | sp. # 2 | H10 | BOLD:ADL4710 | 51130 | ZMH K-57669 | Haploniscidae | W | IOM | 99 | X | X |
| HaplJP023 | DSB_2066 | *Mastigoniscus* | sp. # 3 | H22 | BOLD:ADL2982 | 51131 | ZMH K-57670 | Haploniscidae | W | BC | 118 | X | |
| HaplJP024 | DSB_2067 | *Mastigoniscus* | sp. # 3007 | H19 | BOLD:ADL2985 | 51132 | ZMH K-57671 | Haploniscidae | W | BC | 118 | | X |
| HaplJP025 | DSB_2068 | *Mastigoniscus* | sp. # 3 | H22 | BOLD:ADL2982 | 51133 | ZMH K-57672 | Haploniscidae | W | BC | 118 | X | |
| HaplJP026 | DSB_2069 | *Mastigoniscus* | sp. # 3 | H22 | BOLD:ADL2982 | 51134 | ZMH K-57673 | Haploniscidae | W | BC | 118 | X | |
| HaplJP027 | DSB_2070 | *Mastigoniscus* | sp. # 3007 | H19 | BOLD:ADL2985 | 51135 | ZMH K-57674 | Haploniscidae | W | BC | 118 | X | X |
| HaplJP028 | DSB_2071 | *Mastigoniscus* | sp. # 3 | H22 | BOLD:ADL2982 | 51136 | ZMH K-57675 | Haploniscidae | W | BC | 118 | X | |
| HaplJP029 | DSB_2072 | *Mastigoniscus* | sp. # 3007 | H19 | BOLD:ADL2985 | 51137 | ZMH K-57676 | Haploniscidae | W | BC | 118 | X | X |



| | | | | | | | | | | | | | |
|---|---|---|---|---|---|---|---|---|---|---|---|---|---|
| HaplJP031 | DSB_2074 | *Mastigoniscus* | sp. # 3 | H22 | BOLD:ADL2982 | 51139 | ZMH K-57677 | Haploniscidae | W | BC | 118 | X | |
| HaplJP032 | DSB_2075 | *Haploniscus* | sp. # 2 | H10 | BOLD:ADL3968 | 51140 | ZMH K-57678 | Haploniscidae | W | BC | 118 | X | |
| HaplJP033 | DSB_2076 | *Haploniscus* | sp. # 2 | H10 | BOLD:ADL3968 | 51141 | ZMH K-57679 | Haploniscidae | W | BC | 118 | X | X |
| HaplJP034 | DSB_2077 | *Mastigoniscus* | sp. # 3 | H22 | BOLD:ADL2982 | 51142 | ZMH K-57680 | Haploniscidae | W | BC | 118 | X | |
| HaplJP035 | DSB_2078 | *Mastigoniscus* | sp. # 3 | H22 | BOLD:ADL2982 | 51143 | ZMH K-57681 | Haploniscidae | W | BC | 133 | X | |
| HaplJP036 | DSB_2079 | *Mastigoniscus* | sp. # 3 | H22 | BOLD:ADL2982 | 51144 | ZMH K-57682 | Haploniscidae | W | BC | 133 | X | |
| HaplJP037 | DSB_2080 | *Mastigoniscus* | sp. # 3 | H22 | BOLD:ADL2982 | 51145 | ZMH K-57683 | Haploniscidae | W | BC | 133 | X | |
| HaplJP038 | DSB_2081 | *Mastigoniscus* | sp. # 3007 | H19 | BOLD:ADL2985 | 51146 | ZMH K-57684 | Haploniscidae | W | BC | 133 | X | X |
| HaplJP039 | DSB_2082 | *Mastigoniscus* | sp. # 3007 | H19 | BOLD:ADL2985 | 51147 | ZMH K-57685 | Haploniscidae | W | BC | 133 | X | X |
| HaplJP040 | DSB_2083 | *Mastigoniscus* | sp. | H22 | BOLD:ADL2982 | 51148 | ZMH K-57686 | Haploniscidae | W | BC | 133 | X | X |
| HaplJP041 | DSB_2084 | *Mastigoniscus* | sp. # 3 | H21 | BOLD:ADL3128 | 51149 | ZMH K-57687 | Haploniscidae | W | FC | 158 | X | X |
| HaplJP042 | DSB_2085 | *Mastigoniscus* | sp. # 3007 | H18 | BOLD:ADL6560 | 51150 | ZMH K-57688 | Haploniscidae | W | FC | 158 | X | X |
| HaplJP043 | DSB_2086 | *Haploniscus* | sp. # 2 | H10 | BOLD:ADL3123 | 51151 | ZMH K-57689 | Haploniscidae | W | FC | 158 | X | |
| HaplJP044 | DSB_2087 | *Haploniscus* | sp. # 1a | H05 | BOLD:ADL6397 | 51152 | ZMH K-57690 | Haploniscidae | W | FC | 158 | X | X |
| HaplJP045 | DSB_2088 | *Haploniscus* | sp. # 1 | H04 | BOLD:ADL3129 | 51153 | ZMH K-57691 | Haploniscidae | W | FC | 158 | X | |
| HaplJP046 | DSB_2089 | *Haploniscus* | sp. # 1 | H04 | BOLD:ADL3129 | 51154 | ZMH K-57692 | Haploniscidae | W | FC | 158 | X | |
| HaplJP047 | DSB_2090 | *Mastigoniscus* | sp. # 7 | H13 | BOLD:ADL3125 | 51155 | ZMH K-57693 | Haploniscidae | W | FC | 171 | X | X |
| HaplJP048 | DSB_2091 | *Mastigoniscus* | sp. # 3 | H21 | BOLD:ADL3128 | 51156 | ZMH K-57694 | Haploniscidae | W | FC | 171 | X | |
| HaplJP049 | DSB_2092 | *Haploniscus* | sp. # 2 | H10 | BOLD:ADL2986 | 51157 | ZMH K-57695 | Haploniscidae | W | FC | 171 | X | X |
| HaplJP050 | DSB_2093 | *Haploniscus* | sp. # 2 | H10 | BOLD:ADL3123 | 51158 | ZMH K-57696 | Haploniscidae | W | FC | 171 | X | X |
| HaplJP051 | DSB_2094 | *Mastigoniscus* | sp. # 3 | H21 | BOLD:ADL3128 | 51159 | ZMH K-57697 | Haploniscidae | W | FC | 171 | X | |
| HaplJP052 | DSB_2095 | *Haploniscus* | sp. # 1 | H03 | BOLD:ADL2989 | 51160 | ZMH K-57698 | Haploniscidae | W | FC | 171 | X | X |
| HaplJP053 | DSB_2096 | *Mastigoniscus* | sp. # 7 | H13 | BOLD:ADL3125 | 51161 | ZMH K-57699 | Haploniscidae | W | FC | 171 | X | X |
| HaplJP054 | DSB_2097 | *Haploniscus* | sp. # 2 | H10 | BOLD:ADL2986 | 51162 | ZMH K-57700 | Haploniscidae | W | FC | 171 | X | |
| HaplJP055 | DSB_2098 | *Haploniscus* | sp. # 1 | H03 | BOLD:ADL2989 | 51163 | ZMH K-57701 | Haploniscidae | W | FC | 171 | X | X |
| HaplJP057 | DSB_2100 | *Mastigoniscus* | sp. # 8 | H02 | BOLD:ADL3126 | 51165 | ZMH K-57702 | Haploniscidae | W | A3 | 210 | X | X |
| HaplJP058 | DSB_2101 | *Haploniscus* | sp. # 2 | H11 | BOLD:ADL2820 | 51166 | ZMH K-57703 | Haploniscidae | W | A3 | 210 | X | X |
| HaplJP059 | DSB_2102 | *Mastigoniscus* | sp. # 5 | H08 | BOLD:ADL2984 | 51167 | ZMH K-57704 | Haploniscidae | W | A3 | 210 | X | |
| HaplJP060 | DSB_2103 | *Haploniscus* | sp. # 2 | H11 | BOLD:ADL3124 | 51168 | ZMH K-57705 | Haploniscidae | W | A3 | 210 | X | |
| HaplJP061 | DSB_2104 | *Mastigoniscus* | sp. # 5 | H08 | BOLD:ADL2984 | 51169 | ZMH K-57706 | Haploniscidae | W | A3 | 210 | X | |
| HaplJP062 | DSB_2105 | *Haploniscus* | sp. # 2 | H11 | BOLD:ADL3124 | 51170 | ZMH K-57707 | Haploniscidae | W | A3 | 210 | X | |
| HaplJP063 | DSB_2106 | *Mastigoniscus* | sp. # 3 | H01 | BOLD:ADL3280 | 51171 | ZMH K-57708 | Haploniscidae | W | A3 | 210 | X | X |
| HaplJP064 | DSB_2107 | *Mastigoniscus* | sp. # 5 | H08 | BOLD:ADL2984 | 51172 | ZMH K-57709 | Haploniscidae | W | A3 | 210 | X | |
| HaplJP065 | DSB_2108 | *Haploniscus* | sp. # 2 | H11 | BOLD:ADL3124 | 51173 | ZMH K-57710 | Haploniscidae | W | A3 | 197 | X | X |
| HaplJP066 | DSB_2109 | *Mastigoniscus* | sp. # 3 | H01 | BOLD:ADL3130 | 51174 | ZMH K-57711 | Haploniscidae | W | A3 | 197 | | X |
| HaplJP067 | DSB_2110 | *Haploniscus* | sp. # 2 | H11 | BOLD:ADL2820 | 51175 | ZMH K-57712 | Haploniscidae | W | A3 | 192 | X | |
| HaplJP068 | DSB_2111 | *Mastigoniscus* | sp. # 5 | H20 | BOLD:ADL2987 | 51176 | ZMH K-57713 | Haploniscidae | W | A3 | 192 | X | |
| HaplJP069 | DSB_2112 | *Mastigoniscus* | sp. # 5 | H20 | BOLD:ADL2987 | 51177 | ZMH K-57714 | Haploniscidae | W | A3 | 192 | X | |
| HaplJP070 | DSB_2113 | *Haploniscus* | sp. # 2 | H11 | BOLD:ADL2820 | 51178 | ZMH K-57715 | Haploniscidae | W | A3 | 192 | X | |
| HaplJP071 | DSB_2114 | *Haploniscus* | sp. # 2 | H11 | BOLD:ADL3124 | 51179 | ZMH K-57716 | Haploniscidae | W | A3 | 192 | X | |
| HaplJP072 | DSB_2115 | *Mastigoniscus* | sp. # 8 | H02 | BOLD:ADL3126 | 51180 | ZMH K-57717 | Haploniscidae | W | A3 | 192 | X | |
| HaplJP073 | DSB_2116 | *Mastigoniscus* | sp. # 5 | H20 | BOLD:ADL2987 | 51181 | ZMH K-57718 | Haploniscidae | W | A3 | 192 | X | |
| HaplJP074 | DSB_2117 | *Mastigoniscus* | sp. # 3 | H22 | BOLD:ADL6395 | 51182 | ZMH K-57719 | Haploniscidae | W | IOM | 81 | X | |
| MacrDi04 | DSB_1208 | *Macrostylis* | sp. 03 | M02 | BOLD:ADL4636 | 50271 | SMF 54160 | Macrostylidae | BS | DISCOL | 37 | X | |
| MacrDi09 | DSB_1291 | *Macrostylis* | sp. 03 | M02 | BOLD:ADL3134 | 50356 | SMF 54164 | Macrostylidae | BS | DISCOL | 117 | X | |
| MacrDi06 | DSB_1310 | *Macrostylis* | sp. 04 | M00 | BOLD:ADL9487 | 50375 | SMF 54161 | Macrostylidae | BS | DISCOL | 85 | X | |
| MacrDi07 | DSB_1311 | *Macrostylis* | sp. 03 | M02 | BOLD:ADL4636 | 50376 | SMF 54162 | Macrostylidae | BS | DISCOL | 85 | X | |
| MacrDi08 | DSB_1312 | *Macrostylis* | sp. 03 | M02 | BOLD:ADL4636 | 50377 | SMF 54163 | Macrostylidae | BS | DISCOL | 81 | X | |
| MacrJP001 | DSB_1668 | *Macrostylis* | sp. M cf. *metallicola* | M05 | BOLD:ADG1797 | 50733 | SMF 54165 | Macrostylidae | BS | GC | 20 | X | X |
| MacrJP002 | DSB_1669 | *Macrostylis* | sp. M cf. *metallicola* | M05 | BOLD:ADG1797 | 50734 | SMF 54166 | Macrostylidae | BS | GC | 20 | X | X |
| MacrJP003 | DSB_1670 | *Macrostylis* | sp. M cf. *metallicola* | M05 | BOLD:ADG1798 | 50735 | SMF 54167 | Macrostylidae | BS | GC | 20 | X | X |
| MacrJP004 | DSB_1671 | *Macrostylis* | sp. M cf. *metallicola* | M05 | BOLD:ADG1798 | 50736 | SMF 54168 | Macrostylidae | BS | GC | 20 | X | |
| MacrJP005 | DSB_1672 | *Macrostylis* | sp. # 1 | M01 | BOLD:ADL4493 | 50737 | SMF 54169 | Macrostylidae | BS | GC | 20 | X | |
| MacrJP006 | DSB_1673 | *Macrostylis* | sp. M cf. *metallicola* | M05 | BOLD:ADG1798 | 50738 | SMF 54170 | Macrostylidae | BS | GC | 20 | X | X |
| MacrJP007 | DSB_1674 | *Macrostylis* | sp. M cf. *metallicola* | M05 | BOLD:ADG1798 | 50739 | SMF 54171 | Macrostylidae | BS | GC | 20 | X | X |
| MacrJP008 | DSB_1675 | *Macrostylis* | sp. M cf. *metallicola* | M05 | BOLD:ADG1798 | 50740 | SMF 54172 | Macrostylidae | BS | GC | 20 | X | X |
| MacrJP009 | DSB_1676 | *Macrostylis* | sp. M cf. *metallicola* | M05 | BOLD:ADL3953 | 50741 | SMF 54173 | Macrostylidae | BS | GC | 20 | X | X |
| MacrJP010 | DSB_1677 | *Macrostylis* | sp. M cf. *metallicola* | M01 | | | SMF 54174 | Macrostylidae | BS | GC | 20 | | X |
| MacrJP011 | DSB_1678 | *Macrostylis* | sp. M cf. *metallicola* | M05 | BOLD:ADG1797 | 50743 | SMF 54175 | Macrostylidae | BS | GC | 20 | X | X |
| MacrJP012 | DSB_1679 | *Macrostylis* | sp. M cf. *metallicola* | M05 | BOLD:ADG1798 | 50744 | SMF 54176 | Macrostylidae | BS | GC | 24 | X | X |
| MacrJP013 | DSB_1680 | *Macrostylis* | sp. M cf. *metallicola* | M05 | BOLD:ADL2993 | 50745 | SMF 54177 | Macrostylidae | BS | GC | 24 | X | X |
| MacrJP014 | DSB_1681 | *Macrostylis* | sp. M cf. *metallicola* | M05 | BOLD:ADL4161 | 50746 | SMF 54178 | Macrostylidae | BS | GC | 24 | X | X |
| MacrJP015 | DSB_1682 | *Macrostylis* | sp. M cf. *metallicola* | M05 | BOLD:ADG1797 | 50747 | SMF 54179 | Macrostylidae | BS | GC | 24 | X | X |
| MacrJP016 | DSB_1683 | *Macrostylis* | cf. sp. # 1 | M01 | BOLD:ADL4493 | 50748 | SMF 54180 | Macrostylidae | BS | GC | 24 | X | |



| | | | | | | | | | | | | |
|---|---|---|---|---|---|---|---|---|---|---|---|---|
| MacrJP017 | DSB_1684 | *Macrostylis* | sp. # 1 | M01 | BOLD:ADL4493 | 50749 | SMF 54181 | Macrostylidae | BS | GC | 24 | X | |
| MacrJP018 | DSB_1685 | *Macrostylis* | sp. M cf. *metallicola* | M05 | BOLD:ADG1798 | 50750 | SMF 54182 | Macrostylidae | BS | GC | 24 | X | X |
| MacrJP019 | DSB_1686 | *Macrostylis* | sp. M cf. *metallicola* | M05 | BOLD:ADG1797 | 50751 | SMF 54183 | Macrostylidae | BS | GC | 24 | X | X |
| MacrJP020 | DSB_1687 | *Macrostylis* | sp. M cf. *metallicola* | M05 | BOLD:ADL2993 | 50752 | SMF 54184 | Macrostylidae | BS | GC | 59 | X | X |
| MacrJP021 | DSB_1688 | *Macrostylis* | sp. M cf. *metallicola* | M05 | BOLD:ADL2993 | 50753 | SMF 54185 | Macrostylidae | BS | GC | 59 | X | X |
| MacrJP022 | DSB_1689 | *Macrostylis* | sp. M cf. *metallicola* | M05 | BOLD:ADL2993 | 50754 | SMF 54186 | Macrostylidae | BS | GC | 59 | X | X |
| MacrJP023 | DSB_1690 | *Macrostylis* | sp. M cf. *metallicola* | M05 | BOLD:ADL4161 | 50755 | SMF 54187 | Macrostylidae | BS | GC | 59 | X | X |
| MacrJP024 | DSB_1691 | *Macrostylis* | sp. M cf. *metallicola* | M05 | BOLD:ADL2993 | 50756 | SMF 54188 | Macrostylidae | BS | GC | 59 | X | X |
| MacrJP025 | DSB_1692 | *Macrostylis* | sp. M cf. *metallicola* | M05 | BOLD:ADL2993 | 50757 | SMF 54189 | Macrostylidae | BS | GC | 59 | X | X |
| MacrJP026 | DSB_1693 | *Macrostylis* | sp. M cf. *metallicola* | M06 | BOLD:ADL4496 | 50758 | SMF 54190 | Macrostylidae | BS | GC | 50 | X | X |
| MacrJP027 | DSB_1694 | *Macrostylis* | sp. M cf. *metallicola* | M06 | BOLD:ADL4496 | 50759 | SMF 54191 | Macrostylidae | BS | GC | 50 | X | X |
| MacrJP028 | DSB_1695 | *Macrostylis* | sp. M cf. *metallicola* | M06 | BOLD:ADL4496 | 50760 | SMF 54192 | Macrostylidae | BS | GC | 50 | X | X |
| MacrJP029 | DSB_1696 | *Macrostylis* | sp. M cf. *metallicola* | M05 | BOLD:ADL2993 | 50761 | SMF 54193 | Macrostylidae | BS | GC | 50 | X | X |
| MacrJP030 | DSB_1697 | *Macrostylis* | sp. M cf. *metallicola* | M06 | BOLD:ADL4496 | 50762 | SMF 54194 | Macrostylidae | BS | GC | 50 | X | X |
| MacrJP031 | DSB_1698 | *Macrostylis* | sp. M cf. *metallicola* | M06 | BOLD:ADL4496 | 50763 | SMF 54195 | Macrostylidae | BS | GC | 50 | X | X |
| MacrJP032 | DSB_1699 | *Macrostylis* | sp. M cf. *metallicola* | M06 | BOLD:ADL4496 | 50764 | SMF 54196 | Macrostylidae | BS | GC | 50 | X | X |
| MacrJP033 | DSB_1700 | *Macrostylis* | sp. M cf. *metallicola* | M06 | BOLD:ADL4496 | 50765 | SMF 54197 | Macrostylidae | BS | GC | 50 | X | X |
| MacrJP034 | DSB_1701 | *Macrostylis* | sp. M cf. *metallicola* | M06 | BOLD:ADL4496 | 50766 | SMF 54198 | Macrostylidae | BS | GC | 50 | X | X |
| MacrJP035 | DSB_1702 | *Macrostylis* | sp. M cf. *metallicola* | M05 | BOLD:ADL2993 | 50767 | SMF 54199 | Macrostylidae | BS | IOM | 81 | X | X |
| MacrJP036 | DSB_1703 | *Macrostylis* | sp. M cf. *metallicola* | M05 | BOLD:ADL4637 | 50768 | SMF 54200 | Macrostylidae | BS | IOM | 81 | X | X |
| MacrJP037 | DSB_1704 | *Macrostylis* | sp. M cf. *metallicola* | M05 | BOLD:ADL2993 | 50769 | SMF 54201 | Macrostylidae | BS | IOM | 81 | X | X |
| MacrJP038 | DSB_1705 | *Macrostylis* | cf. sp. # 1 | M01 | BOLD:ADL4493 | 50770 | SMF 54202 | Macrostylidae | BS | IOM | 81 | X | |
| MacrJP039 | DSB_1706 | *Macrostylis* | sp. M cf. *metallicola* | M05 | BOLD:ADL4637 | 50771 | SMF 54203 | Macrostylidae | BS | IOM | 99 | X | X |
| MacrJP040 | DSB_1707 | *Macrostylis* | sp. M cf. *metallicola* | M05 | BOLD:ADL4637 | 50772 | SMF 54204 | Macrostylidae | BS | IOM | 99 | X | X |
| MacrJP041 | DSB_1708 | *Macrostylis* | sp. M cf. *metallicola* | M05 | | 50773 | SMF 54205 | Macrostylidae | BS | IOM | 99 | X | |
| MacrJP042 | DSB_1709 | *Macrostylis* | sp. M cf. *metallicola* | M05 | BOLD:ADL4637 | 50774 | SMF 54206 | Macrostylidae | BS | IOM | 99 | X | X |
| MacrJP043 | DSB_1710 | *Macrostylis* | sp. M cf. *metallicola* | M05 | BOLD:ADL4490 | 50775 | SMF 54207 | Macrostylidae | BS | BC | 118 | X | X |
| MacrJP044 | DSB_1711 | *Macrostylis* | sp. M cf. *metallicola* | M05 | BOLD:ADL4490 | 50776 | SMF 54208 | Macrostylidae | BS | BC | 118 | X | X |
| MacrJP045 | DSB_1712 | *Macrostylis* | sp. M cf. *metallicola* | M05 | BOLD:ADL4490 | 50777 | SMF 54209 | Macrostylidae | BS | BC | 118 | X | X |
| MacrJP046 | DSB_1713 | *Macrostylis* | sp. M cf. *metallicola* | M07 | BOLD:ADL3132 | 50778 | SMF 54210 | Macrostylidae | BS | BC | 118 | X | X |
| MacrJP047 | DSB_1714 | *Macrostylis* | sp. M cf. *metallicola* | M07 | BOLD:ADL3132 | 50779 | SMF 54211 | Macrostylidae | BS | BC | 118 | X | X |
| MacrJP048 | DSB_1715 | *Macrostylis* | sp. M cf. *metallicola* | M05 | BOLD:ADL4490 | 50780 | SMF 54212 | Macrostylidae | BS | BC | 133 | X | X |
| MacrJP049 | DSB_1716 | *Macrostylis* | sp. M cf. *metallicola* | M05 | BOLD:ADL4490 | 50781 | SMF 54213 | Macrostylidae | BS | BC | 133 | X | X |
| MacrJP050 | DSB_1717 | *Macrostylis* | sp. M cf. *metallicola* | M03 | BOLD:ADL3133 | 50782 | SMF 54214 | Macrostylidae | BS | BC | 133 | X | |
| MacrJP052 | DSB_1719 | *Macrostylis* | sp. M cf. *metallicola* | M05 | BOLD:ADL4490 | 50784 | SMF 54215 | Macrostylidae | BS | BC | 133 | X | X |
| MacrJP053 | DSB_1720 | *Macrostylis* | sp. M cf. *metallicola* | M03 | BOLD:ADL3133 | 50785 | SMF 54216 | Macrostylidae | BS | BC | 133 | X | |
| MacrJP054 | DSB_1721 | *Macrostylis* | sp. M cf. *metallicola* | M03 | BOLD:ADL3133 | 50786 | SMF 54217 | Macrostylidae | BS | BC | 133 | X | |
| MacrJP055 | DSB_1722 | *Macrostylis* | sp. M cf. *metallicola* | M03 | BOLD:ADL9486 | 50787 | SMF 54218 | Macrostylidae | BS | FC | 158 | X | X |
| MacrJP057 | DSB_1724 | *Macrostylis* | sp. M cf. *metallicola* | M03 | BOLD:ADL9675 | 50789 | SMF 54219 | Macrostylidae | BS | FC | 171 | X | X |
| MacrJP058 | DSB_1725 | *Macrostylis* | sp. M cf. *metallicola* | M05 | BOLD:ADL4492 | 50790 | SMF 54220 | Macrostylidae | BS | FC | 171 | X | X |
| MacrJP059 | DSB_1726 | *Macrostylis* | sp. M cf. *metallicola* | M05 | BOLD:ADL3135 | 50791 | SMF 54221 | Macrostylidae | BS | FC | 171 | X | X |
| MacrJP060 | DSB_1727 | *Macrostylis* | sp. M cf. *metallicola* | M03 | BOLD:ADL9300 | 50792 | SMF 54222 | Macrostylidae | BS | FC | 171 | X | |
| MacrJP061 | DSB_1728 | *Macrostylis* | sp. M cf. *metallicola* | M03 | BOLD:ADL9300 | 50793 | SMF 54223 | Macrostylidae | BS | FC | 171 | X | |
| MacrJP062 | DSB_1729 | *Macrostylis* | sp. M cf. *metallicola* | M03 | BOLD:ADL9674 | 50794 | SMF 54224 | Macrostylidae | BS | FC | 171 | X | |
| MacrJP063 | DSB_1730 | *Macrostylis* | sp. M cf. *metallicola* | M05 | BOLD:ADL3136 | 50795 | SMF 54225 | Macrostylidae | BS | FC | 171 | X | X |
| MacrJP064 | DSB_1731 | *Macrostylis* | sp. M cf. *metallicola* | M05 | BOLD:ADL4492 | 50796 | SMF 54226 | Macrostylidae | BS | FC | 171 | X | X |
| MacrJP065 | DSB_1732 | *Macrostylis* | sp. M cf. *metallicola* | M05 | BOLD:ADL4492 | 50797 | SMF 54227 | Macrostylidae | BS | FC | 171 | X | X |
| MacrJP066 | DSB_1733 | *Macrostylis* | sp. M cf. *metallicola* | M05 | BOLD:ADL4492 | 50798 | SMF 54228 | Macrostylidae | BS | FC | 171 | X | X |
| MacrJP067 | DSB_1734 | *Macrostylis* | sp. M cf. *metallicola* | M03 | BOLD:ADL9300 | 50799 | SMF 54229 | Macrostylidae | BS | FC | 171 | X | X |
| MacrJP068 | DSB_1735 | *Macrostylis* | sp. M cf. *metallicola* | M05 | BOLD:ADL3135 | 50800 | SMF 54230 | Macrostylidae | BS | FC | 171 | X | X |
| MacrJP069 | DSB_1736 | *Macrostylis* | sp. M cf. *metallicola* | M05 | BOLD:AAG9589 | 50801 | SMF 54231 | Macrostylidae | BS | FC | 171 | X | X |
| MacrJP070 | DSB_1737 | *Macrostylis* | sp. M cf. *metallicola* | M05 | BOLD:ADL3135 | 50802 | SMF 54232 | Macrostylidae | BS | FC | 171 | X | X |
| MacrJP071 | DSB_1738 | *Macrostylis* | sp. M cf. *metallicola* | M05 | BOLD:ADL4492 | 50803 | SMF 54233 | Macrostylidae | BS | FC | 171 | X | X |
| MacrJP072 | DSB_1739 | *Macrostylis* | sp. M cf. *metallicola* | M03 | BOLD:ADL9675 | 50804 | SMF 54234 | Macrostylidae | BS | FC | 171 | X | |
| MacrJP073 | DSB_1740 | *Macrostylis* | sp. M cf. *metallicola* | M05 | BOLD:ADL4492 | 50805 | SMF 54235 | Macrostylidae | BS | A3 | 192 | X | X |
| MacrJP074 | DSB_1741 | *Macrostylis* | sp. # 2 | M08 | BOLD:ADL4491 | 50806 | SMF 54236 | Macrostylidae | BS | A3 | 210 | X | X |
| MacrJP075 | DSB_1742 | *Macrostylis* | sp. M cf. *metallicola* | M04 | BOLD:ADL4494 | 50807 | SMF 54237 | Macrostylidae | BS | A3 | 210 | X | X |
| MacrJP076 | DSB_1743 | *Macrostylis* | sp. M cf. *metallicola* | M04 | BOLD:ADL4494 | 50808 | SMF 54238 | Macrostylidae | BS | A3 | 197 | X | X |
| MacrJP077 | DSB_1744 | *Macrostylis* | sp. M cf. *metallicola* | M04 | BOLD:ADL4494 | 50809 | SMF 54239 | Macrostylidae | BS | A3 | 197 | X | X |
| MacrJP078 | DSB_1745 | *Macrostylis* | sp. M cf. *metallicola* | M04 | BOLD:ADL4494 | 50810 | SMF 54240 | Macrostylidae | BS | A3 | 197 | X | X |
| MacrJP079 | DSB_1746 | *Macrostylis* | sp. M cf. *metallicola* | M04 | BOLD:ADL4494 | 50811 | SMF 54241 | Macrostylidae | BS | A3 | 197 | X | X |
| MacrJP080 | DSB_1747 | *Macrostylis* | sp. M cf. *metallicola* | M04 | BOLD:ADL4494 | 50812 | SMF 54242 | Macrostylidae | BS | A3 | 197 | X | X |
| MacrJP081 | DSB_1748 | *Macrostylis* | sp. M cf. *metallicola* | M04 | BOLD:ADL4494 | 50813 | SMF 54243 | Macrostylidae | BS | A3 | 192 | X | X |
| MacrJP082 | DSB_1749 | *Macrostylis* | sp. M cf. *metallicola* | M04 | BOLD:ADL4494 | 50814 | SMF 54244 | Macrostylidae | BS | A3 | 192 | X | X |



| | | | | | | | | | | | | | |
|---|---|---|---|---|---|---|---|---|---|---|---|---|---|
| MacrJP083 | DSB_1750 | *Macrostylis* | sp. # 2 | M08 | BOLD:ADL4497 | 50815 | SMF 54245 | Macrostylidae | BS | A3 | 192 | X | X |
| MacrJP084 | DSB_1751 | *Macrostylis* | sp. # 2 | M08 | BOLD:ADL4497 | 50816 | SMF 54246 | Macrostylidae | BS | A3 | 192 |  | X |
| MacrJP085 | DSB_1752 | *Macrostylis* | sp. # 2 | M08 | BOLD:ADL4491 | 50817 | SMF 54247 | Macrostylidae | BS | A3 | 192 | X | X |
| MacrJP086 | DSB_1753 | *Macrostylis* | sp. # 2 | M08 | BOLD:ADL4491 | 50818 | SMF 54248 | Macrostylidae | BS | A3 | 192 |  | X |
| MacrJP087 | DSB_1754 | *Macrostylis* | sp. # 2 | M08 | BOLD:ADL4491 | 50819 | SMF 54249 | Macrostylidae | BS | A3 | 192 | X | X |
| MacrJP088 | DSB_1755 | *Macrostylis* | sp. # 2 | M08 | BOLD:ADL4491 | 50820 | SMF 54250 | Macrostylidae | BS | A3 | 192 | X | X |
| MacrJP089 | DSB_1756 | *Macrostylis* | sp. M cf. *metallicola* | M04 | BOLD:ADL4494 | 50821 | SMF 54251 | Macrostylidae | BS | A3 | 192 | X | X |
| MacrJP090 | DSB_1757 | *Macrostylis* | sp. M cf. *metallicola* | M04 | BOLD:ADL4494 | 50822 | SMF 54252 | Macrostylidae | BS | A3 | 192 | X | X |
| MacrJP091 | DSB_1758 | *Macrostylis* | sp. # 2 | M08 | BOLD:ADL4495 | 50823 | SMF 54253 | Macrostylidae | BS | A3 | 192 | X | X |
| MunpDi02 | DSB_1203 | *Eurycope* | sp. 05 | Mu28 | BOLD:ADL4531 | 50266 | SMF 54254 | Munnopsidae | S | DISCOL | 37 | X | X |
| MunpDi03 | DSB_1204 | *Betamorpha* | sp. 01 | Mu20 | BOLD:ADM1455 | 50267 | SMF 54255 | Munnopsidae | S | DISCOL | 37 | X | X |
| MunpDi05 | DSB_1220 | *Eurycope* | sp. 05 | Mu28 | BOLD:ADL4531 | 50283 | SMF 54256 | Munnopsidae | S | DISCOL | 81 | X | X |
| MunpDi08 | DSB_1223 | *Eurycope* | sp. 05 | Mu29 | BOLD:ADL4916 | 50286 | SMF 54259 | Munnopsidae | S | DISCOL | 85 | X | X |
| MunpDi09 | DSB_1224 | *Munneurycope* | sp. 01 | Mu80 | BOLD:ADL4913 | 50287 | SMF 54260 | Munnopsidae | S | DISCOL | 85 | X | X |
| MunpDi06 | DSB_1225 | *Eurycope* | sp. 05 | Mu28 | BOLD:ADL4531 | 50288 | SMF 54257 | Munnopsidae | S | DISCOL | 81 | X | X |
| MunpDi07 | DSB_1229 | *Betamorpha* | sp. 02 | Mu45 | BOLD:ADL3690 | 50294 | SMF 54258 | Munnopsidae | S | DISCOL | 81 | X |  |
| MunpDi12 | DSB_1230 | *Betamorpha* | sp. 02 | Mu45 | BOLD:ADL3690 | 50295 | SMF 54261 | Munnopsidae | S | DISCOL | 85 | X |  |
| MunpDi13 | DSB_1231 | *Stortyngura* | *parka* | Mu62 | BOLD:ADL3683 | 50296 | SMF 54262 | Munnopsidae | S | DISCOL | 93 | X |  |
| MunpDi15 | DSB_1233 | *Disconectes* | sp. 01 | Mu02 | BOLD:ADL3686 | 50298 | SMF 54263 | Munnopsidae | S | DISCOL | 93 | X |  |
| MunpDi16 | DSB_1234 | *Rectisura* | sp. 01 | Mu57 | BOLD:ADL3689 | 50299 | SMF 54264 | Munnopsidae | S | DISCOL | 93 | X |  |
| MunpDi20 | DSB_1238 | *Munneurycope* | sp. 03 | Mu63 | BOLD:ADL5073 | 50303 | SMF 54265 | Munnopsidae | S | DISCOL | 93 | X | X |
| MunpDi21 | DSB_1239 | *Disconectes* | sp. 01 | Mu07 | BOLD:ADL4533 | 50304 | SMF 54266 | Munnopsidae | S | DISCOL | 93 | X | X |
| MunpDi22 | DSB_1240 | *Eurycope* | sp. 05 | Mu37 | BOLD:ADL5069 | 50305 | SMF 54267 | Munnopsidae | S | DISCOL | 93 | X | X |
| MunpDi23 | DSB_1241 | *Munneurycope sp 6* | sp. 01 | Mu48 | BOLD:ADL4419 | 50306 | SMF 54268 | Munnopsidae | S | DISCOL | 93 | X | X |
| MunpDi24 | DSB_1242 | *Eurycope* | sp. 05 | Mu27 | BOLD:ADL5075 | 50307 | SMF 54269 | Munnopsidae | S | DISCOL | 93 | X | X |
| MunpDi26 | DSB_1244 | *Disconectes* | sp. 02 | Mu76 |  | 50309 | SMF 54270 | Munnopsidae | S | DISCOL | 93 |  | X |
| MunpDi27 | DSB_1245 | *Disconectes* | sp. 02 | Mu72 | BOLD:ADL5096 | 50310 | SMF 54271 | Munnopsidae | S | DISCOL | 93 | X | X |
| MunpDi28 | DSB_1246 | *Munneurycope* | sp. 01 | Mu63 | BOLD:ADL5073 | 50311 | SMF 54272 | Munnopsidae | S | DISCOL | 93 | X | X |
| MunpDi29 | DSB_1247 | *Stortyngura* | *parka* | Mu62 | BOLD:ADL3683 | 50312 | SMF 54273 | Munnopsidae | S | DISCOL | 93 | X |  |
| MunpDi30 | DSB_1248 | *Disconectes* | sp. 02 | Mu72 | BOLD:ADL5096 | 50313 | SMF 54274 | Munnopsidae | S | DISCOL | 93 | X | X |
| MunpDi31 | DSB_1249 | *Paralipomera* | sp. 01 | Mu84 | BOLD:ADL9556 | 50314 | SMF 54275 | Munnopsidae | S | DISCOL | 93 | X | X |
| MunpDi32 | DSB_1250 | *Disconectes* | sp. 02a | Mu72 | BOLD:ADL5096 | 50315 | SMF 54276 | Munnopsidae | S | DISCOL | 93 | X | X |
| MunpDi33 | DSB_1253 | *Eurycope* | sp. | Mu37 | BOLD:ADL4914 | 50318 | SMF 54277 | Munnopsidae | S | DISCOL | 85 | X | X |
| MunpDi36 | DSB_1256 | *Disconectes* | sp. 01 | Mu07 |  | 50321 | SMF 54278 | Munnopsidae | S | DISCOL | 93 |  | X |
| MunpDi37 | DSB_1257 | *Paralipomera* | sp. 01 | Mu84 | BOLD:ADL9556 | 50322 | SMF 54279 | Munnopsidae | S | DISCOL | 93 | X | X |
| MunpDi38 | DSB_1258 | *Munneurycope* | sp. 03 | Mu80 | BOLD:ADL4913 | 50323 | SMF 54280 | Munnopsidae | S | DISCOL | 93 | X | X |
| MunpDi41 | DSB_1259 | *Disconectes* | sp. 02 | Mu72 | BOLD:ADL5096 | 50324 | SMF 54281 | Munnopsidae | S | DISCOL | 117 | X | X |
| MunpDi43 | DSB_1261 | *Eurycope* | sp. 01 | Mu37 | BOLD:ADL4914 | 50326 | SMF 54282 | Munnopsidae | S | DISCOL | 122 | X | X |
| MunpDi45 | DSB_1263 | *Disconectes* | sp. 01 | Mu02 | BOLD:ADL3686 | 50328 | SMF 54283 | Munnopsidae | S | DISCOL | 122 | X |  |
| MunpDi46 | DSB_1264 | *Disconectes* | sp. 01 | Mu07 | BOLD:ADL4533 | 50329 | SMF 54284 | Munnopsidae | S | DISCOL | 122 | X |  |
| MunpDi47 | DSB_1265 | *Munneurycope* | sp. 03 | Mu63 | BOLD:ADL3688 | 50330 | SMF 54285 | Munnopsidae | S | DISCOL | 122 | X | X |
| MunpDi49 | DSB_1267 | *Disconectes* | sp. 01 | Mu75 | BOLD:ADL4918 | 50332 | SMF 54286 | Munnopsidae | S | DISCOL | 122 | X | X |
| MunpDi50 | DSB_1268 | *Disconectes* | sp. 01 | Mu07 | BOLD:ADL4533 | 50333 | SMF 54287 | Munnopsidae | S | DISCOL | 122 | X | X |
| MunpDi51 | DSB_1269 | *Munneurycope* | sp. | Mu63 | BOLD:ADL5073 | 50334 | SMF 54288 | Munnopsidae | S | DISCOL | 122 | X | X |
| MunpDi53 | DSB_1271 | *Rectisura* | sp. 02 | Mu57 | BOLD:ADL3687 | 50336 | SMF 54289 | Munnopsidae | S | DISCOL | 122 | X |  |
| MunpDi54 | DSB_1272 | *Disconectes* | sp. 02 | Mu72 | BOLD:ADL3684 | 50337 | SMF 54290 | Munnopsidae | S | DISCOL | 117 | X | X |
| MunpDi55 | DSB_1273 | *Betamorpha* | sp. 02 | Mu45 | BOLD:ADL3690 | 50338 | SMF 54291 | Munnopsidae | S | DISCOL | 122 | X |  |
| MunpDi56 | DSB_1274 | *Disconectes* | sp. 01 | Mu02 | BOLD:ADL3686 | 50339 | SMF 54292 | Munnopsidae | S | DISCOL | 122 | X |  |
| MunpDi57 | DSB_1275 | *Disconectes* | sp. 02 | Mu72 | BOLD:ADL5096 | 50340 | SMF 54293 | Munnopsidae | S | DISCOL | 122 | X | X |
| MunpDi60 | DSB_1278 | *Betamorpha* | sp. 01 | Mu20 | BOLD:ADM1455 | 50343 | SMF 54294 | Munnopsidae | S | DISCOL | 126 | X |  |
| MunpDi62 | DSB_1280 | *Munnopsis* | *abyssalis* /sp. 01 | Mu39 | BOLD:ADL3685 | 50345 | SMF 54295 | Munnopsidae | S | DISCOL | 126 | X | X |
| MunpDi63 | DSB_1281 | *Betamorpha* | sp. 01 | Mu45 | BOLD:ADL3690 | 50346 | SMF 54296 | Munnopsidae | S | DISCOL | 126 | X |  |
| MunpDi64 | DSB_1282 | *Paramunnopsis* | sp. 01 | Mu38 | BOLD:ADL3456 | 50347 | SMF 54297 | Munnopsidae | S | DISCOL | 126 | X | X |
| MunpDi65 | DSB_1283 | *Disconectes* | sp. 03 | Mu01 |  | 50348 | SMF 54298 | Munnopsidae | S | DISCOL | 126 |  | X |
| MunpDi66 | DSB_1284 | *Paralipomera* | sp. 01 | Mu84 | BOLD:ADL9556 | 50349 | SMF 54299 | Munnopsidae | S | DISCOL | 126 | X | X |
| MunpJP001 | DSB_1386 |  | sp. 1 | Mu11 | BOLD:ADL4575 | 50451 | SMF 54300 | Munnopsidae | S | GC | 50 | X |  |
| MunpJP002 | DSB_1387 |  | sp. 1 | Mu11 | BOLD:ADL4593 | 50452 | SMF 54301 | Munnopsidae | S | GC | 50 | X |  |
| MunpJP003 | DSB_1388 | *Disconectes* | sp. 1 | Mu08 | BOLD:ADL3511 | 50453 | SMF 54302 | Munnopsidae | S | GC | 50 | X | X |
| MunpJP004 | DSB_1389 | *Eurycope* | sp. 5 | Mu26 | BOLD:ADL4378 | 50454 | SMF 54303 | Munnopsidae | S | GC | 50 | X | X |
| MunpJP005 | DSB_1390 | *Disconectes* | sp. 1 | Mu11 | BOLD:ADL4575 | 50455 | SMF 54304 | Munnopsidae | S | GC | 50 | X | X |
| MunpJP006 | DSB_1391 | *Disconectes* | sp. 1 | Mu11 | BOLD:ADL4575 | 50456 | SMF 54305 | Munnopsidae | S | GC | 50 | X | X |
| MunpJP008 | DSB_1393 | *Disconectes* | sp. 1 | Mu11 | BOLD:ADL4593 | 50458 | SMF 54306 | Munnopsidae | S | GC | 50 | X |  |
| MunpJP009 | DSB_1394 | *Disconectes* | sp. 3 | Mu11 | BOLD:ADL4379 | 50459 | SMF 54307 | Munnopsidae | S | GC | 50 | X | X |
| MunpJP010 | DSB_1395 | *Eurycope* | sp. 5 | Mu26 | BOLD:ADL4378 | 50460 | SMF 54308 | Munnopsidae | S | GC | 50 | X | X |





| ID | DSB | Genus | species | Mu | BOLD | No. | SMF | Family | S | Loc | Depth | | |
|---|---|---|---|---|---|---|---|---|---|---|---|---|---|
| MunpJP012 | DSB_1397 | *Disconectes* | sp. 1 | Mu08 | BOLD:ADL5266 | 50462 | SMF 54309 | Munnopsidae | S | GC | 50 | X | X |
| MunpJP013 | DSB_1398 | *Disconectes* | sp. 1 | Mu08 | BOLD:ADL3511 | 50463 | SMF 54310 | Munnopsidae | S | GC | 50 | X | X |
| MunpJP014 | DSB_1399 | *Syneurycope* | sp. | Mu86 | BOLD:ADM2063 | 50464 | SMF 54311 | Munnopsidae | S | GC | 50 | X | X |
| MunpJP016 | DSB_1400 | *Disconectes* | sp. 1 | Mu11 | BOLD:ADL4575 | 50465 | SMF 54312 | Munnopsidae | S | GC | 50 | X | X |
| MunpJP015 | DSB_1401 | *Munneurycope* | sp. | Mu67 | BOLD:ADL2777 | 50466 | SMF 54313 | Munnopsidae | S | GC | 50 | X | X |
| MunpJP018 | DSB_1402 | *Disconectes* | sp. 1 | Mu11 | BOLD:ADL5267 | 50467 | SMF 54314 | Munnopsidae | S | GC | 50 | X | X |
| MunpJP019 | DSB_1403 | *Disconectes* | sp. 1 | Mu11 | BOLD:ADL4575 | 50468 | SMF 54315 | Munnopsidae | S | GC | 50 | X | X |
| MunpJP020 | DSB_1404 | *Disconectes* | sp. 1 | Mu11 | BOLD:ADL4379 | 50469 | SMF 54316 | Munnopsidae | S | GC | 50 | X | X |
| MunpJP021 | DSB_1405 | *Eurycope* | sp. 3 | Mu22 | BOLD:ADL2909 | 50470 | SMF 54317 | Munnopsidae | S | GC | 50 | X | X |
| MunpJP022 | DSB_1406 | *Disconectes* | sp. 1 | Mu11 | BOLD:ADL4575 | 50471 | SMF 54318 | Munnopsidae | S | GC | 50 | X | X |
| MunpJP023 | DSB_1407 | *Disconectes* | sp. 1 | Mu11 | BOLD:ADL4575 | 50472 | SMF 54319 | Munnopsidae | S | GC | 50 | X | X |
| MunpJP024 | DSB_1408 | *Paramunnopsis* | indet | Mu38 | BOLD:ADL3456 | 50473 | SMF 54320 | Munnopsidae | S | GC | 50 | X | X |
| MunpJP025 | DSB_1409 | *Disconectes* | sp. 1 | Mu11 | BOLD:ADL4593 | 50474 | SMF 54321 | Munnopsidae | S | GC | 50 | X | X |
| MunpJP026 | DSB_1410 | *Disconectes* | sp. 1 | Mu11 | BOLD:ADL4575 | 50475 | SMF 54322 | Munnopsidae | S | GC | 50 | X | X |
| MunpJP027 | DSB_1411 | *Disconectes* | sp. 2a | Mu37 | BOLD:ADL5268 | 50476 | SMF 54323 | Munnopsidae | S | GC | 50 | X | X |
| MunpJP028 | DSB_1412 | *Eurycope* | sp. 5 | Mu37 | BOLD:ADL3657 | 50477 | SMF 54324 | Munnopsidae | S | GC | 50 | X | X |
| MunpJP029 | DSB_1413 | *Disconectes* | sp. 1 | Mu11 | BOLD:ADL4575 | 50478 | SMF 54325 | Munnopsidae | S | GC | 50 | | X |
| MunpJP032 | DSB_1416 | *Disconectes* | sp. 1 | Mu08 | BOLD:ADL5266 | 50481 | SMF 54326 | Munnopsidae | S | GC | 50 | X | X |
| MunpJP033 | DSB_1417 | *Disconectes* | sp. 1 | Mu11 | BOLD:ADL4575 | 50482 | SMF 54327 | Munnopsidae | S | GC | 50 | X | X |
| MunpJP034 | DSB_1418 | *Munneurycope* | sp. | Mu67 | BOLD:ADL2777 | 50483 | SMF 54328 | Munnopsidae | S | GC | 20 | X | X |
| MunpJP035 | DSB_1419 | *Munneurycope* | sp. 3 | Mu67 | BOLD:ADL2777 | 50484 | SMF 54329 | Munnopsidae | S | GC | 20 | X | X |
| MunpJP036 | DSB_1420 | *Munneurycope* | indet | Mu81 | BOLD:ADL4576 | 50485 | SMF 54330 | Munnopsidae | S | GC | 20 | X | X |
| MunpJP038 | DSB_1422 | *Eurycope* | sp. 5 | Mu35 | BOLD:ADL4447 | 50487 | SMF 54331 | Munnopsidae | S | GC | 20 | X | |
| MunpJP039 | DSB_1423 | *Eurycope* | sp. 5 | Mu37 | BOLD:ADL3657 | 50488 | SMF 54332 | Munnopsidae | S | GC | 20 | X | X |
| MunpJP041 | DSB_1425 | *Betamorpha* | indet | Mu70 | BOLD:ADL4078 | 50490 | SMF 54333 | Munnopsidae | S | GC | 20 | X | X |
| MunpJP042 | DSB_1426 | *Munneurycope* | indet | Mu81 | BOLD:ADL4576 | 50491 | SMF 54334 | Munnopsidae | S | GC | 20 | X | X |
| MunpJP043 | DSB_1427 | *Disconectes* | sp. 1 | Mu08 | BOLD:ADL3511 | 50492 | SMF 54335 | Munnopsidae | S | GC | 20 | X | |
| MunpJP044 | DSB_1428 | *Munneurycope* | indet | Mu81 | BOLD:ADL4576 | 50493 | SMF 54336 | Munnopsidae | S | GC | 20 | X | |
| MunpJP046 | DSB_1430 | *Eurycope* | sp. 5 | Mu32 | BOLD:ADL4077 | 50495 | SMF 54337 | Munnopsidae | S | GC | 20 | X | X |
| MunpJP047 | DSB_1431 | *Eurycope* | sp. 5 | Mu32 | BOLD:ADL4077 | 50496 | SMF 54338 | Munnopsidae | S | GC | 20 | X | |
| MunpJP049 | DSB_1433 | *Eurycope* | sp. 5 | Mu37 | BOLD:ADL3657 | 50498 | SMF 54339 | Munnopsidae | S | GC | 20 | X | X |
| MunpJP051 | DSB_1435 | *Eurycope* | sp. 5 | Mu37 | BOLD:ADL3657 | 50500 | SMF 54340 | Munnopsidae | S | GC | 20 | X | X |
| MunpJP052 | DSB_1436 | *Disconectes* | sp. 1 | Mu08 | | 50501 | SMF 54341 | Munnopsidae | S | GC | 20 | | X |
| MunpJP053 | DSB_1437 | *Disconectes* | sp. 1 | Mu08 | BOLD:ADL3511 | 50502 | SMF 54342 | Munnopsidae | S | GC | 20 | X | X |
| MunpJP054 | DSB_1438 | *Disconectes* | sp. | Mu08 | BOLD:ADL5271 | 50503 | SMF 54343 | Munnopsidae | S | GC | 20 | X | X |
| MunpJP055 | DSB_1439 | *Munneurycope* | indet | Mu67 | BOLD:ADL2777 | 50504 | SMF 54344 | Munnopsidae | S | BC | 133 | X | X |
| MunpJP056 | DSB_1440 | *Munneurycope* | indet | Mu63 | BOLD:ADL3656 | 50505 | SMF 54345 | Munnopsidae | S | BC | 133 | X | X |
| MunpJP057 | DSB_1441 | *Syneurycope* | sp. | Mu85 | BOLD:ADM0298 | 50506 | SMF 54346 | Munnopsidae | S | FC | 171 | X | X |
| MunpJP058 | DSB_1442 | *Eurycope* | sp. 5 | Mu22 | BOLD:ADL4573 | 50507 | SMF 54347 | Munnopsidae | S | FC | 171 | X | X |
| MunpJP060 | DSB_1444 | *Bellibos* | sp. 2 | Mu43 | BOLD:ADL3457 | 50509 | SMF 54348 | Munnopsidae | S | A3 | 192 | X | X |
| MunpJP061 | DSB_1445 | *Acanthocope* | cf. galatheae | Mu56 | BOLD:ADL4687 | 50510 | SMF 54349 | Munnopsidae | S | A3 | 192 | X | X |
| MunpJP062 | DSB_1446 | *Acanthocope* | cf. galatheae | Mu56 | BOLD:ADL4977 | 50511 | SMF 54350 | Munnopsidae | S | A3 | 192 | X | X |
| MunpJP063 | DSB_1447 | *Acanthocope* | cf. galatheae | Mu56 | BOLD:ADL4687 | 50512 | SMF 54351 | Munnopsidae | S | A3 | 192 | X | X |
| MunpJP064 | DSB_1448 | *Disconectes* | sp. 1 | Mu11 | BOLD:ADL4379 | 50513 | SMF 54352 | Munnopsidae | S | GC | 59 | X | X |
| MunpJP065 | DSB_1449 | *Disconectes* | sp. 1 | Mu11 | BOLD:ADL5272 | 50514 | SMF 54353 | Munnopsidae | S | GC | 59 | X | X |
| MunpJP066 | DSB_1450 | *Disconectes* | sp. 1 | Mu11 | BOLD:ADL4575 | 50515 | SMF 54354 | Munnopsidae | S | GC | 59 | X | |
| MunpJP067 | DSB_1451 | *Betamorpha* | indet | Mu69 | BOLD:ADL4076 | 50516 | SMF 54355 | Munnopsidae | S | A3 | 197 | X | X |
| MunpJP068 | DSB_1452 | *Munneurycope* | indet | Mu68 | BOLD:ADL4075 | 50517 | SMF 54356 | Munnopsidae | S | A3 | 197 | X | X |
| MunpJP069 | DSB_1453 | *Munneurycope* | indet | Mu79 | BOLD:ADL4074 | 50518 | SMF 54357 | Munnopsidae | S | A3 | 197 | X | X |
| MunpJP070 | DSB_1454 | *Betamorpha* | fusiformis | Mu47 | BOLD:ADL4073 | 50519 | SMF 54358 | Munnopsidae | S | A3 | 197 | X | |
| MunpJP071 | DSB_1455 | *Syneurycope* | indet | Mu86 | BOLD:ADM2063 | 50520 | SMF 54359 | Munnopsidae | S | GC | 59 | X | X |
| MunpJP072 | DSB_1456 | *Disconectes* | sp. 1 | Mu11 | BOLD:ADL4575 | 50521 | SMF 54360 | Munnopsidae | S | GC | 59 | X | X |
| MunpJP073 | DSB_1457 | *Rectisura* | slavai | Mu59 | BOLD:ADL5293 | 50522 | SMF 54361 | Munnopsidae | S | BC | 118 | X | X |
| MunpJP074 | DSB_1458 | *Munneurycope* | indet | Mu64 | BOLD:ADL3754 | 50523 | SMF 54362 | Munnopsidae | S | BC | 118 | X | X |
| MunpJP075 | DSB_1459 | *Disconectes* | sp. 1 | Mu08 | BOLD:ADL5271 | 50524 | SMF 54363 | Munnopsidae | S | BC | 118 | X | X |
| MunpJP076 | DSB_1460 | *Eurycope* | sp. 5 | Mu23 | BOLD:ADL2919 | 50525 | SMF 54364 | Munnopsidae | S | BC | 118 | X | X |
| MunpJP077 | DSB_1461 | *Disconectes* | indet | Mu08 | BOLD:ADL5271 | 50526 | SMF 54365 | Munnopsidae | S | BC | 118 | X | X |
| MunpJP078 | DSB_1462 | *Munneurycope* | nodifrons | Mu67 | BOLD:ADL2777 | 50527 | SMF 54366 | Munnopsidae | S | BC | 118 | X | X |
| MunpJP079 | DSB_1463 | *Eurycope* | sp. 5 | Mu37 | BOLD:ADL2918 | 50528 | SMF 54367 | Munnopsidae | S | BC | 118 | X | X |
| MunpJP080 | DSB_1464 | *Eurycope* | sp. 5 | Mu37 | BOLD:ADL2918 | 50529 | SMF 54368 | Munnopsidae | S | BC | 118 | X | X |
| MunpJP081 | DSB_1465 | *Eurycope* | sp. 5 | Mu26 | BOLD:ADL3755 | 50530 | SMF 54369 | Munnopsidae | S | FC | 171 | X | X |
| MunpJP082 | DSB_1466 | *Betamorpha* | sp. 2 | Mu40 | BOLD:ADL4072 | 50531 | SMF 54370 | Munnopsidae | S | FC | 171 | X | X |
| MunpJP083 | DSB_1467 | *Eurycope* | sp. 5 | Mu24 | BOLD:ADL3752 | 50532 | SMF 54371 | Munnopsidae | S | FC | 171 | X | X |
| MunpJP084 | DSB_1468 | *Betamorpha* | sp. 2 | Mu40 | BOLD:ADL4072 | 50533 | SMF 54372 | Munnopsidae | S | FC | 171 | X | |





| | | | | | | | | | | | | | |
|---|---|---|---|---|---|---|---|---|---|---|---|---|---|
| MunpJP085 | DSB_1469 | *Eurycope* | sp. 2 | Mu18 | BOLD:ADL4333 | 50534 | SMF 54373 | Munnopsidae | S | FC | 171 | X | X |
| MunpJP086 | DSB_1470 | *Eurycope* | sp. 5 | Mu37 | BOLD:ADL3753 | 50535 | SMF 54374 | Munnopsidae | S | IOM | 99 | X | X |
| MunpJP088 | DSB_1472 | *Betamorpha* | sp. | Mu46 | BOLD:ADL4080 | 50537 | SMF 54375 | Munnopsidae | S | IOM | 99 | X | |
| MunpJP089 | DSB_1473 | *Disconectes* | sp. 1 | Mu04 | BOLD:ADL4015 | 50538 | SMF 54376 | Munnopsidae | S | FC | 158 | X | X |
| MunpJP090 | DSB_1474 | *Aspidarachna* | indet | Mu53 | BOLD:ADL5239 | 50539 | SMF 54377 | Munnopsidae | S | FC | 158 | X | X |
| MunpJP091 | DSB_1475 | *Disconectes* | sp. 3 | Mu01 | BOLD:ADL4465 | 50540 | SMF 54378 | Munnopsidae | S | FC | 158 | X | X |
| MunpJP092 | DSB_1476 | *Munneurycope* | gen. nov sp. 3 | Mu48 | BOLD:ADL4079 | 50541 | SMF 54379 | Munnopsidae | S | FC | 158 | X | X |
| MunpJP093 | DSB_1477 | *Eurycope* | sp. 5 | Mu29 | BOLD:ADL3751 | 50542 | SMF 54380 | Munnopsidae | S | GC | 20 | X | |
| MunpJP094 | DSB_1478 | *Eurycope* | sp. 5 | Mu32 | BOLD:ADL4077 | 50543 | SMF 54381 | Munnopsidae | S | GC | 20 | X | |
| MunpJP095 | DSB_1479 | *Acanthocope* | cf. galatheae | Mu56 | BOLD:ADL4687 | 50544 | SMF 54382 | Munnopsidae | S | A3 | 197 | X | X |
| MunpJP096 | DSB_1480 | *Munneurycope* | indet | Mu67 | BOLD:ADL2777 | 50545 | SMF 54383 | Munnopsidae | S | GC | 24 | X | X |
| MunpJP097 | DSB_1481 | *Munneurycope* | indet | Mu67 | BOLD:ADL2777 | 50546 | SMF 54384 | Munnopsidae | S | GC | 24 | X | X |
| MunpJP098 | DSB_1482 | *Munneurycope* | indet | Mu67 | BOLD:ADL2777 | 50547 | SMF 54385 | Munnopsidae | S | GC | 24 | X | X |
| MunpJP099 | DSB_1483 | *Eurycope* | sp. 5 | Mu37 | BOLD:ADL3657 | 50548 | SMF 54386 | Munnopsidae | S | GC | 24 | X | X |
| MunpJP100 | DSB_1484 | *Eurycope* | sp. 5 | Mu32 | BOLD:ADL4077 | 50549 | SMF 54387 | Munnopsidae | S | GC | 24 | X | X |
| MunpJP101 | DSB_1485 | *Eurycope* | sp. 5 | Mu25 | BOLD:ADG1444 | 50550 | SMF 54388 | Munnopsidae | S | GC | 24 | X | X |
| MunpJP104 | DSB_1488 | *Pirinectes* | gen. nov | Mu77 | BOLD:ADL5270 | 50553 | SMF 54389 | Munnopsidae | S | GC | 24 | X | X |
| MunpJP105 | DSB_1489 | *Pirinectes* | gen. nov | Mu77 | BOLD:ADL5270 | 50554 | SMF 54390 | Munnopsidae | S | GC | 24 | X | X |
| MunpJP106 | DSB_1490 | *Rectisura* | *slavai* | Mu60 | | 50555 | SMF 54391 | Munnopsidae | S | GC | 24 | | X |
| MunpJP107 | DSB_1491 | *Disconectes* | sp. 1 | Mu08 | | 50556 | SMF 54392 | Munnopsidae | S | GC | 24 | | X |
| MunpJP109 | DSB_1493 | *Eurycope* | sp. 5 | Mu32 | BOLD:ADL4077 | 50558 | SMF 54393 | Munnopsidae | S | GC | 24 | X | X |
| MunpJP110 | DSB_1494 | *Munneurycope* | indet | Mu81 | BOLD:ADL4576 | 50559 | SMF 54394 | Munnopsidae | S | GC | 24 | X | |
| MunpJP111 | DSB_1495 | *Rectisura* | *slavai* | Mu60 | BOLD:ADL4976 | 50560 | SMF 54395 | Munnopsidae | S | GC | 24 | X | X |
| MunpJP112 | DSB_1496 | *Munneurycope* | indet | Mu81 | BOLD:ADL4576 | 50561 | SMF 54396 | Munnopsidae | S | GC | 24 | X | X |
| MunpJP113 | DSB_1497 | *Betamorpha* | sp. | Mu70 | BOLD:ADL4078 | 50562 | SMF 54397 | Munnopsidae | S | GC | 24 | X | X |
| MunpJP114 | DSB_1498 | *Syneurycope* | sp. | Mu85 | | 50563 | SMF 54398 | Munnopsidae | S | FC | 171 | X | X |
| MunpJP115 | DSB_1499 | *Syneurycope* | sp. 1 | Mu85 | | 50564 | SMF 54399 | Munnopsidae | S | FC | 171 | X | X |
| MunpJP116 | DSB_1500 | *Syneurycope* | sp. 1 | Mu85 | | 50565 | SMF 54400 | Munnopsidae | S | FC | 171 | X | X |
| MunpJP117 | DSB_1501 | *Betamorpha* | sp. 1 | Mu42 | BOLD:ADL3458 | 50566 | SMF 54401 | Munnopsidae | S | FC | 171 | X | |
| MunpJP119 | DSB_1503 | *Pirinectes* | gen. nov | Mu78 | BOLD:ADL5438 | 50568 | SMF 54402 | Munnopsidae | S | FC | 171 | X | X |
| MunpJP120 | DSB_1504 | *Eurycope* | sp. 1 | Mu37 | BOLD:ADL2918 | 50569 | SMF 54403 | Munnopsidae | S | FC | 171 | X | X |
| MunpJP121 | DSB_1505 | *Syneurycope* | indet | Mu85 | | 50570 | SMF 54404 | Munnopsidae | S | FC | 171 | X | X |
| MunpJP122 | DSB_1506 | *Eurycope* | sp. 1 | Mu18 | BOLD:ADL4333 | 50571 | SMF 54405 | Munnopsidae | S | FC | 171 | X | X |
| MunpJP124 | DSB_1508 | *Eurycope* | sp. 5 | Mu24 | BOLD:ADL3752 | 50573 | SMF 54406 | Munnopsidae | S | FC | 171 | X | X |
| MunpJP125 | DSB_1509 | *Eurycope* | sp. 1 | Mu18 | BOLD:ADL4333 | 50574 | SMF 54407 | Munnopsidae | S | FC | 171 | X | X |
| MunpJP126 | DSB_1510 | *Eurycope* | sp. 1 | Mu18 | BOLD:ADL4333 | 50575 | SMF 54408 | Munnopsidae | S | FC | 171 | X | X |
| MunpJP128 | DSB_1512 | *Eurycope* | sp. 5 | Mu21 | BOLD:ADL2775 | 50577 | SMF 54409 | Munnopsidae | S | FC | 171 | X | X |
| MunpJP129 | DSB_1513 | *Eurycope* | sp. 2 | Mu19 | BOLD:ADL4334 | 50578 | SMF 54410 | Munnopsidae | S | FC | 171 | X | X |
| MunpJP130 | DSB_1514 | *Ilyarachna* | indet | Mu54 | BOLD:ADL5631 | 50579 | SMF 54411 | Munnopsidae | S | FC | 171 | X | X |
| MunpJP131 | DSB_1515 | *Disconectes* | sp. 1 | Mu03 | BOLD:ADL5437 | 50580 | SMF 54412 | Munnopsidae | S | FC | 171 | X | X |
| MunpJP133 | DSB_1517 | *Eurycope* | sp. 2 | Mu18 | BOLD:ADL4333 | 50582 | SMF 54413 | Munnopsidae | S | FC | 171 | X | X |
| MunpJP134 | DSB_1518 | *Microcope* | sp. | Mu83 | BOLD:ADL5442 | 50583 | SMF 54414 | Munnopsidae | S | FC | 171 | X | X |
| MunpJP135 | DSB_1519 | *Microcope* | sp. | Mu83 | BOLD:ADL5442 | 50584 | SMF 54415 | Munnopsidae | S | FC | 171 | X | X |
| MunpJP136 | DSB_1520 | *Syneurycope* | sp. | Mu85 | BOLD:ADM0298 | 50585 | SMF 54416 | Munnopsidae | S | FC | 171 | X | X |
| MunpJP137 | DSB_1521 | *Ilyarachna* | indet | Mu54 | BOLD:ADL5631 | 50586 | SMF 54417 | Munnopsidae | S | FC | 171 | X | X |
| MunpJP138 | DSB_1522 | *Rectisura* | slavai | Mu60 | | 50587 | SMF 54418 | Munnopsidae | S | IOM | 81 | X | |
| MunpJP139 | DSB_1523 | *Disconectes* | sp. 3 | Mu04 | BOLD:ADL5441 | 50588 | SMF 54419 | Munnopsidae | S | IOM | 81 | X | |
| MunpJP140 | DSB_1524 | *Disconectes* | sp. 3 | Mu05 | BOLD:ADL5440 | 50589 | SMF 54420 | Munnopsidae | S | IOM | 81 | X | |
| MunpJP141 | DSB_1525 | *Disconectes* | sp. 1 | Mu11 | BOLD:ADL5439 | 50590 | SMF 54421 | Munnopsidae | S | IOM | 81 | X | X |
| MunpJP142 | DSB_1526 | *Eurycope* | indet | Mu15 | BOLD:ADL2778 | 50591 | SMF 54422 | Munnopsidae | S | IOM | 81 | X | X |
| MunpJP143 | DSB_1527 | *Disconectes* | sp. 2 | Mu74 | BOLD:ADL5444 | 50592 | SMF 54423 | Munnopsidae | S | IOM | 99 | X | X |
| MunpJP144 | DSB_1528 | *Disconectes* | sp. 1 | Mu12 | BOLD:ADL5443 | 50593 | SMF 54424 | Munnopsidae | S | IOM | 99 | X | X |
| MunpJP145 | DSB_1529 | *Eurycope* | sp. 2 | Mu22 | BOLD:ADL9336 | 50594 | SMF 54425 | Munnopsidae | S | IOM | 99 | X | X |
| MunpJP146 | DSB_1530 | *Eurycope* | sp. 2 | Mu23 | | 50595 | SMF 54426 | Munnopsidae | S | IOM | 99 | | X |
| MunpJP147 | DSB_1531 | *Tytthocope* | indet | Mu49 | BOLD:ADL2887 | 50596 | SMF 54427 | Munnopsidae | S | IOM | 99 | X | |
| MunpJP148 | DSB_1532 | *Tytthocope* | indet | Mu49 | BOLD:ADL2887 | 50597 | SMF 54428 | Munnopsidae | S | IOM | 99 | X | |
| MunpJP149 | DSB_1533 | *Disconectes* | sp. 1 | Mu08 | BOLD:ADL5271 | 50598 | SMF 54429 | Munnopsidae | S | IOM | 99 | X | X |
| MunpJP150 | DSB_1534 | *Eurycope* | sp. 5 | Mu37 | | 50599 | SMF 54430 | Munnopsidae | S | IOM | 99 | X | X |
| MunpJP153 | DSB_1537 | *Eurycope* | sp. 4 | Mu15 | BOLD:ADL2778 | 50602 | SMF 54431 | Munnopsidae | S | IOM | 99 | X | X |
| MunpJP154 | DSB_1538 | *Disconectes* | sp. 2 | Mu71 | BOLD:ADL5605 | 50603 | SMF 54432 | Munnopsidae | S | IOM | 99 | | X |
| MunpJP155 | DSB_1539 | *Disconectes* | sp. 2 | Mu74 | BOLD:ADL5444 | 50604 | SMF 54433 | Munnopsidae | S | IOM | 99 | X | X |
| MunpJP156 | DSB_1540 | *Munneurycope* | sp. | Mu81 | BOLD:ADL3749 | 50605 | SMF 54434 | Munnopsidae | S | BC | 133 | X | X |
| MunpJP157 | DSB_1541 | *Disconectes* | sp. 3 | Mu04 | BOLD:ADL5441 | 50606 | SMF 54435 | Munnopsidae | S | BC | 133 | X | X |
| MunpJP158 | DSB_1542 | *Eurycope* | sp. | Mu23 | | 50607 | SMF 54436 | Munnopsidae | S | BC | 133 | | X |





| ID | DSB | Genus | Species | Mu | BOLD | No. | SMF | Family | | Station | Depth | | |
|---|---|---|---|---|---|---|---|---|---|---|---|---|---|
| MunpJP159 | DSB_1543 | *Rectisura* | *slavai* | Mu59 | BOLD:ADL5293 | 50608 | SMF 54437 | Munnopsidae | S | BC | 133 | X | X |
| MunpJP160 | DSB_1544 | *Eurycope* | *longiflagrata* | Mu25 | BOLD:ADL2917 | 50609 | SMF 54438 | Munnopsidae | S | BC | 133 | X | X |
| MunpJP161 | DSB_1545 | *Munneurycope* | indet | Mu67 | BOLD:ADL2777 | 50610 | SMF 54439 | Munnopsidae | S | BC | 133 | X | X |
| MunpJP162 | DSB_1546 | *Eurycope* | sp. 5 | Mu32 | BOLD:ADM2325 | 50611 | SMF 54440 | Munnopsidae | S | BC | 133 | X | X |
| MunpJP163 | DSB_1547 | *Eurycope* | sp. 5 | Mu37 | BOLD:ADL2918 | 50612 | SMF 54441 | Munnopsidae | S | BC | 133 | X | X |
| MunpJP164 | DSB_1548 | *Munneurycope* | indet | Mu81 | BOLD:ADL3750 | 50613 | SMF 54442 | Munnopsidae | S | BC | 133 | X | X |
| MunpJP165 | DSB_1549 | *Munneurycope* | indet | Mu67 | BOLD:ADL2777 | 50614 | SMF 54443 | Munnopsidae | S | BC | 133 | X | X |
| MunpJP166 | DSB_1550 | *Eurycope* | sp. 5 | Mu34 | BOLD:ADL4574 | 50615 | SMF 54444 | Munnopsidae | S | BC | 133 | X | X |
| MunpJP168 | DSB_1552 | *Disconectes* | sp. 3 | Mu04 | BOLD:ADL5606 | 50617 | SMF 54445 | Munnopsidae | S | BC | 133 | X | X |
| MunpJP169 | DSB_1553 | *Disconectes* | sp. 1 | Mu08 | | 50618 | SMF 54446 | Munnopsidae | S | BC | 133 | | X |
| MunpJP170 | DSB_1554 | *Disconectes* | sp. 3 | Mu05 | BOLD:ADL5440 | 50619 | SMF 54447 | Munnopsidae | S | BC | 133 | X | X |
| MunpJP171 | DSB_1555 | *Munneurycope* | indet | Mu81 | BOLD:ADL3750 | 50620 | SMF 54448 | Munnopsidae | S | BC | 133 | X | X |
| MunpJP172 | DSB_1556 | *Munneurycope* | indet | Mu81 | BOLD:ADL3750 | 50621 | SMF 54449 | Munnopsidae | S | BC | 133 | X | X |
| MunpJP173 | DSB_1557 | *Munneurycope* | indet | Mu68 | BOLD:ADL5567 | 50622 | SMF 54450 | Munnopsidae | S | FC | 158 | X | X |
| MunpJP174 | DSB_1558 | *Pirinectes* | gen. nov | Mu78 | BOLD:ADL5438 | 50623 | SMF 54451 | Munnopsidae | S | FC | 158 | X | X |
| MunpJP177 | DSB_1559 | *Disconectes* | sp. 1 | Mu13 | BOLD:ADL5603 | 50624 | SMF 54452 | Munnopsidae | S | FC | 158 | X | X |
| MunpJP175 | DSB_1560 | *Aspidarachna* | sp. | Mu53 | BOLD:ADL5239 | 50625 | SMF 54453 | Munnopsidae | S | FC | 158 | | X |
| MunpJP176 | DSB_1561 | *Disconectes* | sp. 3 | Mu01 | | 50626 | SMF 54454 | Munnopsidae | S | FC | 158 | | X |
| MunpJP178 | DSB_1562 | *Disconectes* | sp. 3 | Mu01 | BOLD:ADL5604 | 50627 | SMF 54455 | Munnopsidae | S | FC | 158 | X | X |
| MunpJP179 | DSB_1563 | *Disconectes* | sp. 1 | Mu11 | BOLD:ADL5439 | 50628 | SMF 54456 | Munnopsidae | S | FC | 158 | X | X |
| MunpJP183 | DSB_1567 | *Disconectes* | sp. 3 | Mu01 | | 50632 | SMF 54457 | Munnopsidae | S | FC | 158 | | X |
| MunpJP184 | DSB_1568 | *Disconectes* | sp. 1 | Mu11 | BOLD:ADL5439 | 50633 | SMF 54458 | Munnopsidae | S | FC | 158 | X | X |
| MunpJP185 | DSB_1569 | *Eurycope* | sp. 5 | Mu34 | BOLD:ADL4574 | 50634 | SMF 54459 | Munnopsidae | S | FC | 158 | X | X |
| MunpJP186 | DSB_1570 | *Eurycope* | sp. 1 | Mu19 | BOLD:ADL4334 | 50635 | SMF 54460 | Munnopsidae | S | FC | 158 | X | X |
| MunpJP187 | DSB_1571 | *Eurycope* | sp. 5 | Mu21 | BOLD:ADL2775 | 50636 | SMF 54461 | Munnopsidae | S | FC | 158 | X | |
| MunpJP188 | DSB_1572 | *Eurycope* | sp. 4 | Mu16 | BOLD:ADL4755 | 50637 | SMF 54462 | Munnopsidae | S | FC | 158 | X | X |
| MunpJP189 | DSB_1573 | *Eurycope* | sp. 3 | Mu16 | | 50638 | SMF 54463 | Munnopsidae | S | FC | 158 | | X |
| MunpJP197 | DSB_1580 | *Disconectes* | sp. 1 | Mu10 | BOLD:ADL5601 | 50645 | SMF 54464 | Munnopsidae | S | FC | 158 | X | X |
| MunpJP198 | DSB_1581 | *Disconectes* | sp. 3 | Mu04 | BOLD:ADL4015 | 50646 | SMF 54465 | Munnopsidae | S | FC | 158 | X | X |
| MunpJP199 | DSB_1582 | *Disconectes* | sp. 1 | Mu11 | BOLD:ADL4379 | 50647 | SMF 54466 | Munnopsidae | S | FC | 158 | X | |
| MunpJP200 | DSB_1583 | *Disconectes* | sp. 2 | Mu73 | BOLD:ADM0870 | 50648 | SMF 54467 | Munnopsidae | S | FC | 158 | X | X |
| MunpJP201 | DSB_1584 | *Disconectes* | sp. 3 | Mu01 | BOLD:ADL5602 | 50649 | SMF 54468 | Munnopsidae | S | FC | 158 | X | X |
| MunpJP203 | DSB_1586 | *Munneurycope* | *nodifrons* | Mu67 | BOLD:ADL2777 | 50651 | SMF 54469 | Munnopsidae | S | FC | 158 | X | X |
| MunpJP204 | DSB_1587 | *Munneurycope* | *nodifrons* | Mu67 | BOLD:ADL2777 | 50652 | SMF 54470 | Munnopsidae | S | BC | 118 | X | X |
| MunpJP205 | DSB_1588 | *Eurycope* | sp. 5 | Mu25 | BOLD:ADL2917 | 50653 | SMF 54471 | Munnopsidae | S | BC | 118 | X | X |
| MunpJP206 | DSB_1589 | *Munnopsurus* | indet | Mu51 | BOLD:ADL3118 | 50654 | SMF 54472 | Munnopsidae | S | BC | 118 | X | |
| MunpJP207 | DSB_1590 | *Disconectes* | sp. 3 | Mu04 | BOLD:ADL5441 | 50655 | SMF 54473 | Munnopsidae | S | BC | 118 | X | X |
| MunpJP208 | DSB_1591 | *Disconectes* | sp. 3 | Mu04 | BOLD:ADL5441 | 50656 | SMF 54474 | Munnopsidae | S | BC | 118 | X | X |
| MunpJP209 | DSB_1592 | *Rectisura* | *slavai* | Mu59 | BOLD:ADL5293 | 50657 | SMF 54475 | Munnopsidae | S | BC | 118 | X | X |
| MunpJP210 | DSB_1593 | *Paramunnopsis* | sp. 1 | Mu38 | BOLD:ADL3456 | 50658 | SMF 54476 | Munnopsidae | S | BC | 118 | X | X |
| MunpJP211 | DSB_1594 | *Eurycope* | sp. 5 | Mu25 | | 50659 | SMF 54477 | Munnopsidae | S | BC | 118 | | X |
| MunpJP212 | DSB_1595 | *Munneurycope* | indet | Mu68 | BOLD:ADL3937 | 50660 | SMF 54478 | Munnopsidae | S | BC | 118 | X | X |
| MunpJP213 | DSB_1596 | *Eurycope* | sp. 5 | Mu25 | BOLD:ADL2917 | 50661 | SMF 54479 | Munnopsidae | S | BC | 118 | X | X |
| MunpJP214 | DSB_1597 | *Eurycope* | sp. 5 | Mu37 | BOLD:ADL2918 | 50662 | SMF 54480 | Munnopsidae | S | BC | 118 | X | X |
| MunpJP215 | DSB_1598 | *Disconectes* | sp. 1 | Mu08 | BOLD:ADL5271 | 50663 | SMF 54481 | Munnopsidae | S | BC | 118 | X | X |
| MunpJP216 | DSB_1599 | *Disconectes* | sp. 1 | Mu08 | BOLD:ADL5271 | 50664 | SMF 54482 | Munnopsidae | S | BC | 118 | X | X |
| MunpJP218 | DSB_1601 | *Tytthocope* | sp. | Mu50 | BOLD:ADL4751 | 50666 | SMF 54483 | Munnopsidae | S | BC | 118 | X | X |
| MunpJP219 | DSB_1602 | *Eurycope* | sp. 5 | Mu37 | BOLD:ADL2918 | 50667 | SMF 54484 | Munnopsidae | S | BC | 118 | X | X |
| MunpJP220 | DSB_1603 | *Disconectes* | sp. 1 | Mu08 | BOLD:ADL5271 | 50668 | SMF 54485 | Munnopsidae | S | BC | 118 | X | X |
| MunpJP221 | DSB_1604 | *Eurycope* | sp. 5 | Mu23 | BOLD:ADL2919 | 50669 | SMF 54486 | Munnopsidae | S | GC | 20 | X | X |
| MunpJP222 | DSB_1605 | *Eurycope* | sp. 5 | Mu37 | BOLD:ADL3657 | 50670 | SMF 54487 | Munnopsidae | S | GC | 20 | X | X |
| MunpJP223 | DSB_1606 | *Disconectes* | sp. 1 | Mu08 | BOLD:ADL5271 | 50671 | SMF 54488 | Munnopsidae | S | GC | 20 | X | X |
| MunpJP224 | DSB_1607 | *Betamorpha* | sp. | Mu70 | BOLD:ADL4078 | 50672 | SMF 54489 | Munnopsidae | S | GC | 20 | X | X |
| MunpJP225 | DSB_1608 | *Betamorpha* | sp. | Mu70 | BOLD:ADL4078 | 50673 | SMF 54490 | Munnopsidae | S | GC | 20 | X | X |
| MunpJP226 | DSB_1609 | *Munneurycope* | sp. | Mu68 | BOLD:ADL2922 | 50674 | SMF 54491 | Munnopsidae | S | IOM | 81 | X | X |
| MunpJP227 | DSB_1610 | *Eurycope* | sp. 5 | Mu25 | BOLD:ADG1444 | 50675 | SMF 54492 | Munnopsidae | S | IOM | 81 | X | X |
| MunpJP228 | DSB_1611 | *Eurycope* | sp. 5 | Mu25 | BOLD:ADG1444 | 50676 | SMF 54493 | Munnopsidae | S | IOM | 81 | X | X |
| MunpJP229 | DSB_1612 | *Munneurycope* | cf. *nodifrons* | Mu64 | BOLD:ADL2920 | 50677 | SMF 54494 | Munnopsidae | S | IOM | 81 | X | X |
| MunpJP230 | DSB_1613 | *Munneurycope* | cf. *nodifrons* | Mu67 | BOLD:ADL2777 | 50678 | SMF 54495 | Munnopsidae | S | IOM | 81 | X | X |
| MunpJP231 | DSB_1614 | *Disconectes* | sp. 3 | Mu04 | BOLD:ADL5441 | 50679 | SMF 54496 | Munnopsidae | S | IOM | 81 | X | |
| MunpJP232 | DSB_1615 | *Disconectes* | sp. 1 | Mu05 | BOLD:ADL5440 | 50680 | SMF 54497 | Munnopsidae | S | IOM | 81 | X | X |
| MunpJP233 | DSB_1616 | *Disconectes* | sp. 1 | Mu05 | BOLD:ADL5440 | 50681 | SMF 54498 | Munnopsidae | S | IOM | 81 | X | X |
| MunpJP234 | DSB_1617 | *Disconectes* | sp. 3 | Mu01 | BOLD:ADL2645 | 50682 | SMF 54499 | Munnopsidae | S | IOM | 81 | X | X |
| MunpJP235 | DSB_1618 | *Disconectes* | sp. 1 | Mu08 | BOLD:ADL5271 | 50683 | SMF 54500 | Munnopsidae | S | IOM | 81 | X | X |





| | | | | | | | | | | | | |
|---|---|---|---|---|---|---|---|---|---|---|---|---|
| MunpJP236 | DSB_1619 | *Disconectes* | sp. 1 | Mu09 | BOLD:ADL5599 | 50684 SMF 54501 | Munnopsidae | S | IOM | 81 | X | X |
| MunpJP237 | DSB_1620 | *Disconectes* | sp. 3 | Mu05 | BOLD:ADL5440 | 50685 SMF 54502 | Munnopsidae | S | IOM | 81 | X | X |
| MunpJP238 | DSB_1621 | *Rectisura* | *slavai* | Mu60 | BOLD:ADL4976 | 50686 SMF 54503 | Munnopsidae | S | IOM | 81 | X | X |
| MunpJP239 | DSB_1622 | *Eurycope* | sp | Mu37 | BOLD:ADL2921 | 50687 SMF 54504 | Munnopsidae | S | IOM | 81 | X | X |
| MunpJP240 | DSB_1623 | *Rectisura* | *slavai* | Mu60 | BOLD:ADL4976 | 50688 SMF 54505 | Munnopsidae | S | IOM | 81 | X | X |
| MunpJP241 | DSB_1624 | *Paramunnopsis* | sp. 2 | Mu39 | BOLD:ADL3685 | 50689 SMF 54506 | Munnopsidae | S | IOM | 81 | X | X |
| MunpJP242 | DSB_1625 | *Munneurycope* | indet | Mu68 | BOLD:ADL4075 | 50690 SMF 54507 | Munnopsidae | S | A3 | 210 | X | X |
| MunpJP243 | DSB_1626 | *Betamorpha* | sp. | Mu69 | BOLD:ADL4076 | 50691 SMF 54508 | Munnopsidae | S | A3 | 210 | | X |
| MunpJP244 | DSB_1627 | *Betamorpha* | sp. | Mu69 | BOLD:ADL4076 | 50692 SMF 54509 | Munnopsidae | S | A3 | 210 | | X |
| MunpJP245 | DSB_1628 | *Disconectes* | sp. 1 | Mu09 | BOLD:ADL5600 | 50693 SMF 54510 | Munnopsidae | S | A3 | 210 | X | X |
| MunpJP246 | DSB_1629 | *Rectisura* | sp. | Mu61 | BOLD:ADL3110 | 50694 SMF 54511 | Munnopsidae | S | A3 | 210 | X | X |
| MunpJP247 | DSB_1630 | *Bellibos* | sp. 1 | Mu44 | BOLD:ADL4771 | 50695 SMF 54512 | Munnopsidae | S | A3 | 210 | X | |
| MunpJP248 | DSB_1631 | *Rectisura* | *slavai* | Mu58 | BOLD:ADL4772 | 50696 SMF 54513 | Munnopsidae | S | A3 | 210 | X | X |
| MunpJP249 | DSB_1632 | *Disconectes* | sp. 3 | Mu06 | BOLD:ADL5597 | 50697 SMF 54514 | Munnopsidae | S | A3 | 210 | X | X |
| MunpJP250 | DSB_1633 | *Disconectes* | sp. 3 | Mu03 | BOLD:ADL5598 | 50698 SMF 54515 | Munnopsidae | S | A3 | 210 | X | |
| MunpJP252 | DSB_1635 | *Bellibos* | sp. 1 | Mu44 | BOLD:ADL4771 | 50700 SMF 54516 | Munnopsidae | S | A3 | 210 | X | |
| MunpJP253 | DSB_1636 | *Munneurycope* | sp. | Mu68 | BOLD:ADL4075 | 50701 SMF 54517 | Munnopsidae | S | A3 | 210 | X | X |
| MunpJP254 | DSB_1637 | *Disconectes* | sp. 3 | Mu03 | BOLD:ADL5598 | 50702 SMF 54518 | Munnopsidae | S | A3 | 210 | X | X |
| MunpJP256 | DSB_1639 | *Eurycope* | sp. 5 | Mu30 | BOLD:ADL4756 | 50704 SMF 54519 | Munnopsidae | S | A3 | 210 | X | |
| MunpJP257 | DSB_1640 | *Eurycope* | sp. 5 | Mu33 | BOLD:ADL2967 | 50705 SMF 54520 | Munnopsidae | S | A3 | 210 | X | |
| MunpJP258 | DSB_1641 | *Eurycope* | sp. 5 | Mu33 | BOLD:ADL2967 | 50706 SMF 54521 | Munnopsidae | S | A3 | 210 | X | X |
| MunpJP259 | DSB_1642 | *Eurycope* | sp. 5 | Mu33 | BOLD:ADL2967 | 50707 SMF 54522 | Munnopsidae | S | A3 | 210 | X | X |
| MunpJP260 | DSB_1643 | *Ilyarachna* | indet | Mu55 | BOLD:ADL2886 | 50708 SMF 54523 | Munnopsidae | S | A3 | 197 | X | X |
| MunpJP261 | DSB_1644 | *Rectisura* | *slavai* | Mu58 | BOLD:ADL4772 | 50709 SMF 54524 | Munnopsidae | S | A3 | 197 | X | X |
| MunpJP262 | DSB_1645 | *Disconectes* | sp. 3 | Mu01 | | 50710 SMF 54525 | Munnopsidae | S | A3 | 197 | | X |
| MunpJP263 | DSB_1646 | *Rectisura* | *slavai* | Mu58 | BOLD:ADL4772 | 50711 SMF 54526 | Munnopsidae | S | A3 | 197 | X | X |
| MunpJP264 | DSB_1647 | *Eurycope* | sp. 5 | Mu31 | BOLD:ADL2968 | 50712 SMF 54527 | Munnopsidae | S | A3 | 197 | X | X |
| MunpJP265 | DSB_1648 | *Munneurycope* | indet | Mu65 | BOLD:ADL4773 | 50713 SMF 54528 | Munnopsidae | S | A3 | 197 | X | X |
| MunpJP266 | DSB_1649 | *Acanthocope* | cf. *galatheae* | Mu56 | BOLD:ADL4687 | 50714 SMF 54529 | Munnopsidae | S | A3 | 197 | X | X |
| MunpJP267 | DSB_1650 | *Acanthocope* | cf. *galatheae* | Mu56 | BOLD:ADL4687 | 50715 SMF 54530 | Munnopsidae | S | A3 | 197 | X | X |
| MunpJP268 | DSB_1651 | *Betamorpha* | indet | Mu41 | BOLD:ADL5566 | 50716 SMF 54531 | Munnopsidae | S | A3 | 197 | X | X |
| MunpJP269 | DSB_1652 | *Disconectes* | sp. 3 | Mu03 | BOLD:ADL2644 | 50717 SMF 54532 | Munnopsidae | S | A3 | 197 | X | |
| MunpJP270 | DSB_1653 | *Munneurycope* | indet | Mu66 | BOLD:ADL4774 | 50718 SMF 54533 | Munnopsidae | S | A3 | 197 | X | X |
| MunpJP272 | DSB_1655 | *Eurycope* | sp. 5 | Mu33 | BOLD:ADL2967 | 50720 SMF 54534 | Munnopsidae | S | A3 | 197 | X | X |
| MunpJP274 | DSB_1657 | *Acanthocope* | cf. *galatheae* | Mu56 | BOLD:ADL4687 | 50722 SMF 54535 | Munnopsidae | S | A3 | 197 | X | X |
| MunpJP276 | DSB_1659 | *Bellibos* | sp. 2 | Mu43 | BOLD:ADL3457 | 50724 SMF 54536 | Munnopsidae | S | A3 | 192 | X | X |
| MunpJP277 | DSB_1660 | *Betamorpha* | sp. | Mu69 | | 50725 SMF 54537 | Munnopsidae | S | A3 | 192 | | X |
| MunpJP278 | DSB_1661 | *Bellibos* | sp. 2 | Mu43 | BOLD:ADL3457 | 50726 SMF 54538 | Munnopsidae | S | A3 | 192 | X | X |
| MunpJP279 | DSB_1662 | *Disconectes* | sp. 3 | Mu01 | BOLD:ADL9613 | 50727 SMF 54539 | Munnopsidae | S | A3 | 192 | X | X |
| MunpJP280 | DSB_1663 | *Disconectes* | sp. 1 | Mu09 | BOLD:ADL2643 | 50728 SMF 54540 | Munnopsidae | S | A3 | 192 | X | X |
| MunpJP281 | DSB_1664 | *Microcope* | sp. | Mu82 | BOLD:ADM0130 | 50729 SMF 54541 | Munnopsidae | S | A3 | 192 | X | |
| MunpJP282 | DSB_1665 | *Munneurycope sp.6* | gen. nov sp. 1 | Mu52 | BOLD:ADL4064 | 50730 SMF 54542 | Munnopsidae | S | A3 | 192 | X | |
| MunpJP283 | DSB_1666 | *Munneurycope* | indet | Mu65 | BOLD:ADL4773 | 50731 SMF 54543 | Munnopsidae | S | A3 | 192 | X | X |
| MunpJP284 | DSB_1667 | *Eurycope* | sp. 5 | Mu36 | BOLD:ADL2969 | 50732 SMF 54544 | Munnopsidae | S | A3 | 192 | X | X |
| MunpJP285 | DSB_1761 | *Munneurycope sp.6* | gen. nov sp. 1 | Mu52 | BOLD:ADL4064 | 50826 SMF 54545 | Munnopsidae | S | A3 | 192 | X | |
| MunpJP286 | DSB_2011 | *Eurycope* | sp | Mu68 | BOLD:ADL2922 | 51076 SMF 54546 | Munnopsidae | S | IOM | 82 | X | X |
| MunpJP288 | DSB_2013 | *Betamorpha* | cf. *fusiformis* | Mu69 | BOLD:ADL4076 | 51078 SMF 54547 | Munnopsidae | S | A3 | 210 | X | X |





**Table 2 Summary of diversity parameters per sampled area. Sites = number of Epibenthic Sledge deployments, N = number of specimens, S = number of Species, Usp = number of unique species, Chao±SE = Chao estimated number of species with standard error, ACE±SE = ACE estimated number of species with standard error, H'= Shannon Diversity, 1-D= Simpson Diversity and J = Jaccard's Evenness. βSOR, βSIM and βSNE express multiple-site total beta diversity, multiple-site species turnover and multiple-site nestedness respectively. Note that in the rows of each area the beta-diversity values are the result of excluding this area, except for the row Total (which includes all areas) and CCZ only (which includes all but APEI3 and DISCOL.**

| AREA | Sites | N | S | Usp (%) | Chao±SE | ACE±SE | H' | 1-D | J | $\beta_{SOR}$ | $\beta_{SIM}$ | $\beta_{SNE}$ |
|---|---|---|---|---|---|---|---|---|---|---|---|---|
| APEI3 | 3 | 110 | 44 | 14 (32%) | 59.3±9.5 | 63.7±4.4 | 3.52 | 0.96 | 0.93 | 0.845 | 0.833 | 0.011 |
| GSR | 2 | 90 | 38 | 18 (47%) | 80±25.9 | 69.2±5 | 3.34 | 0.95 | 0.91 | 0.900 | 0.894 | 0.005 |
| DISCOL | 8 | 84 | 41 | 31 (76%) | 62.1±12.6 | 59.5±3.8 | 3.53 | 0.96 | 0.95 | 0.845 | 0.833 | 0.011 |
| IFREMER | 2 | 106 | 50 | 34 (68%) | 64±7.7 | 80.3±5.9 | 3.66 | 0.96 | 0.93 | 0.873 | 0.868 | 0.004 |
| BGR | 4 | 163 | 43 | 11 (25%) | 49.5±4.8 | 53.2±3.2 | 3.34 | 0.94 | 0.88 | 0.892 | 0.884 | 0.007 |
| IOM | 3 | 66 | 40 | 36 (90%) | 63±12.4 | 77±5.4 | 3.51 | 0.96 | 0.95 | 0.897 | 0.890 | 0.007 |
| CCZ only | 11 | 425 | 117 | 99 (84%) | 146 | 137 | 4.21 | 0.97 | 0.80 | 0.767 | 0.746 | 0.021 |
| Total | 22 | 619 | 187 | - | 235.46 | 252.15 | 4.76 | 0.98 | 0.91 | 0.885 | 0.878 | 0.007 |

**Table 3 Faunistic similarity between areas. Upper diagonal = number of shared species, lower diagonal = number of not shared species.**

| Not shared\shared | APEI3 | GSR | DISCOL | IFREMER | BGR | IOM |
|---|---|---|---|---|---|---|
| APEI3 | 0\44 | 2 | 1 | 6 | 1 | 5 |
| GSR | 78 | 0\38 | 4 | 11 | 16 | 16 |
| DISCOL | 83 | 71 | 0\41 | 2 | 5 | 2 |
| IFREMER | 82 | 66 | 87 | 0\50 | 8 | 12 |
| BGR | 85 | 49 | 74 | 77 | 0\43 | 15 |
| IOM | 74 | 46 | 77 | 66 | 53 | 0\40 |

**Table 4 Beta-diversity decomposition of Isopod Families. N = number of specimens, S = number of Species. βSOR, βSIM and βSNE express multiple-site total beta diversity, multiple-site species turnover and multiple-site nestedness respectively. Columns ccz N, ccz S, ccz βSOR, ccz βSIM and ccz βSNE consider only samples taken within the CCZ (excluding APEI3 and DISCOL)**

| | N | S | Chao±SE | $\beta_{SOR}$ | $\beta_{SIM}$ | $\beta_{SNE}$ | ccz N | ccz S | ccz $\beta_{SOR}$ | ccz $\beta_{SIM}$ | ccz $\beta_{SNE}$ |
|---|---|---|---|---|---|---|---|---|---|---|---|
| Munnopsidae | 294 | 91 | 110±8.9 | 0.873 | 0.860 | 0.013 | 199 | 55 | 0.743 | 0.704 | 0.039 |
| Desmosomatidae | 143 | 63 | 98.7±17 | 0.904 | 0.895 | 0.009 | 103 | 43 | 0.817 | 0.802 | 0.014 |
| Haploniscidae | 88 | 24 | 24.6±1.1 | 0.916 | 0.898 | 0.0183 | 53 | 14 | 0.803 | 0.739 | 0.067 |
| Macrostylidae | 94 | 9 | 9.0±0.2 | 0.809 | 0.777 | 0.031 | 70 | 5 | 0.583 | 0.500 | 0.083 |
| Total | 619 | 187 | 235.46 | 0.885 | 0.878 | 0.007 | 425 | 117 | 0.767 | 0.746 | 0.021 |

**Table 5 Number and percentage of species of the studied families present in only 1 to 6 areas. Total considers all areas together.**

| | 1 area | 2 areas | 3 areas | 4 areas | 5 areas | 6 areas | Total |
|---|---|---|---|---|---|---|---|
| Munnopsidae | 69 (75.8%) | 11 (12%) | 7 (7.6%) | 3 (3.2%) | 1 (1%) | 0 (0%) | 91 (48.6%) |
| Desmosomatidae | 49 (77.7%) | 10 (15.8%) | 3 (4.7%) | 1 (1.5%) | 0 (0%) | 0 (0%) | 63 (33.6%) |
| Haploniscidae | 20 (83 %) | 3 (12.5%) | 0 (0%) | 1 (4.1%) | 0 (0%) | 0 (0%) | 24 (12.8%) |
| Macrostylidae | 6 (66.6%) | 2 (22.2%) | 0 (0%) | 0 (0%) | 1 (11.1%) | 0 (0%) | 9 (4.8%) |
| Total | 144 (77%) | 26 (13.9%) | 10 (5.3%) | 5 (2.6%) | 2 (1%) | 0 (0%) | 187 |