# Peer review of "Adult life strategy affects distribution patterns in abyssal isopods – implications for conservation in Pacific nodule areas"

_Biogeosciences, 2019_

## Referee Comment (RC1) · Anonymous Referee #1 · 12 Nov 2019

General comments: This manuscript discusses an important problem in deep-sea benthic science: how widely are the species spread – and not least: how can we know that we have sampled enough to cover a species? By using isopod-data from the CCZ (and the DISCOL-area as a "control") and combining it with knowledge about the different life (and distribution) strategies for various taxa the authors present their suggestion for a method. The manuscript is generally well written (though sometimes it is a bit "massive") and is based on a large and sound dataset.

Specific comments: I think the manuscript starts out really well, and all analyses are sound and well reasoned. The discussion is also interesting (though sometimes you

bring in a plethora of ideas and thoughts at the same time, so even when it is structured, it feels a bit unstructured), and then the conclusion is about something slightly off (but still important) than what you start out with. You start out with wanting to examine the distribution patterns based on life strategy (as your title also claims), and you conclude with the problem we face when we need to delimit between species, and there is an astonishing (impressive!) high number of species possibly new to science in your dataset. I do understand that it is difficult to answer the original question precisely when you work with such a high percentage of "unknowns" – and I appreciate your efforts to make a try anyway. But maybe you need to include this question (about the taxonomic impediment) in your scope? Or discuss it a bit before?

Technical correctios: I attach an annotated version of your manuscript where I have suggested some corrections and pointed out a few minor errors. In general, I think there is a lot of information being "bombarded" on the reader, and thus the manuscript becomes heavy and a bit difficult to read. I understand that this is necessary if you want to discuss all the things you discuss, and it is also the style of the journal? But helping the readers follow your data and following arguments makes for better understanding. . .

Please also note the supplement to this comment:
https://www.biogeosciences-discuss.net/bg-2019-358/bg-2019-358-RC1-supplement.pdf

---

## Referee Comment (RC2) · Anonymous Referee #2 · 28 Mar 2020

Review for bg-2019-358

This paper represents an extremely important dataset and evaluation given our lack of understanding on abyssal macrofauna, deep-sea isopods, and future impacts of deep-sea mining. Concepts and methods aren't exactly new, but the data are and genetic vs. morphological comparisons are extremely important. The approach and methods are valid, and there are many good comparisons to other studies. However, the quality of presentation is lacking. The focus of different sections don't align with the title of the paper. There is a lot of unnecessary information, and not a clear structure. I suggest major revisions to the paper, reorganizing the introduction, results, and discussion to tell a clear story of isopod lifestyle and taxonomic ranges. The discussion and conclusion needs special attention as it is extremely difficult to read through.

General:

Misuse of commas throughout. Also other general problems with word usage such as adding/excluding s, verb usage, etc. Suggest having an native English speaker read through.

Abstract:

General: This abstract includes a lot of rather detailed results. I would suggest talking somewhat more broadly about adult life strategy in the distribution sentences, as your title states, rather than just copying results. Also there is no mention of implications for conservation in the abstract which is the other part of your title.

Lines 23 – 24: "The proportion of species present in a single area increased in this sequence…" I do not understand what this sentence is trying to say.

Lines 27 – 28: "The CCZ areas show the highest number of shared species" Compared to what?

Introduction:

General: Your title states that isopods will be the focus of your study, but they do not come up in the introduction for several paragraphs and are only briefly mentioned. I think your second or third paragraph (at a minimum, although it makes sense to open with this as it is your title) needs to macrofauna, their use in conservation, and a little about isopods in particular.

General: The introduction is rather piecemeal and a bit hard to follow as it is now. It could be a little more concise and needs to be reordered, focusing on macrofauna/isopods, deep-sea mining, lifestyle, and conservation.

General: There is no mention of APEIs in the introduction. I would argue that one of the most important things in understanding species distributions is making sure that APEIs are protecting the same species that are being destroyed in contractor areas.

Line 37: change "becoming" to "become"

Line 37: Jones et al., 2017 discusses impact studies. I do not think it is an appropriate citation economic interests or advancing technology.

Lines 44 – 45: Why will the ecological footprint determine whether mining operations will be feasible long-term? Mining operations on land are extremely destructive but still take place. You may need a sentence or two here discussing the ISA's role as regulator and their duel mission to encourage mining and protect the environment.

Line 46: Recolonization kind of comes out of nowhere here. An introductory sentence along the lines of "As mining will completely destroy communities along large swathes of the seafloor, recovery will only take place through recolonization from surrounding areas." Or something like that.

Lines 46 – 50: Geographic distributions also greatly affect the likelihood of species extinctions, which are also important for conservation.

Line 61: Change to "reproductive strategies. These strategies"

Line 63: delete "with"

Lines 73 – 74: This sentence seems very cherry-picked. A superfamily is the most numerous and diverse crustacean taxon? Tanaids are generally more abundant than isopods. Seems like a very broad statement. Suggest changing to something like "Isopoda is generally one of the dominant taxa in abyssal benthic samples." Or something more general like this.

Line 80: delete "still"

Line 95: "This information would be essential for conservation planning." How? This really needs to be fleshed out as it is part of your title.

Materials and Methods:

Lines 105 – 131: This section is sort of all over the place. Sample collection and processing I get, but the first paragraph also includes a lot of information on databases while the second paragraph discusses outgroups. Suggest breaking up the first paragraph into two, and moving the outgroups section to molecular methods.

Lines 106 – 108: You need a map to cite here.

Line 108: APEI's kind of come out of nowhere here. As you talk about implications for conservation in the introduction, you should mention APEI's and the importance of making sure they are representative of the contractor areas. In theory, this is where recolonization may come from.

Line 117: Shouldn't it be "DISCOL"

Line 118: Do you have a DOI for the dataset?

Line 123:  Delete "and specific"

Results:

General:  The results are rather disorganized.  It needs to flow more and have less repeats.  It is also strange to have subheadings for a sentence or two.  Suggest keeping section "Diversity by area" and having other sections such as "Shared species/similarity among areas", "Family/species ranges", "Beta diversity", "Molecular data".  Your current molecular section includes shared species and diversity components as well.  Not sure if the best way, but could also include a molecular and morphological component to each section instead of having a specific molecular section.

General:  Suggest discussing families in order of expected dispersal potential in each section and stating this at the start.  The different lifestyles are pretty much lost in the results.

Line 207: "clades with fewer samples" What exactly does this mean?  Fewer than what?

Line 237 – 238:  This is a repeat of above, don't include in both places.

Line 286:  Not exactly sure what "species abundance diversity" means.

Discussion:

General:  Like above sections, the discussion is not clearly organized, making it hard to follow.  Subsections would help.

General:  For discussion points, start the paragraph with the main point and how your data support/don't support this point.  Then go into what other studies found.

Line 307:  I have no clue what this first sentence means.

Lines 307 – 309:  Not really sure the point of this first little paragraph.  If your paper is attempting to establish a method for defining species, you need to talk about it throughout the paper, not just introduce it in the discussion.

Line 314: Need citation in place of "(citation here)"

Lines 311 – 317: This is a repeat of the introduction.  Instead cite some studies that talk about lifestyles and dispersal of isopods.

Lines 311 – 338:  You need to incorporate data from this study and previous studies more cohesively.  May be easier to read if you have a paragraph for each of the families, from highest to lowest dispersal capabilities, starting each paragraph talking about what your data say about their dispersal and comparing that to their lifestyles.

Line 320: another "(citation)"

Lines 339 – 344: Interesting point, but doesn't really fit in with what you are talking about. Maybe have a section labelled "Taxonomy" or something to that effect with bits like this?

Line 345 – 349: Delete this section. Not sure why you would need to remark that a study in one specific area is not representative of global diversity.

Lines 350 – 364: The first sentence is kind of contradictory to all the things you say about isopod lifestyles earlier. Could include some of this in the taxonomy section mentioned above, but seems like unnecessary information except for maybe a sentence or two about sexual dimorphism.

Line 365 – 369: Interesting information, but is not really tied in with your work at all as it is now.

Lines 370 – 383: Start with your data first in a paragraph, then discuss others. Also not sure what lines 370 – 373 are trying to say.

Line 388: "more likely" what?

Lines 401 – 407: suggest deleting, what does shallow vs. deep comparison have to do with this study?

Lines 407 – 414: Again suggest deleting. This is really muddying up the story you are trying to tell.

Lines 427 – 431: This doesn't flow at all. Maybe have a section on each family like suggested for results?

Lines 438 – 446: Why are you ending with species distinctions and not even those in your data? May be good to have a discussion section on taxonomy like suggested for results.

End: You don't discussion implications for conservation really at all here. Is APEI3 good? Are many of these species at risk of extinction because of singletons? Does genetic differentiation vs. morphology tell different stories for regulators? Could you focus future work on the least highly-dispersed family as they are likely to be most impacted by mining? Etc. You need a paragraph or two at the end really tying this all together and telling people why it is important work.

Conclusions:

General: Conclusions are almost all other people's work, and much of it is taxonomic problems which don't seem to be the focus of this paper. What are the main points of this study (e.g., isopod lifestyle, CCZ similarity, etc.)

Lines 448 – 450: Now I am confused. This study is focused on taxonomic incompleteness? I thought is was focused on isopod lifestyle and ranges. You need to have the conclusion talk about the main points you are trying to make, and taxonomy isn't even in the title.

Lines 448 – 458:  This is way too much of other people's work for a conclusion in this paper. You can have a sentence or two about the need for more molecular work and morphological problems, but it needs to be shorter and not at the start, unless you want to change the focus of this paper.

Line 459:  How do you know distance and locomotion are "most" important?  You tested everything?  What else did you examine besides distance and locomotion?

Line 460 – 461:  "Long-distance populations… patchy/local populations" What does this mean exactly?

Lines 460 – 468:  Again, your conclusion is almost all other people's work.  Need conclusions for this study.

Tables and Figures:

Table 1 should be supplementary.

---

## Author Comment (AC1) · 1 Sep 2020

The referee comment and the answers are listed according to the manuscript structure: Introduction: General: Your title states that isopods will be the focus of your study, but they do not come up in the introduction for several paragraphs and are only briefly mentioned. I think your second or third paragraph (at a minimum, although it makes sense to open with this as it is your title) needs to macrofauna, their use in conservation, and a little about isopods in particular. REPLY: we moved the paragraph on isopods further up, and added a couple of lines on their use(fulness) in conservational studies

General: The introduction is rather piecemeal and a bit hard to follow as it is now.

[Figure]
It could be a little more concise and needs to be reordered, focusing on macro-fauna/isopods, deep-sea mining, lifestyle, and conservation. REPLY: we removed two paragraphs from the discussion to make it more concise

General: There is no mention of APEIs in the introduction. I would argue that one of the most important things in understanding species distributions is making sure that APEIs are protecting the same species that are being destroyed in contractor areas. REPLY: we now introduced the concept of APEIs in the introduction

Line 37: change "becoming" to "become" Line 37: Jones et al., 2017 discusses impact studies. I do not think it is an appropriate citation economic interests or advancing technology. REPLY: Done

Lines 44 – 45: Why will the ecological footprint determine whether mining operations will be feasible long-term? Mining operations on land are extremely destructive but still take place. You may need a sentence or two here discussing the ISA's role as regulator and their duel mission to encourage mining and protect the environment. REPLY: we agree with the reviewer and deleted the lineages

Line 46: Recolonization kind of comes out of nowhere here. An introductory sentence along the lines of "As mining will completely destroy communities along large swathes of the seafloor, recovery will only take place through recolonization from surrounding areas." Or something like that. REPLY: We added the line according to the reviewer's suggestion

Lines 46 – 50: Geographic distributions also greatly affect the likelihood of species extinctions, which are also important for conservation. REPLY: We changed the line to "That is, species with a broader distribution and better dispersal ability likely have a greater potential to recolonize impacted areas compared to species with a narrow geographic area that likely have an increased risk of extinction following localized impacts (Roberts & Hawkins 1999)."

[Figure]

Line 61: Change to "reproductive strategies. These strategies" REPLY: Changed to "reproductive strategies. The latter are"

Line 63: delete "with" REPLY: Done

Lines 73 – 74: This sentence seems very cherry-picked. A superfamily is the most numerous and diverse crustacean taxon? Tanaids are generally more abundant than isopods. Seems like a very broad statement. Suggest changing to something like "Isopoda is generally one of the dominant taxa in abyssal benthic samples." Or something more general like this. REPLY: Done

Line 80: delete "still" REPLY: Done

Line 95: "This information would be essential for conservation planning." How? This really needs to be fleshed out as it is part of your title. REPLY: Done

Materials and Methods: Lines 105 – 131: This section is sort of all over the place. Sample collection and processing I get, but the first paragraph also includes a lot of information on databases while the second paragraph discusses outgroups. Suggest breaking up the first paragraph into two, and moving the outgroups section to molecular methods. REPLY: The text has been rewritten reflecting this referee comment.

Lines 106 – 108: You need a map to cite here. REPLY: The text has been rewritten reflecting this referee comment.

Line 108: APEI's kind of come out of nowhere here. As you talk about implications for conservation in the introduction, you should mention APEI's and the importance of making sure they are representative of the contractor areas. In theory, this is where recolonization may come from. REPLY: The text has been rewritten reflecting this referee comment.

Line 117: Shouldn't it be "DISCOL" REPLY: Yes.

Line 118: Do you have a DOI for the dataset? REPLY: Not at present. We are in

contact with BoLD.

Line 123: Delete "and specific" Results: General: The results are rather disorganized. It needs to flow more and have less repeats. It is also strange to have subheadings for a sentence or two. Suggest keeping section "Diversity by area" and having other sections such as "Shared species/similarity among areas", "Family/species ranges", "Beta diversity", "Molecular data". Your current molecular section includes shared species and diversity components as well. Not sure if the best way, but could also include a molecular and morphological component to each section instead of having a specific molecular section. General: Suggest discussing families in order of expected dispersal potential in each section and stating this at the start. The different lifestyles are pretty much lost in the results. REPLY: The text has been rewritten reflecting this referee comment.

Line 207: "clades with fewer samples" What exactly does this mean? Fewer than what? REPLY: The text has been rewritten reflecting this referee comment.

Line 237 – 238: This is a repeat of above, don't include in both places. REPLY: The text has been rewritten reflecting this referee comment.

Line 286: Not exactly sure what "species abundance diversity" means. Discussion: General: Like above sections, the discussion is not clearly organized, making it hard to follow. Subsections would help. General: For discussion points, start the paragraph with the main point and how your data support/don't support this point. Then go into what other studies found. REPLY: The text has been rewritten reflecting this referee comment.

Line 307: I have no clue what this first sentence means. REPLY: The text has been rewritten reflecting this referee comment.

Lines 307 – 309: Not really sure the point of this first little paragraph. If your paper is attempting to establish a method for defining species, you need to talk about it throughout

the paper, not just introduce it in the discussion. REPLY: The text has been rewritten reflecting this referee comment.

Line 314: Need citation in place of "(citation here)" REPLY: given.

Lines 311 – 317: This is a repeat of the introduction. Instead cite some studies that talk about lifestyles and dispersal of isopods. REPLY: The text has been rewritten reflecting this referee comment.

Lines 311 – 338: You need to incorporate data from this study and previous studies more cohesively. May be easier to read if you have a paragraph for each of the families, from highest to lowest dispersal capabilities, starting each paragraph talking about what your data say about their dispersal and comparing that to their lifestyles. Line 320: another "(citation)" REPLY: The text has been rewritten reflecting this referee comment.

Lines 339 – 344: Interesting point, but doesn't really fit in with what you are talking about. Maybe have a section labelled "Taxonomy" or something to that effect with bits like this? REPLY: The text has been rewritten reflecting this referee comment.

Line 345 – 349: Delete this section. Not sure why you would need to remark that a study in one specific area is not representative of global diversity. REPLY: The text has been rewritten reflecting this referee comment.

Lines 350 – 364: The first sentence is kind of contradictory to all the things you say about isopod lifestyles earlier. Could include some of this in the taxonomy section mentioned above, but seems like unnecessary information except for maybe a sentence or two about sexual dimorphism. REPLY: The text has been rewritten reflecting this referee comment.

Line 365 – 369: Interesting information, but is not really tied in with your work at all as it is now. REPLY: The text has been rewritten reflecting this referee comment.

Lines 370 – 383: Start with your data first in a paragraph, then discuss others. Also

not sure what REPLY: The text has been rewritten reflecting this referee comment.

lines 370 – 373 are trying to say. REPLY: The text has been rewritten reflecting this referee comment.

Line 388: "more likely" what? REPLY: The text has been rewritten reflecting this referee comment.

Lines 401 – 407: suggest deleting, what does shallow vs. deep comparison have to do with this study? REPLY: The text has been rewritten reflecting this referee comment.

Lines 407 – 414: Again suggest deleting. This is really muddying up the story you are trying to tell. REPLY: The text has been rewritten reflecting this referee comment.

Lines 427 – 431: This doesn't flow at all. Maybe have a section on each family like suggested for results? REPLY: The text has been rewritten reflecting this referee comment.

Lines 438 – 446: Why are you ending with species distinctions and not even those in your data? May be good to have a discussion section on taxonomy like suggested for results. End: You don't discussion implications for conservation really at all here. Is APEI3 good? Are many of these species at risk of extinction because of singletons? Does genetic differentiation vs. morphology tell different stories for regulators? Could you focus future work on the least highly-dispersed family as they are likely to be most impacted by mining? Etc. You need a paragraph or two at the end really tying this all together and telling people why it is important work. Conclusions: General: Conclusions are almost all other people's work, and much of it is taxonomic problems which don't seem to be the focus of this paper. What are the main points of this study (e.g., isopod lifestyle, CCZ similarity, etc.) REPLY: The text has been rewritten reflecting this referee comment.

Lines 448 – 450: Now I am confused. This study is focused on taxonomic incompleteness? I thought is was focused on isopod lifestyle and ranges. You need to have the

conclusion talk about the main points you are trying to make, and taxonomy isn't even in the title. REPLY: The text has been rewritten reflecting this referee comment.

Lines 448 – 458: This is way too much of other people's work for a conclusion in this paper. You can have a sentence or two about the need for more molecular work and morphological problems, but it needs to be shorter and not at the start, unless you want to change the focus of this paper. REPLY: The text has been rewritten reflecting this referee comment.

Line 459: How do you know distance and locomotion are "most" important? You tested everything? What else did you examine besides distance and locomotion? REPLY: The text has been rewritten reflecting this referee comment.

Line 460 – 461: "Long-distance populations. . . patchy/local populations" What does this mean exactly? REPLY: The text has been rewritten reflecting this referee comment.

Lines 460 – 468: Again, your conclusion is almost all other people's work. Need conclusions for this study. REPLY: The text has been rewritten reflecting this referee comment.

Tables and Figures: Table 1 should be supplementary. REPLY: It is now.

Please also note the supplement to this comment:
https://bg.copernicus.org/preprints/bg-2019-358/bg-2019-358-AC1-supplement.pdf

———————————

[Figure]

**Fig. 1.** Map of the locations of the EBS sampling sites (red dots) within the manganese nodule contractor and the DISCOL Experimental Area (DEA) areas in the north- and south-eastern Pacific. The colourcode in

[Figure]

**Fig. 2.** Illustration of the locomotion of the four isopod families. From right to left: Munnopsidae – swimming, Desmosomatidae – walking/swimming, Haploniscidae – walking, Macrostylidae – burrowing.

Decreasing Mobility

**Fig. 3.** Phylogenetic tree of all munnopsid samples based on 16S and COI sequences for 294 specimens. Colours indicate collection location, with black indicating outgroups. All unsupported branches were colla

**Fig. 4.** Phylogenetic tree of all desmosomatid samples based on 16S and COI sequences for 143 specimens. Colors indicate collection location, with black indicating outgroups. All unsupported branches were co

**Fig. 5.** Phylogenetic tree of all haploniscid samples based on 16S and COI sequences for 88 specimens. Colors indicate collection location, with black indicating outgroups. All unsupported branches were coll

**Fig. 6.** Phylogenetic tree of all macrostylid samples based on 16S and COI sequences for 94 specimens. Colors indicate collection location, with black indicating outgroups. All unsupported branches were coll

[Figure]

**Fig. 7.** Rarefaction analysis by isopod family, considering all areas together.

Fig. 8. Rarefaction analysis by area, considering all families together.

[Figure]

**Fig. 9.** nMDS ordination plot of Chord-distance between areas.

[Figure]

**Fig. 10.** nMDS ordination plot of Euclidean-distance between areas of presence-absence trans-
formed data.

**Chord distance to other areas**

[revised manuscript text omitted]

---

## Author Comment (AC2) · 1 Sep 2020

Dear referees, thank you very much for the constructive comments on our manuscript. Especially the direct implementation in the pdf made it highly valuable to work with. AllsSpecific comments from the annotated pdf were implented in the word doc manuscript version and will be visible via the track changes version. We did consider all comments from the annotated pdf.

Please also note the supplement to this comment:
https://bg.copernicus.org/preprints/bg-2019-358/bg-2019-358-AC2-supplement.pdf

[revised manuscript text omitted]

---

## Author Response (AR1)

[revised manuscript text omitted]

**Seite 1: [1] Formatvorlagendefinition**  **Osborn**  **31.07.2020 15:28:00**

Standard (Web)

**Seite 1: [2] Formatvorlagendefinition**  **Osborn**  **31.07.2020 15:28:00**

Listenabsatz

**Seite 1: [3] Formatvorlagendefinition**  **Osborn**  **31.07.2020 15:28:00**

Kommentartext: Schriftart: (Standard) Times New Roman, (Asiatisch) Chinesisch (China), (Andere) Englisch (Vereinigtes Königreich), Block, Abstand Nach: 0 Pt., Zeilenabstand: einfach, Keine Silbentrennung

**Seite 1: [4] Formatvorlagendefinition**  **Osborn**  **31.07.2020 15:28:00**

Kommentarthema: (Asiatisch) Chinesisch (China), Zeilenabstand: 1,5 Zeilen, Keine Silbentrennung

**Seite 1: [5] Formatvorlagendefinition**  **Osborn**  **31.07.2020 15:28:00**

Sprechblasentext: Schriftart: (Standard) Segoe UI, 9 Pt., (Asiatisch) Chinesisch (China), Keine Silbentrennung

**Seite 1: [6] Formatvorlagendefinition**  **Osborn**  **31.07.2020 15:28:00**

Fußzeile: (Asiatisch) Chinesisch (China), Zeilennummern unterdrücken, Keine Silbentrennung

**Seite 1: [7] Formatvorlagendefinition**  **Osborn**  **31.07.2020 15:28:00**

Equation: (Asiatisch) Chinesisch (China), (Andere) Englisch (Vereinigtes Königreich), Keine Silbentrennung

**Seite 1: [8] Formatvorlagendefinition**  **Osborn**  **31.07.2020 15:28:00**

Copernicus_Word_template: (Asiatisch) Chinesisch (China), (Andere) Englisch (Vereinigtes Königreich), Keine Silbentrennung

**Seite 1: [9] Formatvorlagendefinition**  **Osborn**  **31.07.2020 15:28:00**

Name: (Asiatisch) Chinesisch (China), Keine Silbentrennung

**Seite 1: [10] Formatvorlagendefinition**  **Osborn**  **31.07.2020 15:28:00**

Kontakt: (Asiatisch) Chinesisch (China), Keine Silbentrennung

**Seite 1: [11] Formatvorlagendefinition**  **Osborn**  **31.07.2020 15:28:00**

Kopfzeile: (Asiatisch) Chinesisch (China), Zeilennummern unterdrücken, Keine Silbentrennung

**Seite 1: [12] Formatvorlagendefinition**  **Osborn**  **31.07.2020 15:28:00**

Bullets: (Asiatisch) Chinesisch (China), (Andere) Englisch (Vereinigtes Königreich), Keine Aufzählungen oder Nummerierungen, Keine Silbentrennung

**Seite 1: [13] Formatvorlagendefinition**  **Osborn**  **31.07.2020 15:28:00**

Betreff: (Asiatisch) Chinesisch (China), Keine Silbentrennung

**Seite 1: [14] Formatvorlagendefinition**  **Osborn**  **31.07.2020 15:28:00**

Beschriftung: Schriftart: 12 Pt., Nicht Fett, Kursiv, (Asiatisch) Chinesisch (China), Abstand Vor: 6 Pt., Nach: 6 Pt., Zeilenabstand: 1,5 Zeilen, Zeilennummern unterdrücken, Keine Silbentrennung

**Seite 1: [15] Formatvorlagendefinition**  **Osborn**  **31.07.2020 15:28:00**

Kommentarzeichen

**Seite 1: [16] Formatvorlagendefinition**  **Osborn**  **31.07.2020 15:28:00**

Authors Char: (Keine Überprüfung)

**Seite 1: [17] Formatvorlagendefinition          Osborn                    31.07.2020 15:28:00**

Correspondence Char: (Keine Überprüfung)

**Seite 1: [18] Formatvorlagendefinition          Osborn                    31.07.2020 15:28:00**

Affiliation Char: (Keine Überprüfung)

**Seite 1: [19] Formatvorlagendefinition          Osborn                    31.07.2020 15:28:00**

MS title Char: (Keine Überprüfung)

**Seite 1: [20] Formatvorlagendefinition          Osborn                    31.07.2020 15:28:00**

Überschrift 4: (Asiatisch) Chinesisch (China), Mit Gliederung + Ebene: 4 + Ausgerichtet an: 0 cm + Einzug bei: 0 cm, Keine Silbentrennung

**Seite 1: [21] Formatvorlagendefinition          Osborn                    31.07.2020 15:28:00**

Überschrift 3: (Asiatisch) Chinesisch (China), Mit Gliederung + Ebene: 3 + Ausgerichtet an: 0 cm + Einzug bei: 0 cm, Keine Silbentrennung

**Seite 1: [22] Formatvorlagendefinition          Osborn                    31.07.2020 15:28:00**

Überschrift 2: (Asiatisch) Chinesisch (China), Mit Gliederung + Ebene: 2 + Ausgerichtet an: 0 cm + Einzug bei: 0 cm, Keine Silbentrennung

**Seite 1: [23] Formatvorlagendefinition          Osborn                    31.07.2020 15:28:00**

Überschrift 1: (Asiatisch) Chinesisch (China), Mit Gliederung + Ebene: 1 + Ausgerichtet an: 0 cm + Einzug bei: 0 cm, Keine Silbentrennung

**Seite 1: [24] Formatvorlagendefinition          Osborn                    31.07.2020 15:28:00**

Standard: (Asiatisch) Chinesisch (China), Keine Silbentrennung

**Seite 1: [25] hat formatiert          Osborn                    31.07.2020 15:28:00**

Schriftart: 12 Pt.

**Seite 1: [26] Formatiert          Osborn                    31.07.2020 15:28:00**

Oben: 1,37 cm, Breite: 21 cm, Kopfzeilenabstand vom Rand: 1,27 cm

**Seite 1: [27] hat formatiert          Osborn                    31.07.2020 15:28:00**

Schriftart: 12 Pt., Nicht Fett

**Seite 1: [28] hat formatiert          Osborn                    31.07.2020 15:28:00**

Hochgestellt

**Seite 1: [29] hat formatiert          Osborn                    31.07.2020 15:28:00**

Schriftart: 12 Pt.

**Seite 1: [30] hat formatiert          Osborn                    31.07.2020 15:28:00**

Schriftart: 12 Pt.

**Seite 1: [31] hat formatiert          Osborn                    31.07.2020 15:28:00**

Schriftart: 12 Pt., Nicht Kursiv

**Seite 1: [32] hat formatiert          Osborn                    31.07.2020 15:28:00**

Schriftart: 12 Pt.

| | | |
|---|---|---|
| **Seite 1: [33] hat formatiert** | **Osborn** | **31.07.2020 15:28:00** |

Schriftart: 12 Pt.

| | | |
|---|---|---|
| **Seite 1: [34] hat formatiert** | **Osborn** | **31.07.2020 15:28:00** |

Schriftart: 12 Pt.

| | | |
|---|---|---|
| **Seite 1: [35] hat formatiert** | **Osborn** | **31.07.2020 15:28:00** |

Überschrift 1 Zchn, Schriftart: 12 Pt.

| | | |
|---|---|---|
| **Seite 1: [36] Formatiert** | **Osborn** | **31.07.2020 15:28:00** |

Mit Gliederung + Ebene: 1 + Ausgerichtet an: 0 cm + Einzug bei: 0 cm

| | | |
|---|---|---|
| **Seite 1: [37] hat formatiert** | **Osborn** | **31.07.2020 15:28:00** |

Schriftart: 12 Pt.

| | | |
|---|---|---|
| **Seite 7: [38] hat formatiert** | **Osborn** | **31.07.2020 15:28:00** |

Schriftart: 12 Pt., Englisch (Vereinigte Staaten)

| | | |
|---|---|---|
| **Seite 7: [39] hat formatiert** | **Osborn** | **31.07.2020 15:28:00** |

Schriftart: 12 Pt., Englisch (Vereinigte Staaten)

| | | |
|---|---|---|
| **Seite 7: [40] hat formatiert** | **Osborn** | **31.07.2020 15:28:00** |

Schriftart: 12 Pt., Englisch (Vereinigte Staaten)

| | | |
|---|---|---|
| **Seite 7: [41] hat formatiert** | **Osborn** | **31.07.2020 15:28:00** |

Schriftart: 12 Pt., Englisch (Vereinigte Staaten)

| | | |
|---|---|---|
| **Seite 7: [42] hat formatiert** | **Osborn** | **31.07.2020 15:28:00** |

Schriftart: 12 Pt., Englisch (Vereinigte Staaten)

| | | |
|---|---|---|
| **Seite 7: [43] hat formatiert** | **Osborn** | **31.07.2020 15:28:00** |

Schriftart: 12 Pt., Englisch (Vereinigte Staaten)

| | | |
|---|---|---|
| **Seite 7: [44] hat formatiert** | **Osborn** | **31.07.2020 15:28:00** |

Schriftart: 12 Pt., Englisch (Vereinigte Staaten)

| | | |
|---|---|---|
| **Seite 7: [45] hat formatiert** | **Osborn** | **31.07.2020 15:28:00** |

Schriftart: 12 Pt., Englisch (Vereinigte Staaten)

| | | |
|---|---|---|
| **Seite 7: [46] hat formatiert** | **Osborn** | **31.07.2020 15:28:00** |

Schriftart: 12 Pt., Englisch (Vereinigte Staaten)

| | | |
|---|---|---|
| **Seite 7: [47] hat formatiert** | **Osborn** | **31.07.2020 15:28:00** |

Schriftart: 12 Pt., Englisch (Vereinigte Staaten)

| | | |
|---|---|---|
| **Seite 16: [48] hat formatiert** | **Osborn** | **31.07.2020 15:28:00** |

Schriftart: 12 Pt., Fett, Nicht Kursiv

| | | |
|---|---|---|
| **Seite 16: [49] hat formatiert** | **Osborn** | **31.07.2020 15:28:00** |

Schriftart: 12 Pt., Fett, Nicht Kursiv

| Seite 16: [50] hat formatiert | Osborn | 31.07.2020 15:28:00 |

Schriftart: 12 Pt., Nicht Kursiv

| Seite 16: [51] Formatiert | Osborn | 31.07.2020 15:28:00 |

Einzug: Erste Zeile:  1,27 cm

| Seite 16: [52] hat formatiert | Osborn | 31.07.2020 15:28:00 |

Schriftart: 12 Pt.

| Seite 16: [53] hat formatiert | Osborn | 31.07.2020 15:28:00 |

Schriftart: 12 Pt.

| Seite 16: [54] hat formatiert | Osborn | 31.07.2020 15:28:00 |

Schriftart: 12 Pt.

| Seite 16: [55] hat formatiert | Osborn | 31.07.2020 15:28:00 |

Schriftart: 12 Pt.

| Seite 16: [56] hat formatiert | Osborn | 31.07.2020 15:28:00 |

Schriftart: 12 Pt.

| Seite 16: [57] hat formatiert | Osborn | 31.07.2020 15:28:00 |

Schriftart: 12 Pt.

| Seite 16: [58] hat formatiert | Osborn | 31.07.2020 15:28:00 |

Schriftart: 12 Pt.

| Seite 16: [59] hat formatiert | Osborn | 31.07.2020 15:28:00 |

Schriftart: 12 Pt.

| Seite 16: [60] hat formatiert | Osborn | 31.07.2020 15:28:00 |

Schriftart: 12 Pt.

| Seite 16: [61] hat formatiert | Osborn | 31.07.2020 15:28:00 |

Schriftart: 12 Pt.

| Seite 16: [62] hat formatiert | Osborn | 31.07.2020 15:28:00 |

Schriftart: 12 Pt.

| Seite 16: [63] hat formatiert | Osborn | 31.07.2020 15:28:00 |

Schriftart: 12 Pt.

| Seite 16: [64] hat formatiert | Osborn | 31.07.2020 15:28:00 |

Schriftart: 12 Pt.

| Seite 16: [65] hat formatiert | Osborn | 31.07.2020 15:28:00 |

Schriftart: 12 Pt.

| Seite 16: [66] hat formatiert | Osborn | 31.07.2020 15:28:00 |

Schriftart: 12 Pt.

| Seite 16: [67] hat formatiert | Osborn | 31.07.2020 15:28:00 |

Schriftart: 12 Pt.

| Seite 16: [68] hat formatiert | Osborn | 31.07.2020 15:28:00 |

Schriftart: 12 Pt.

| Seite 16: [69] hat formatiert | Osborn | 31.07.2020 15:28:00 |

Schriftart: 12 Pt.

| Seite 16: [70] hat formatiert | Osborn | 31.07.2020 15:28:00 |

Schriftart: 12 Pt.

| Seite 16: [71] hat formatiert | Osborn | 31.07.2020 15:28:00 |

Schriftart: 12 Pt.

| Seite 16: [72] hat formatiert | Osborn | 31.07.2020 15:28:00 |

Schriftart: 12 Pt.

| Seite 16: [73] hat formatiert | Osborn | 31.07.2020 15:28:00 |

Schriftart: 12 Pt.

| Seite 16: [74] hat formatiert | Osborn | 31.07.2020 15:28:00 |

Schriftart: 12 Pt.

| Seite 16: [75] hat formatiert | Osborn | 31.07.2020 15:28:00 |

Schriftart: 12 Pt.

| Seite 16: [76] hat formatiert | Osborn | 31.07.2020 15:28:00 |

Schriftart: 12 Pt.

| Seite 16: [77] hat formatiert | Osborn | 31.07.2020 15:28:00 |

Schriftart: 12 Pt.

| Seite 16: [78] hat formatiert | Osborn | 31.07.2020 15:28:00 |

Schriftart: 12 Pt.

| Seite 16: [79] hat formatiert | Osborn | 31.07.2020 15:28:00 |

Schriftart: 12 Pt.

| Seite 16: [80] hat formatiert | Osborn | 31.07.2020 15:28:00 |

Schriftart: 12 Pt.

| Seite 16: [81] hat formatiert | Osborn | 31.07.2020 15:28:00 |

Schriftart: 12 Pt.

| Seite 16: [82] hat formatiert | Osborn | 31.07.2020 15:28:00 |

Schriftart: 12 Pt.

| Seite 16: [83] hat formatiert | Osborn | 31.07.2020 15:28:00 |

Schriftart: 12 Pt.

| Seite 16: [84] hat formatiert | Osborn | 31.07.2020 15:28:00 |

Schriftart: 12 Pt.

**Seite 16: [85] hat formatiert** | **Osborn** | **31.07.2020 15:28:00**

Schriftart: 12 Pt.

**Seite 24: [86] hat formatiert** | **Osborn** | **31.07.2020 15:28:00**

Schriftart: 12 Pt., Englisch (Vereinigte Staaten)

**Seite 24: [87] hat formatiert** | **Osborn** | **31.07.2020 15:28:00**

Schriftart: 12 Pt., Englisch (Vereinigte Staaten)

**Seite 24: [88] hat formatiert** | **Osborn** | **31.07.2020 15:28:00**

Schriftart: 12 Pt., Englisch (Vereinigte Staaten)

**Seite 24: [89] hat formatiert** | **Osborn** | **31.07.2020 15:28:00**

Schriftart: 12 Pt., Englisch (Vereinigte Staaten)

**Seite 24: [90] hat formatiert** | **Osborn** | **31.07.2020 15:28:00**

Schriftart: 12 Pt., Englisch (Vereinigte Staaten)

**Seite 24: [91] hat formatiert** | **Osborn** | **31.07.2020 15:28:00**

Schriftart: 12 Pt., Englisch (Vereinigte Staaten)

**Seite 24: [92] hat formatiert** | **Osborn** | **31.07.2020 15:28:00**

Schriftart: 12 Pt., Englisch (Vereinigte Staaten)

**Seite 24: [93] hat formatiert** | **Osborn** | **31.07.2020 15:28:00**

Schriftart: 12 Pt., Englisch (Vereinigte Staaten)

**Seite 24: [94] hat formatiert** | **Osborn** | **31.07.2020 15:28:00**

Schriftart: 12 Pt., Englisch (Vereinigte Staaten)

**Seite 24: [95] hat formatiert** | **Osborn** | **31.07.2020 15:28:00**

Schriftart: 12 Pt., Englisch (Vereinigte Staaten)

**Seite 24: [96] hat formatiert** | **Osborn** | **31.07.2020 15:28:00**

Schriftart: 12 Pt., Englisch (Vereinigte Staaten)

**Seite 24: [97] hat formatiert** | **Osborn** | **31.07.2020 15:28:00**

Schriftart: 12 Pt., Englisch (Vereinigte Staaten)

**Seite 24: [98] hat formatiert** | **Osborn** | **31.07.2020 15:28:00**

Schriftart: 12 Pt., Englisch (Vereinigte Staaten)

**Seite 24: [99] hat formatiert** | **Osborn** | **31.07.2020 15:28:00**

Schriftart: 12 Pt., Englisch (Vereinigte Staaten)

**Seite 24: [100] hat formatiert** | **Osborn** | **31.07.2020 15:28:00**

Schriftart: 12 Pt., Englisch (Vereinigte Staaten)

**Seite 24: [101] hat formatiert** | **Osborn** | **31.07.2020 15:28:00**

Schriftart: 12 Pt., Englisch (Vereinigte Staaten)

| Seite 24: [102] hat formatiert | Osborn | 31.07.2020 15:28:00 |
|---|---|---|

Schriftart: 12 Pt., Englisch (Vereinigte Staaten)

| Seite 24: [103] hat formatiert | Osborn | 31.07.2020 15:28:00 |
|---|---|---|

Schriftart: 12 Pt., Englisch (Vereinigte Staaten)

| Seite 24: [104] hat formatiert | Osborn | 31.07.2020 15:28:00 |
|---|---|---|

Schriftart: 12 Pt., Englisch (Vereinigte Staaten)

| Seite 62: [105] hat formatiert | Osborn | 31.07.2020 15:28:00 |
|---|---|---|

Schriftart: Nicht Kursiv

| Seite 62: [106] hat formatiert | Osborn | 31.07.2020 15:28:00 |
|---|---|---|

Schriftart: Times New Roman, 10 Pt.

| Seite 62: [106] hat formatiert | Osborn | 31.07.2020 15:28:00 |
|---|---|---|

Schriftart: Times New Roman, 10 Pt.

| Seite 62: [107] Formatiert | Unknown | 31.07.2020 15:28:00 |
|---|---|---|

Zentriert

| Seite 62: [108] hat formatiert | Osborn | 31.07.2020 15:28:00 |
|---|---|---|

Schriftart: Times New Roman, 10 Pt.

| Seite 62: [108] hat formatiert | Osborn | 31.07.2020 15:28:00 |
|---|---|---|

Schriftart: Times New Roman, 10 Pt.

| Seite 62: [109] hat formatiert | Osborn | 31.07.2020 15:28:00 |
|---|---|---|

Schriftart: Times New Roman, 10 Pt.

| Seite 62: [109] hat formatiert | Osborn | 31.07.2020 15:28:00 |
|---|---|---|

Schriftart: Times New Roman, 10 Pt.

| Seite 62: [110] hat formatiert | Osborn | 31.07.2020 15:28:00 |
|---|---|---|

Schriftart: Times New Roman, 10 Pt.

| Seite 62: [110] hat formatiert | Osborn | 31.07.2020 15:28:00 |
|---|---|---|

Schriftart: Times New Roman, 10 Pt.

| Seite 62: [111] hat formatiert | Osborn | 31.07.2020 15:28:00 |
|---|---|---|

Schriftart: Times New Roman, 10 Pt.

| Seite 62: [111] hat formatiert | Osborn | 31.07.2020 15:28:00 |
|---|---|---|

Schriftart: Times New Roman, 10 Pt.

| Seite 62: [112] hat formatiert | Osborn | 31.07.2020 15:28:00 |
|---|---|---|

Schriftart: Times New Roman, 10 Pt.

| Seite 62: [112] hat formatiert | Osborn | 31.07.2020 15:28:00 |
|---|---|---|

Schriftart: Times New Roman, 10 Pt.

| Seite 62: [113] hat formatiert | Osborn | 31.07.2020 15:28:00 |

Schriftart: Times New Roman, 10 Pt.

| Seite 62: [113] hat formatiert | Osborn | 31.07.2020 15:28:00 |

Schriftart: Times New Roman, 10 Pt.

| Seite 62: [114] hat formatiert | Osborn | 31.07.2020 15:28:00 |

Schriftart: Times New Roman, 10 Pt.

| Seite 62: [114] hat formatiert | Osborn | 31.07.2020 15:28:00 |

Schriftart: Times New Roman, 10 Pt.

| Seite 62: [115] hat formatiert | Osborn | 31.07.2020 15:28:00 |

Schriftart: Times New Roman, 10 Pt.

| Seite 62: [115] hat formatiert | Osborn | 31.07.2020 15:28:00 |

Schriftart: Times New Roman, 10 Pt.

| Seite 62: [116] hat formatiert | Osborn | 31.07.2020 15:28:00 |

Schriftart: Times New Roman, 10 Pt.

| Seite 62: [116] hat formatiert | Osborn | 31.07.2020 15:28:00 |

Schriftart: Times New Roman, 10 Pt.

| Seite 62: [117] hat formatiert | Osborn | 31.07.2020 15:28:00 |

Schriftart: Times New Roman, 10 Pt.

| Seite 62: [117] hat formatiert | Osborn | 31.07.2020 15:28:00 |

Schriftart: Times New Roman, 10 Pt.

| Seite 62: [118] hat formatiert | Osborn | 31.07.2020 15:28:00 |

Schriftart: Times New Roman, 10 Pt.

| Seite 62: [118] hat formatiert | Osborn | 31.07.2020 15:28:00 |

Schriftart: Times New Roman, 10 Pt.

| Seite 62: [119] hat formatiert | Osborn | 31.07.2020 15:28:00 |

Schriftart: Times New Roman, 10 Pt.

| Seite 62: [119] hat formatiert | Osborn | 31.07.2020 15:28:00 |

Schriftart: Times New Roman, 10 Pt.

| Seite 62: [120] hat formatiert | Osborn | 31.07.2020 15:28:00 |

Schriftart: Times New Roman, 10 Pt.

| Seite 62: [120] hat formatiert | Osborn | 31.07.2020 15:28:00 |

Schriftart: Times New Roman, 10 Pt.

| Seite 62: [121] hat formatiert | Osborn | 31.07.2020 15:28:00 |

Schriftart: Times New Roman, 10 Pt.

| Seite 62: [122] Formatiert | Unknown | 31.07.2020 15:28:00 |

Zentriert

| **Seite 62: [123] hat formatiert** | **Osborn** | **31.07.2020 15:28:00** |

Schriftart: Times New Roman, 11 Pt.

| **Seite 62: [124] hat formatiert** | **Osborn** | **31.07.2020 15:28:00** |

Schriftart: Times New Roman, 10 Pt.

| **Seite 62: [124] hat formatiert** | **Osborn** | **31.07.2020 15:28:00** |

Schriftart: Times New Roman, 10 Pt.

| **Seite 62: [125] hat formatiert** | **Osborn** | **31.07.2020 15:28:00** |

Schriftart: Times New Roman, 10 Pt.

| **Seite 62: [125] hat formatiert** | **Osborn** | **31.07.2020 15:28:00** |

Schriftart: Times New Roman, 10 Pt.

| **Seite 62: [126] hat formatiert** | **Osborn** | **31.07.2020 15:28:00** |

Schriftart: Times New Roman, 10 Pt.

| **Seite 62: [126] hat formatiert** | **Osborn** | **31.07.2020 15:28:00** |

Schriftart: Times New Roman, 10 Pt.

| **Seite 62: [127] hat formatiert** | **Osborn** | **31.07.2020 15:28:00** |

Schriftart: Times New Roman, 10 Pt.

| **Seite 62: [127] hat formatiert** | **Osborn** | **31.07.2020 15:28:00** |

Schriftart: Times New Roman, 10 Pt.

| **Seite 62: [128] hat formatiert** | **Osborn** | **31.07.2020 15:28:00** |

Schriftart: Times New Roman, 10 Pt.

| **Seite 62: [128] hat formatiert** | **Osborn** | **31.07.2020 15:28:00** |

Schriftart: Times New Roman, 10 Pt.

| **Seite 62: [129] hat formatiert** | **Osborn** | **31.07.2020 15:28:00** |

Schriftart: Times New Roman, 10 Pt.

| **Seite 62: [129] hat formatiert** | **Osborn** | **31.07.2020 15:28:00** |

Schriftart: Times New Roman, 10 Pt.

| **Seite 62: [130] hat formatiert** | **Osborn** | **31.07.2020 15:28:00** |

Schriftart: Times New Roman, 10 Pt.

| **Seite 62: [130] hat formatiert** | **Osborn** | **31.07.2020 15:28:00** |

Schriftart: Times New Roman, 10 Pt.

| **Seite 62: [131] hat formatiert** | **Osborn** | **31.07.2020 15:28:00** |

Schriftart: Times New Roman, 10 Pt.

| **Seite 62: [131] hat formatiert** | **Osborn** | **31.07.2020 15:28:00** |

Schriftart: Times New Roman, 10 Pt.

| Seite 62: [132] hat formatiert | Osborn | 31.07.2020 15:28:00 |
|---|---|---|

Schriftart: Times New Roman, 10 Pt.

| Seite 62: [132] hat formatiert | Osborn | 31.07.2020 15:28:00 |
|---|---|---|

Schriftart: Times New Roman, 10 Pt.

| Seite 62: [133] hat formatiert | Osborn | 31.07.2020 15:28:00 |
|---|---|---|

Schriftart: Times New Roman, 10 Pt.

| Seite 62: [133] hat formatiert | Osborn | 31.07.2020 15:28:00 |
|---|---|---|

Schriftart: Times New Roman, 10 Pt.

| Seite 62: [134] hat formatiert | Osborn | 31.07.2020 15:28:00 |
|---|---|---|

Schriftart: Times New Roman, 10 Pt.

| Seite 62: [134] hat formatiert | Osborn | 31.07.2020 15:28:00 |
|---|---|---|

Schriftart: Times New Roman, 10 Pt.

| Seite 62: [135] hat formatiert | Osborn | 31.07.2020 15:28:00 |
|---|---|---|

Schriftart: Times New Roman, 10 Pt.

| Seite 62: [135] hat formatiert | Osborn | 31.07.2020 15:28:00 |
|---|---|---|

Schriftart: Times New Roman, 10 Pt.

| Seite 62: [136] hat formatiert | Osborn | 31.07.2020 15:28:00 |
|---|---|---|

Schriftart: Times New Roman, 10 Pt.

| Seite 62: [137] Formatiert | Unknown | 31.07.2020 15:28:00 |
|---|---|---|

Zentriert

| Seite 62: [138] hat formatiert | Osborn | 31.07.2020 15:28:00 |
|---|---|---|

Schriftart: Times New Roman, 11 Pt.

| Seite 62: [139] hat formatiert | Osborn | 31.07.2020 15:28:00 |
|---|---|---|

Schriftart: Times New Roman, 10 Pt.

| Seite 62: [139] hat formatiert | Osborn | 31.07.2020 15:28:00 |
|---|---|---|

Schriftart: Times New Roman, 10 Pt.

| Seite 62: [140] hat formatiert | Osborn | 31.07.2020 15:28:00 |
|---|---|---|

Schriftart: Times New Roman, 10 Pt.

| Seite 62: [140] hat formatiert | Osborn | 31.07.2020 15:28:00 |
|---|---|---|

Schriftart: Times New Roman, 10 Pt.

| Seite 62: [141] hat formatiert | Osborn | 31.07.2020 15:28:00 |
|---|---|---|

Schriftart: Times New Roman, 10 Pt.

| Seite 62: [141] hat formatiert | Osborn | 31.07.2020 15:28:00 |
|---|---|---|

Schriftart: Times New Roman, 10 Pt.

| **Seite 62: [142] hat formatiert** | **Osborn** | **31.07.2020 15:28:00** |

Schriftart: Times New Roman, 10 Pt.

| **Seite 62: [142] hat formatiert** | **Osborn** | **31.07.2020 15:28:00** |

Schriftart: Times New Roman, 10 Pt.

| **Seite 62: [143] hat formatiert** | **Osborn** | **31.07.2020 15:28:00** |

Schriftart: Times New Roman, 10 Pt.

| **Seite 62: [143] hat formatiert** | **Osborn** | **31.07.2020 15:28:00** |

Schriftart: Times New Roman, 10 Pt.

| **Seite 62: [144] hat formatiert** | **Osborn** | **31.07.2020 15:28:00** |

Schriftart: Times New Roman, 10 Pt.

| **Seite 62: [144] hat formatiert** | **Osborn** | **31.07.2020 15:28:00** |

Schriftart: Times New Roman, 10 Pt.

| **Seite 62: [145] hat formatiert** | **Osborn** | **31.07.2020 15:28:00** |

Schriftart: Times New Roman, 10 Pt.

| **Seite 62: [145] hat formatiert** | **Osborn** | **31.07.2020 15:28:00** |

Schriftart: Times New Roman, 10 Pt.

| **Seite 62: [146] hat formatiert** | **Osborn** | **31.07.2020 15:28:00** |

Schriftart: Times New Roman, 10 Pt.

| **Seite 62: [146] hat formatiert** | **Osborn** | **31.07.2020 15:28:00** |

Schriftart: Times New Roman, 10 Pt.

| **Seite 62: [147] hat formatiert** | **Osborn** | **31.07.2020 15:28:00** |

Schriftart: Times New Roman, 10 Pt.

| **Seite 62: [147] hat formatiert** | **Osborn** | **31.07.2020 15:28:00** |

Schriftart: Times New Roman, 10 Pt.

| **Seite 62: [148] hat formatiert** | **Osborn** | **31.07.2020 15:28:00** |

Schriftart: Times New Roman, 10 Pt.

| **Seite 62: [148] hat formatiert** | **Osborn** | **31.07.2020 15:28:00** |

Schriftart: Times New Roman, 10 Pt.

| **Seite 62: [149] hat formatiert** | **Osborn** | **31.07.2020 15:28:00** |

Schriftart: Times New Roman, 10 Pt.

| **Seite 62: [149] hat formatiert** | **Osborn** | **31.07.2020 15:28:00** |

Schriftart: Times New Roman, 10 Pt.

| **Seite 62: [150] hat formatiert** | **Osborn** | **31.07.2020 15:28:00** |

Schriftart: Times New Roman, 10 Pt.

| **Seite 62: [150] hat formatiert** | **Osborn** | **31.07.2020 15:28:00** |

Schriftart: Times New Roman, 10 Pt.

| **Seite 62: [151] hat formatiert** | **Osborn** | **31.07.2020 15:28:00** |

Schriftart: Times New Roman, 10 Pt.

| **Seite 62: [152] Formatiert** | **Unknown** | **31.07.2020 15:28:00** |

Zentriert

| **Seite 62: [153] hat formatiert** | **Osborn** | **31.07.2020 15:28:00** |

Schriftart: Times New Roman, 11 Pt.

| **Seite 62: [154] hat formatiert** | **Osborn** | **31.07.2020 15:28:00** |

Schriftart: Times New Roman, 10 Pt.

| **Seite 62: [154] hat formatiert** | **Osborn** | **31.07.2020 15:28:00** |

Schriftart: Times New Roman, 10 Pt.

| **Seite 62: [155] hat formatiert** | **Osborn** | **31.07.2020 15:28:00** |

Schriftart: Times New Roman, 10 Pt.

| **Seite 62: [155] hat formatiert** | **Osborn** | **31.07.2020 15:28:00** |

Schriftart: Times New Roman, 10 Pt.

| **Seite 62: [156] hat formatiert** | **Osborn** | **31.07.2020 15:28:00** |

Schriftart: Times New Roman, 10 Pt.

| **Seite 62: [156] hat formatiert** | **Osborn** | **31.07.2020 15:28:00** |

Schriftart: Times New Roman, 10 Pt.

| **Seite 62: [157] hat formatiert** | **Osborn** | **31.07.2020 15:28:00** |

Schriftart: Times New Roman, 10 Pt.

| **Seite 62: [157] hat formatiert** | **Osborn** | **31.07.2020 15:28:00** |

Schriftart: Times New Roman, 10 Pt.

| **Seite 62: [158] hat formatiert** | **Osborn** | **31.07.2020 15:28:00** |

Schriftart: Times New Roman, 10 Pt.

| **Seite 62: [158] hat formatiert** | **Osborn** | **31.07.2020 15:28:00** |

Schriftart: Times New Roman, 10 Pt.

| **Seite 62: [159] hat formatiert** | **Osborn** | **31.07.2020 15:28:00** |

Schriftart: Times New Roman, 10 Pt.

| **Seite 62: [159] hat formatiert** | **Osborn** | **31.07.2020 15:28:00** |

Schriftart: Times New Roman, 10 Pt.

| **Seite 62: [160] hat formatiert** | **Osborn** | **31.07.2020 15:28:00** |

Schriftart: Times New Roman, 10 Pt.

| **Seite 62: [160] hat formatiert** | **Osborn** | **31.07.2020 15:28:00** |

Schriftart: Times New Roman, 10 Pt.

| Seite 62: [161] hat formatiert | Osborn | 31.07.2020 15:28:00 |
|---|---|---|

Schriftart: Times New Roman, 10 Pt.

| Seite 62: [161] hat formatiert | Osborn | 31.07.2020 15:28:00 |
|---|---|---|

Schriftart: Times New Roman, 10 Pt.

| Seite 62: [162] hat formatiert | Osborn | 31.07.2020 15:28:00 |
|---|---|---|

Schriftart: Times New Roman, 10 Pt.

| Seite 62: [162] hat formatiert | Osborn | 31.07.2020 15:28:00 |
|---|---|---|

Schriftart: Times New Roman, 10 Pt.

| Seite 62: [163] hat formatiert | Osborn | 31.07.2020 15:28:00 |
|---|---|---|

Schriftart: Times New Roman, 10 Pt.

| Seite 62: [163] hat formatiert | Osborn | 31.07.2020 15:28:00 |
|---|---|---|

Schriftart: Times New Roman, 10 Pt.

| Seite 62: [164] hat formatiert | Osborn | 31.07.2020 15:28:00 |
|---|---|---|

Schriftart: Times New Roman, 10 Pt.

| Seite 62: [164] hat formatiert | Osborn | 31.07.2020 15:28:00 |
|---|---|---|

Schriftart: Times New Roman, 10 Pt.

| Seite 62: [165] hat formatiert | Osborn | 31.07.2020 15:28:00 |
|---|---|---|

Schriftart: Times New Roman, 10 Pt.

| Seite 62: [165] hat formatiert | Osborn | 31.07.2020 15:28:00 |
|---|---|---|

Schriftart: Times New Roman, 10 Pt.

| Seite 62: [166] hat formatiert | Osborn | 31.07.2020 15:28:00 |
|---|---|---|

Schriftart: Times New Roman, 10 Pt.

| Seite 62: [167] Formatiert | Unknown | 31.07.2020 15:28:00 |
|---|---|---|

Zentriert

| Seite 62: [168] hat formatiert | Osborn | 31.07.2020 15:28:00 |
|---|---|---|

Schriftart: Times New Roman, 11 Pt.

| Seite 62: [169] hat formatiert | Osborn | 31.07.2020 15:28:00 |
|---|---|---|

Schriftart: Times New Roman, 10 Pt.

| Seite 62: [169] hat formatiert | Osborn | 31.07.2020 15:28:00 |
|---|---|---|

Schriftart: Times New Roman, 10 Pt.

| Seite 62: [170] hat formatiert | Osborn | 31.07.2020 15:28:00 |
|---|---|---|

Schriftart: Times New Roman, 10 Pt.

| Seite 62: [170] hat formatiert | Osborn | 31.07.2020 15:28:00 |
|---|---|---|

Schriftart: Times New Roman, 10 Pt.

| | | |
|---|---|---|
| **Seite 62: [171] hat formatiert** | **Osborn** | **31.07.2020 15:28:00** |

Schriftart: Times New Roman, 10 Pt.

| | | |
|---|---|---|
| **Seite 62: [171] hat formatiert** | **Osborn** | **31.07.2020 15:28:00** |

Schriftart: Times New Roman, 10 Pt.

| | | |
|---|---|---|
| **Seite 62: [172] hat formatiert** | **Osborn** | **31.07.2020 15:28:00** |

Schriftart: Times New Roman, 10 Pt.

| | | |
|---|---|---|
| **Seite 62: [172] hat formatiert** | **Osborn** | **31.07.2020 15:28:00** |

Schriftart: Times New Roman, 10 Pt.

| | | |
|---|---|---|
| **Seite 62: [173] hat formatiert** | **Osborn** | **31.07.2020 15:28:00** |

Schriftart: Times New Roman, 10 Pt.

| | | |
|---|---|---|
| **Seite 62: [173] hat formatiert** | **Osborn** | **31.07.2020 15:28:00** |

Schriftart: Times New Roman, 10 Pt.

| | | |
|---|---|---|
| **Seite 62: [174] hat formatiert** | **Osborn** | **31.07.2020 15:28:00** |

Schriftart: Times New Roman, 10 Pt.

| | | |
|---|---|---|
| **Seite 62: [174] hat formatiert** | **Osborn** | **31.07.2020 15:28:00** |

Schriftart: Times New Roman, 10 Pt.

| | | |
|---|---|---|
| **Seite 62: [175] hat formatiert** | **Osborn** | **31.07.2020 15:28:00** |

Schriftart: Times New Roman, 10 Pt.

| | | |
|---|---|---|
| **Seite 62: [175] hat formatiert** | **Osborn** | **31.07.2020 15:28:00** |

Schriftart: Times New Roman, 10 Pt.

| | | |
|---|---|---|
| **Seite 62: [176] hat formatiert** | **Osborn** | **31.07.2020 15:28:00** |

Schriftart: Times New Roman, 10 Pt.

| | | |
|---|---|---|
| **Seite 62: [176] hat formatiert** | **Osborn** | **31.07.2020 15:28:00** |

Schriftart: Times New Roman, 10 Pt.

| | | |
|---|---|---|
| **Seite 62: [177] hat formatiert** | **Osborn** | **31.07.2020 15:28:00** |

Schriftart: Times New Roman, 10 Pt.

| | | |
|---|---|---|
| **Seite 62: [177] hat formatiert** | **Osborn** | **31.07.2020 15:28:00** |

Schriftart: Times New Roman, 10 Pt.

| | | |
|---|---|---|
| **Seite 62: [178] hat formatiert** | **Osborn** | **31.07.2020 15:28:00** |

Schriftart: Times New Roman, 10 Pt.

| | | |
|---|---|---|
| **Seite 62: [178] hat formatiert** | **Osborn** | **31.07.2020 15:28:00** |

Schriftart: Times New Roman, 10 Pt.

| | | |
|---|---|---|
| **Seite 62: [179] hat formatiert** | **Osborn** | **31.07.2020 15:28:00** |

Schriftart: Times New Roman, 10 Pt.

| | | |
|---|---|---|
| **Seite 62: [179] hat formatiert** | **Osborn** | **31.07.2020 15:28:00** |

Schriftart: Times New Roman, 10 Pt.

| | | |
|---|---|---|
| **Seite 62: [180] hat formatiert** | **Osborn** | **31.07.2020 15:28:00** |

Schriftart: Times New Roman, 12 Pt., Fett

| | | |
|---|---|---|
| **Seite 62: [181] Formatiert** | **Osborn** | **31.07.2020 15:28:00** |

Beschriftung, Abstand Nach:  0 Pt., Zeilenabstand:  einfach

| | | |
|---|---|---|
| **Seite 62: [182] hat formatiert** | **Osborn** | **31.07.2020 15:28:00** |

Schriftart: Fett, Nicht Kursiv

| | | |
|---|---|---|
| **Seite 62: [182] hat formatiert** | **Osborn** | **31.07.2020 15:28:00** |

Schriftart: Fett, Nicht Kursiv

| | | |
|---|---|---|
| **Seite 62: [182] hat formatiert** | **Osborn** | **31.07.2020 15:28:00** |

Schriftart: Fett, Nicht Kursiv

| | | |
|---|---|---|
| **Seite 62: [182] hat formatiert** | **Osborn** | **31.07.2020 15:28:00** |

Schriftart: Fett, Nicht Kursiv

| | | |
|---|---|---|
| **Seite 62: [183] hat formatiert** | **Osborn** | **31.07.2020 15:28:00** |

Schriftart: Times New Roman, 12 Pt., Schriftfarbe: Text 1

| | | |
|---|---|---|
| **Seite 62: [184] Formatiert** | **Unknown** | **31.07.2020 15:28:00** |

Zentriert

| | | |
|---|---|---|
| **Seite 62: [185] hat formatiert** | **Osborn** | **31.07.2020 15:28:00** |

Schriftart: Times New Roman, 12 Pt., Schriftfarbe: Text 1

| | | |
|---|---|---|
| **Seite 62: [186] hat formatiert** | **Osborn** | **31.07.2020 15:28:00** |

Schriftart: Times New Roman, 12 Pt., Schriftfarbe: Text 1

| | | |
|---|---|---|
| **Seite 62: [187] hat formatiert** | **Osborn** | **31.07.2020 15:28:00** |

Schriftart: Times New Roman, 12 Pt., Schriftfarbe: Text 1

| | | |
|---|---|---|
| **Seite 62: [188] hat formatiert** | **Osborn** | **31.07.2020 15:28:00** |

Schriftart: Times New Roman, 12 Pt., Schriftfarbe: Text 1

| | | |
|---|---|---|
| **Seite 62: [189] hat formatiert** | **Osborn** | **31.07.2020 15:28:00** |

Schriftart: Times New Roman, 12 Pt., Schriftfarbe: Text 1

| | | |
|---|---|---|
| **Seite 63: [190] hat formatiert** | **Osborn** | **31.07.2020 15:28:00** |

Schriftart: Times New Roman, 12 Pt., Nicht Fett, Schriftfarbe: Text 1

| | | |
|---|---|---|
| **Seite 63: [191] Formatiert** | **Unknown** | **31.07.2020 15:28:00** |

Zentriert

| | | |
|---|---|---|
| **Seite 63: [192] hat formatiert** | **Osborn** | **31.07.2020 15:28:00** |

Schriftart: Times New Roman, 12 Pt., Schriftfarbe: Text 1

| | | |
|---|---|---|
| **Seite 63: [193] hat formatiert** | **Osborn** | **31.07.2020 15:28:00** |

Schriftart: Times New Roman, 12 Pt., Schriftfarbe: Text 1

| | | |
|---|---|---|
| **Seite 63: [194] hat formatiert** | **Osborn** | **31.07.2020 15:28:00** |

Schriftart: Times New Roman, 12 Pt., Schriftfarbe: Text 1

| | | |
|---|---|---|
| **Seite 63: [195] hat formatiert** | **Osborn** | **31.07.2020 15:28:00** |

Schriftart: Times New Roman, 12 Pt., Schriftfarbe: Text 1

| | | |
|---|---|---|
| **Seite 63: [196] hat formatiert** | **Osborn** | **31.07.2020 15:28:00** |

Schriftart: Times New Roman, 12 Pt., Nicht Fett, Schriftfarbe: Text 1

| | | |
|---|---|---|
| **Seite 63: [197] Formatiert** | **Unknown** | **31.07.2020 15:28:00** |

Zentriert

| | | |
|---|---|---|
| **Seite 63: [198] hat formatiert** | **Osborn** | **31.07.2020 15:28:00** |

Schriftart: Times New Roman, 12 Pt., Schriftfarbe: Text 1

| | | |
|---|---|---|
| **Seite 63: [199] hat formatiert** | **Osborn** | **31.07.2020 15:28:00** |

Schriftart: Times New Roman, 12 Pt., Schriftfarbe: Text 1

| | | |
|---|---|---|
| **Seite 63: [200] hat formatiert** | **Osborn** | **31.07.2020 15:28:00** |

Schriftart: Times New Roman, 12 Pt., Schriftfarbe: Text 1

| | | |
|---|---|---|
| **Seite 63: [201] hat formatiert** | **Osborn** | **31.07.2020 15:28:00** |

Schriftart: Times New Roman, 12 Pt., Schriftfarbe: Text 1

| | | |
|---|---|---|
| **Seite 63: [202] hat formatiert** | **Osborn** | **31.07.2020 15:28:00** |

Schriftart: Times New Roman, 12 Pt., Schriftfarbe: Text 1

| | | |
|---|---|---|
| **Seite 63: [203] hat formatiert** | **Osborn** | **31.07.2020 15:28:00** |

Schriftart: Times New Roman, 12 Pt., Schriftfarbe: Text 1

| | | |
|---|---|---|
| **Seite 63: [204] hat formatiert** | **Osborn** | **31.07.2020 15:28:00** |

Schriftart: Times New Roman, 12 Pt., Nicht Fett, Schriftfarbe: Text 1

| | | |
|---|---|---|
| **Seite 63: [205] Formatiert** | **Unknown** | **31.07.2020 15:28:00** |

Zentriert

| | | |
|---|---|---|
| **Seite 63: [206] hat formatiert** | **Osborn** | **31.07.2020 15:28:00** |

Schriftart: Times New Roman, 12 Pt., Fett, Schriftfarbe: Text 1

| | | |
|---|---|---|
| **Seite 63: [207] hat formatiert** | **Osborn** | **31.07.2020 15:28:00** |

Schriftart: Times New Roman, 12 Pt., Fett, Schriftfarbe: Text 1

| | | |
|---|---|---|
| **Seite 63: [208] hat formatiert** | **Osborn** | **31.07.2020 15:28:00** |

Schriftart: Times New Roman, 12 Pt., Schriftfarbe: Text 1

| | | |
|---|---|---|
| **Seite 63: [209] hat formatiert** | **Osborn** | **31.07.2020 15:28:00** |

Schriftart: Times New Roman, 12 Pt., Schriftfarbe: Text 1

**Seite 63: [210] hat formatiert** — **Osborn** — **31.07.2020 15:28:00**

Schriftart: Times New Roman, 12 Pt., Schriftfarbe: Text 1

**Seite 63: [211] hat formatiert** — **Osborn** — **31.07.2020 15:28:00**

Schriftart: Times New Roman, 12 Pt., Schriftfarbe: Text 1

**Seite 63: [212] hat formatiert** — **Osborn** — **31.07.2020 15:28:00**

Schriftart: Times New Roman, 12 Pt., Schriftfarbe: Text 1

**Seite 63: [212] hat formatiert** — **Osborn** — **31.07.2020 15:28:00**

Schriftart: Times New Roman, 12 Pt., Schriftfarbe: Text 1

**Seite 63: [213] Formatiert** — **Unknown** — **31.07.2020 15:28:00**

Zentriert

**Seite 63: [214] hat formatiert** — **Osborn** — **31.07.2020 15:28:00**

Schriftart: Times New Roman, 12 Pt., Schriftfarbe: Text 1

**Seite 63: [215] hat formatiert** — **Osborn** — **31.07.2020 15:28:00**

Schriftart: Times New Roman, 12 Pt., Schriftfarbe: Text 1

**Seite 63: [216] hat formatiert** — **Osborn** — **31.07.2020 15:28:00**

Schriftart: Times New Roman, 12 Pt., Fett, Schriftfarbe: Text 1

**Seite 63: [216] hat formatiert** — **Osborn** — **31.07.2020 15:28:00**

Schriftart: Times New Roman, 12 Pt., Fett, Schriftfarbe: Text 1

**Seite 63: [217] hat formatiert** — **Osborn** — **31.07.2020 15:28:00**

Schriftart: Times New Roman, 12 Pt., Schriftfarbe: Text 1

**Seite 63: [218] hat formatiert** — **Osborn** — **31.07.2020 15:28:00**

Schriftart: Times New Roman, 12 Pt., Schriftfarbe: Text 1

**Seite 63: [219] hat formatiert** — **Osborn** — **31.07.2020 15:28:00**

Schriftart: Times New Roman, 12 Pt., Nicht Fett, Schriftfarbe: Text 1

**Seite 63: [220] hat formatiert** — **Osborn** — **31.07.2020 15:28:00**

Schriftart: Times New Roman, 12 Pt., Schriftfarbe: Text 1

**Seite 63: [221] Formatiert** — **Unknown** — **31.07.2020 15:28:00**

Zentriert

**Seite 63: [222] hat formatiert** — **Osborn** — **31.07.2020 15:28:00**

Schriftart: Times New Roman, 12 Pt., Schriftfarbe: Text 1

**Seite 63: [223] hat formatiert** — **Osborn** — **31.07.2020 15:28:00**

Schriftart: Times New Roman, 12 Pt., Schriftfarbe: Text 1

**Seite 63: [224] hat formatiert** — **Osborn** — **31.07.2020 15:28:00**

Schriftart: Times New Roman, 12 Pt., Schriftfarbe: Text 1

**Seite 63: [225] hat formatiert** — **Osborn** — **31.07.2020 15:28:00**

Schriftart: Times New Roman, 12 Pt., Fett, Schriftfarbe: Text 1

| Seite 63: [226] hat formatiert | Osborn | 31.07.2020 15:28:00 |

Schriftart: Times New Roman, 12 Pt., Schriftfarbe: Text 1

| Seite 63: [227] hat formatiert | Osborn | 31.07.2020 15:28:00 |

Schriftart: Times New Roman, 12 Pt., Nicht Fett, Schriftfarbe: Text 1

| Seite 63: [228] hat formatiert | Osborn | 31.07.2020 15:28:00 |

Schriftart: Times New Roman, 12 Pt., Schriftfarbe: Text 1

| Seite 63: [229] Formatiert | Unknown | 31.07.2020 15:28:00 |

Zentriert

| Seite 63: [230] hat formatiert | Osborn | 31.07.2020 15:28:00 |

Schriftart: Times New Roman, 12 Pt., Schriftfarbe: Text 1

| Seite 63: [231] hat formatiert | Osborn | 31.07.2020 15:28:00 |

Schriftart: Times New Roman, 12 Pt., Schriftfarbe: Text 1

| Seite 63: [232] hat formatiert | Osborn | 31.07.2020 15:28:00 |

Schriftart: Times New Roman, 12 Pt., Schriftfarbe: Text 1

| Seite 63: [233] hat formatiert | Osborn | 31.07.2020 15:28:00 |

Schriftart: Times New Roman, 12 Pt., Schriftfarbe: Text 1

| Seite 63: [234] hat formatiert | Osborn | 31.07.2020 15:28:00 |

Schriftart: Times New Roman, 12 Pt., Schriftfarbe: Text 1

| Seite 63: [235] hat formatiert | Osborn | 31.07.2020 15:28:00 |

Schriftart: Fett, Nicht Kursiv

| Seite 63: [236] Formatiert | Osborn | 31.07.2020 15:28:00 |

Abstand Vor: 0 Pt.

| Seite 63: [237] hat formatiert | Osborn | 31.07.2020 15:28:00 |

Schriftart: Nicht Kursiv

| Seite 63: [237] hat formatiert | Osborn | 31.07.2020 15:28:00 |

Schriftart: Nicht Kursiv

| Seite 63: [238] hat formatiert | Osborn | 31.07.2020 15:28:00 |

Schriftart: Times New Roman

| Seite 63: [239] hat formatiert | Osborn | 31.07.2020 15:28:00 |

Schriftart: Times New Roman, 10 Pt., Nicht Fett

| Seite 63: [240] Formatiert | Unknown | 31.07.2020 15:28:00 |

Zentriert, Position: Horizontal: Links, Gemessen von: Seitenrand, Vertikal: 3,9 cm, Gemessen von: Seite, Horizontal: 0,32 cm, Umschließen

| Seite 63: [241] hat formatiert | Osborn | 31.07.2020 15:28:00 |

Schriftart: Times New Roman, 11 Pt.

| Seite 63: [242] hat formatiert | Osborn | 31.07.2020 15:28:00 |
|---|---|---|

Schriftart: Times New Roman, 10 Pt., Nicht Fett

| Seite 63: [242] hat formatiert | Osborn | 31.07.2020 15:28:00 |
|---|---|---|

Schriftart: Times New Roman, 10 Pt., Nicht Fett

| Seite 63: [243] hat formatiert | Osborn | 31.07.2020 15:28:00 |
|---|---|---|

Schriftart: Times New Roman, 10 Pt., Nicht Fett

| Seite 63: [243] hat formatiert | Osborn | 31.07.2020 15:28:00 |
|---|---|---|

Schriftart: Times New Roman, 10 Pt., Nicht Fett

| Seite 63: [244] hat formatiert | Osborn | 31.07.2020 15:28:00 |
|---|---|---|

Schriftart: Times New Roman, 10 Pt., Nicht Fett

| Seite 63: [244] hat formatiert | Osborn | 31.07.2020 15:28:00 |
|---|---|---|

Schriftart: Times New Roman, 10 Pt., Nicht Fett

| Seite 63: [245] hat formatiert | Osborn | 31.07.2020 15:28:00 |
|---|---|---|

Schriftart: Times New Roman, 10 Pt., Nicht Fett

| Seite 63: [245] hat formatiert | Osborn | 31.07.2020 15:28:00 |
|---|---|---|

Schriftart: Times New Roman, 10 Pt., Nicht Fett

| Seite 63: [246] hat formatiert | Osborn | 31.07.2020 15:28:00 |
|---|---|---|

Schriftart: Times New Roman, 10 Pt., Nicht Fett

| Seite 63: [246] hat formatiert | Osborn | 31.07.2020 15:28:00 |
|---|---|---|

Schriftart: Times New Roman, 10 Pt., Nicht Fett

| Seite 63: [247] hat formatiert | Osborn | 31.07.2020 15:28:00 |
|---|---|---|

Schriftart: Times New Roman, 10 Pt., Nicht Fett

| Seite 63: [247] hat formatiert | Osborn | 31.07.2020 15:28:00 |
|---|---|---|

Schriftart: Times New Roman, 10 Pt., Nicht Fett

| Seite 63: [248] hat formatiert | Osborn | 31.07.2020 15:28:00 |
|---|---|---|

Schriftart: Times New Roman, 10 Pt., Nicht Fett

| Seite 63: [248] hat formatiert | Osborn | 31.07.2020 15:28:00 |
|---|---|---|

Schriftart: Times New Roman, 10 Pt., Nicht Fett

| Seite 63: [249] hat formatiert | Osborn | 31.07.2020 15:28:00 |
|---|---|---|

Schriftart: Times New Roman, 10 Pt., Nicht Fett

| Seite 63: [249] hat formatiert | Osborn | 31.07.2020 15:28:00 |
|---|---|---|

Schriftart: Times New Roman, 10 Pt., Nicht Fett

| Seite 63: [250] hat formatiert | Osborn | 31.07.2020 15:28:00 |
|---|---|---|

Schriftart: Times New Roman, 10 Pt., Nicht Fett

| Seite 63: [250] hat formatiert | Osborn | 31.07.2020 15:28:00 |

Schriftart: Times New Roman, 10 Pt., Nicht Fett

| Seite 63: [251] hat formatiert | Osborn | 31.07.2020 15:28:00 |

Schriftart: Times New Roman, 10 Pt., Nicht Fett

| Seite 63: [251] hat formatiert | Osborn | 31.07.2020 15:28:00 |

Schriftart: Times New Roman, 10 Pt., Nicht Fett

| Seite 63: [252] hat formatiert | Osborn | 31.07.2020 15:28:00 |

Schriftart: Times New Roman, 10 Pt.

| Seite 63: [252] hat formatiert | Osborn | 31.07.2020 15:28:00 |

Schriftart: Times New Roman, 10 Pt.

| Seite 63: [253] hat formatiert | Osborn | 31.07.2020 15:28:00 |

Schriftart: Times New Roman, 10 Pt.

| Seite 63: [254] Formatiert | Unknown | 31.07.2020 15:28:00 |

Zentriert, Position: Horizontal: Links, Gemessen von: Seitenrand, Vertikal:  3,9 cm, Gemessen von: Seite,

Horizontal:  0,32 cm, Umschließen

| Seite 63: [255] hat formatiert | Osborn | 31.07.2020 15:28:00 |

Schriftart: Times New Roman, 11 Pt.

| Seite 63: [256] hat formatiert | Osborn | 31.07.2020 15:28:00 |

Schriftart: Times New Roman, 10 Pt.

| Seite 63: [256] hat formatiert | Osborn | 31.07.2020 15:28:00 |

Schriftart: Times New Roman, 10 Pt.

| Seite 63: [257] hat formatiert | Osborn | 31.07.2020 15:28:00 |

Schriftart: Times New Roman, 10 Pt.

| Seite 63: [257] hat formatiert | Osborn | 31.07.2020 15:28:00 |

Schriftart: Times New Roman, 10 Pt.

| Seite 63: [258] hat formatiert | Osborn | 31.07.2020 15:28:00 |

Schriftart: Times New Roman, 10 Pt.

| Seite 63: [258] hat formatiert | Osborn | 31.07.2020 15:28:00 |

Schriftart: Times New Roman, 10 Pt.

| Seite 63: [259] hat formatiert | Osborn | 31.07.2020 15:28:00 |

Schriftart: Times New Roman, 10 Pt.

| Seite 63: [259] hat formatiert | Osborn | 31.07.2020 15:28:00 |

Schriftart: Times New Roman, 10 Pt.

| Seite 63: [260] hat formatiert | Osborn | 31.07.2020 15:28:00 |

Schriftart: Times New Roman, 10 Pt.

| Seite 63: [260] hat formatiert | Osborn | 31.07.2020 15:28:00 |
|---|---|---|

Schriftart: Times New Roman, 10 Pt.

| Seite 63: [261] hat formatiert | Osborn | 31.07.2020 15:28:00 |
|---|---|---|

Schriftart: Times New Roman, 10 Pt.

| Seite 63: [261] hat formatiert | Osborn | 31.07.2020 15:28:00 |
|---|---|---|

Schriftart: Times New Roman, 10 Pt.

| Seite 63: [262] hat formatiert | Osborn | 31.07.2020 15:28:00 |
|---|---|---|

Schriftart: Times New Roman, 10 Pt.

| Seite 63: [262] hat formatiert | Osborn | 31.07.2020 15:28:00 |
|---|---|---|

Schriftart: Times New Roman, 10 Pt.

| Seite 63: [263] hat formatiert | Osborn | 31.07.2020 15:28:00 |
|---|---|---|

Schriftart: Times New Roman, 10 Pt.

| Seite 63: [263] hat formatiert | Osborn | 31.07.2020 15:28:00 |
|---|---|---|

Schriftart: Times New Roman, 10 Pt.

| Seite 63: [264] hat formatiert | Osborn | 31.07.2020 15:28:00 |
|---|---|---|

Schriftart: Times New Roman, 10 Pt.

| Seite 63: [264] hat formatiert | Osborn | 31.07.2020 15:28:00 |
|---|---|---|

Schriftart: Times New Roman, 10 Pt.

| Seite 63: [265] hat formatiert | Osborn | 31.07.2020 15:28:00 |
|---|---|---|

Schriftart: Times New Roman, 10 Pt.

| Seite 63: [265] hat formatiert | Osborn | 31.07.2020 15:28:00 |
|---|---|---|

Schriftart: Times New Roman, 10 Pt.

| Seite 63: [266] hat formatiert | Osborn | 31.07.2020 15:28:00 |
|---|---|---|

Schriftart: Times New Roman, 10 Pt.

| Seite 63: [266] hat formatiert | Osborn | 31.07.2020 15:28:00 |
|---|---|---|

Schriftart: Times New Roman, 10 Pt.

| Seite 63: [267] hat formatiert | Osborn | 31.07.2020 15:28:00 |
|---|---|---|

Schriftart: Times New Roman, 10 Pt.

| Seite 63: [268] Formatiert | Unknown | 31.07.2020 15:28:00 |
|---|---|---|

Zentriert, Position: Horizontal: Links, Gemessen von: Seitenrand, Vertikal: 3,9 cm, Gemessen von: Seite, Horizontal: 0,32 cm, Umschließen

| Seite 63: [269] hat formatiert | Osborn | 31.07.2020 15:28:00 |
|---|---|---|

Schriftart: Times New Roman, 11 Pt.

| Seite 63: [270] hat formatiert | Osborn | 31.07.2020 15:28:00 |
|---|---|---|

Schriftart: Times New Roman, 10 Pt.

| | | |
|---|---|---|
| **Seite 63: [270] hat formatiert** | **Osborn** | **31.07.2020 15:28:00** |

Schriftart: Times New Roman, 10 Pt.

| | | |
|---|---|---|
| **Seite 63: [271] hat formatiert** | **Osborn** | **31.07.2020 15:28:00** |

Schriftart: Times New Roman, 10 Pt.

| | | |
|---|---|---|
| **Seite 63: [271] hat formatiert** | **Osborn** | **31.07.2020 15:28:00** |

Schriftart: Times New Roman, 10 Pt.

| | | |
|---|---|---|
| **Seite 63: [272] hat formatiert** | **Osborn** | **31.07.2020 15:28:00** |

Schriftart: Times New Roman, 10 Pt.

| | | |
|---|---|---|
| **Seite 63: [272] hat formatiert** | **Osborn** | **31.07.2020 15:28:00** |

Schriftart: Times New Roman, 10 Pt.

| | | |
|---|---|---|
| **Seite 63: [273] hat formatiert** | **Osborn** | **31.07.2020 15:28:00** |

Schriftart: Times New Roman, 10 Pt.

| | | |
|---|---|---|
| **Seite 63: [273] hat formatiert** | **Osborn** | **31.07.2020 15:28:00** |

Schriftart: Times New Roman, 10 Pt.

| | | |
|---|---|---|
| **Seite 63: [274] hat formatiert** | **Osborn** | **31.07.2020 15:28:00** |

Schriftart: Times New Roman, 10 Pt.

| | | |
|---|---|---|
| **Seite 63: [274] hat formatiert** | **Osborn** | **31.07.2020 15:28:00** |

Schriftart: Times New Roman, 10 Pt.

| | | |
|---|---|---|
| **Seite 63: [275] hat formatiert** | **Osborn** | **31.07.2020 15:28:00** |

Schriftart: Times New Roman, 10 Pt.

| | | |
|---|---|---|
| **Seite 63: [275] hat formatiert** | **Osborn** | **31.07.2020 15:28:00** |

Schriftart: Times New Roman, 10 Pt.

| | | |
|---|---|---|
| **Seite 63: [276] hat formatiert** | **Osborn** | **31.07.2020 15:28:00** |

Schriftart: Times New Roman, 10 Pt.

| | | |
|---|---|---|
| **Seite 63: [276] hat formatiert** | **Osborn** | **31.07.2020 15:28:00** |

Schriftart: Times New Roman, 10 Pt.

| | | |
|---|---|---|
| **Seite 63: [277] hat formatiert** | **Osborn** | **31.07.2020 15:28:00** |

Schriftart: Times New Roman, 10 Pt.

| | | |
|---|---|---|
| **Seite 63: [277] hat formatiert** | **Osborn** | **31.07.2020 15:28:00** |

Schriftart: Times New Roman, 10 Pt.

| | | |
|---|---|---|
| **Seite 63: [278] hat formatiert** | **Osborn** | **31.07.2020 15:28:00** |

Schriftart: Times New Roman, 10 Pt.

| | | |
|---|---|---|
| **Seite 63: [278] hat formatiert** | **Osborn** | **31.07.2020 15:28:00** |

Schriftart: Times New Roman, 10 Pt.

| Seite 63: [279] hat formatiert | Osborn | 31.07.2020 15:28:00 |

Schriftart: Times New Roman, 10 Pt.

| Seite 63: [279] hat formatiert | Osborn | 31.07.2020 15:28:00 |

Schriftart: Times New Roman, 10 Pt.

| Seite 63: [280] hat formatiert | Osborn | 31.07.2020 15:28:00 |

Schriftart: Times New Roman, 10 Pt.

| Seite 63: [280] hat formatiert | Osborn | 31.07.2020 15:28:00 |

Schriftart: Times New Roman, 10 Pt.

| Seite 63: [281] hat formatiert | Osborn | 31.07.2020 15:28:00 |

Schriftart: Times New Roman, 10 Pt.

| Seite 63: [282] Formatiert | Unknown | 31.07.2020 15:28:00 |

Zentriert, Position: Horizontal: Links, Gemessen von: Seitenrand, Vertikal: 3,9 cm, Gemessen von: Seite, Horizontal: 0,32 cm, Umschließen

| Seite 63: [283] hat formatiert | Osborn | 31.07.2020 15:28:00 |

Schriftart: Times New Roman, 11 Pt.

| Seite 63: [284] hat formatiert | Osborn | 31.07.2020 15:28:00 |

Schriftart: Times New Roman, 10 Pt.

| Seite 63: [284] hat formatiert | Osborn | 31.07.2020 15:28:00 |

Schriftart: Times New Roman, 10 Pt.

| Seite 63: [285] hat formatiert | Osborn | 31.07.2020 15:28:00 |

Schriftart: Times New Roman, 10 Pt.

| Seite 63: [285] hat formatiert | Osborn | 31.07.2020 15:28:00 |

Schriftart: Times New Roman, 10 Pt.

| Seite 63: [286] hat formatiert | Osborn | 31.07.2020 15:28:00 |

Schriftart: Times New Roman, 10 Pt.

| Seite 63: [286] hat formatiert | Osborn | 31.07.2020 15:28:00 |

Schriftart: Times New Roman, 10 Pt.

| Seite 63: [287] hat formatiert | Osborn | 31.07.2020 15:28:00 |

Schriftart: Times New Roman, 10 Pt.

| Seite 63: [287] hat formatiert | Osborn | 31.07.2020 15:28:00 |

Schriftart: Times New Roman, 10 Pt.

| Seite 63: [288] hat formatiert | Osborn | 31.07.2020 15:28:00 |

Schriftart: Times New Roman, 10 Pt.

| Seite 63: [288] hat formatiert | Osborn | 31.07.2020 15:28:00 |

Schriftart: Times New Roman, 10 Pt.

| Seite 63: [289] hat formatiert | Osborn | 31.07.2020 15:28:00 |
|---|---|---|

Schriftart: Times New Roman, 10 Pt.

| Seite 63: [289] hat formatiert | Osborn | 31.07.2020 15:28:00 |
|---|---|---|

Schriftart: Times New Roman, 10 Pt.

| Seite 63: [290] hat formatiert | Osborn | 31.07.2020 15:28:00 |
|---|---|---|

Schriftart: Times New Roman, 10 Pt.

| Seite 63: [290] hat formatiert | Osborn | 31.07.2020 15:28:00 |
|---|---|---|

Schriftart: Times New Roman, 10 Pt.

| Seite 63: [291] hat formatiert | Osborn | 31.07.2020 15:28:00 |
|---|---|---|

Schriftart: Times New Roman, 10 Pt.

| Seite 63: [291] hat formatiert | Osborn | 31.07.2020 15:28:00 |
|---|---|---|

Schriftart: Times New Roman, 10 Pt.

| Seite 63: [292] hat formatiert | Osborn | 31.07.2020 15:28:00 |
|---|---|---|

Schriftart: Times New Roman, 10 Pt.

| Seite 63: [292] hat formatiert | Osborn | 31.07.2020 15:28:00 |
|---|---|---|

Schriftart: Times New Roman, 10 Pt.

| Seite 63: [293] hat formatiert | Osborn | 31.07.2020 15:28:00 |
|---|---|---|

Schriftart: Times New Roman, 10 Pt.

| Seite 63: [293] hat formatiert | Osborn | 31.07.2020 15:28:00 |
|---|---|---|

Schriftart: Times New Roman, 10 Pt.

| Seite 63: [294] hat formatiert | Osborn | 31.07.2020 15:28:00 |
|---|---|---|

Schriftart: Times New Roman, 10 Pt.

| Seite 63: [294] hat formatiert | Osborn | 31.07.2020 15:28:00 |
|---|---|---|

Schriftart: Times New Roman, 10 Pt.

| Seite 63: [295] hat formatiert | Osborn | 31.07.2020 15:28:00 |
|---|---|---|

Schriftart: Times New Roman, 10 Pt.

| Seite 63: [296] Formatiert | Unknown | 31.07.2020 15:28:00 |
|---|---|---|

Zentriert, Position: Horizontal: Links, Gemessen von: Seitenrand, Vertikal:  3,9 cm, Gemessen von: Seite, Horizontal:  0,32 cm, Umschließen

| Seite 63: [297] hat formatiert | Osborn | 31.07.2020 15:28:00 |
|---|---|---|

Schriftart: Times New Roman, 11 Pt.

| Seite 63: [298] hat formatiert | Osborn | 31.07.2020 15:28:00 |
|---|---|---|

Schriftart: Times New Roman, 10 Pt.

| Seite 63: [298] hat formatiert | Osborn | 31.07.2020 15:28:00 |
|---|---|---|

Schriftart: Times New Roman, 10 Pt.

| Seite 63: [299] hat formatiert | Osborn | 31.07.2020 15:28:00 |

Schriftart: Times New Roman, 10 Pt.

| Seite 63: [299] hat formatiert | Osborn | 31.07.2020 15:28:00 |

Schriftart: Times New Roman, 10 Pt.

| Seite 63: [300] hat formatiert | Osborn | 31.07.2020 15:28:00 |

Schriftart: Times New Roman, 10 Pt.

| Seite 63: [300] hat formatiert | Osborn | 31.07.2020 15:28:00 |

Schriftart: Times New Roman, 10 Pt.

| Seite 63: [301] hat formatiert | Osborn | 31.07.2020 15:28:00 |

Schriftart: Times New Roman, 10 Pt.

| Seite 63: [301] hat formatiert | Osborn | 31.07.2020 15:28:00 |

Schriftart: Times New Roman, 10 Pt.

| Seite 63: [302] hat formatiert | Osborn | 31.07.2020 15:28:00 |

Schriftart: Times New Roman, 10 Pt.

| Seite 63: [302] hat formatiert | Osborn | 31.07.2020 15:28:00 |

Schriftart: Times New Roman, 10 Pt.

| Seite 63: [303] hat formatiert | Osborn | 31.07.2020 15:28:00 |

Schriftart: Times New Roman, 10 Pt.

| Seite 63: [303] hat formatiert | Osborn | 31.07.2020 15:28:00 |

Schriftart: Times New Roman, 10 Pt.

| Seite 63: [304] hat formatiert | Osborn | 31.07.2020 15:28:00 |

Schriftart: Times New Roman, 10 Pt.

| Seite 63: [304] hat formatiert | Osborn | 31.07.2020 15:28:00 |

Schriftart: Times New Roman, 10 Pt.

| Seite 63: [305] hat formatiert | Osborn | 31.07.2020 15:28:00 |

Schriftart: Times New Roman, 10 Pt.

| Seite 63: [305] hat formatiert | Osborn | 31.07.2020 15:28:00 |

Schriftart: Times New Roman, 10 Pt.

| Seite 63: [306] hat formatiert | Osborn | 31.07.2020 15:28:00 |

Schriftart: Times New Roman, 10 Pt.

| Seite 63: [306] hat formatiert | Osborn | 31.07.2020 15:28:00 |

Schriftart: Times New Roman, 10 Pt.

| Seite 63: [307] hat formatiert | Osborn | 31.07.2020 15:28:00 |

Schriftart: Times New Roman, 10 Pt.

| Seite 63: [307] hat formatiert | Osborn | 31.07.2020 15:28:00 |

Schriftart: Times New Roman, 10 Pt.

| **Seite 63: [308] hat formatiert** | **Osborn** | **31.07.2020 15:28:00** |

Schriftart: Times New Roman, 10 Pt.

| **Seite 63: [308] hat formatiert** | **Osborn** | **31.07.2020 15:28:00** |

Schriftart: Times New Roman, 10 Pt.

| **Seite 63: [309] hat formatiert** | **Osborn** | **31.07.2020 15:28:00** |

Schriftart: Times New Roman, 10 Pt.

| **Seite 63: [310] Formatiert** | **Unknown** | **31.07.2020 15:28:00** |

Zentriert, Position: Horizontal: Links, Gemessen von: Seitenrand, Vertikal: 3,9 cm, Gemessen von: Seite, Horizontal: 0,32 cm, Umschließen

| **Seite 63: [311] hat formatiert** | **Osborn** | **31.07.2020 15:28:00** |

Schriftart: Times New Roman, 11 Pt.

| **Seite 63: [312] hat formatiert** | **Osborn** | **31.07.2020 15:28:00** |

Schriftart: Times New Roman, 10 Pt.

| **Seite 63: [312] hat formatiert** | **Osborn** | **31.07.2020 15:28:00** |

Schriftart: Times New Roman, 10 Pt.

| **Seite 63: [313] hat formatiert** | **Osborn** | **31.07.2020 15:28:00** |

Schriftart: Times New Roman, 10 Pt.

| **Seite 63: [313] hat formatiert** | **Osborn** | **31.07.2020 15:28:00** |

Schriftart: Times New Roman, 10 Pt.

| **Seite 63: [314] hat formatiert** | **Osborn** | **31.07.2020 15:28:00** |

Schriftart: Times New Roman, 10 Pt.

| **Seite 63: [314] hat formatiert** | **Osborn** | **31.07.2020 15:28:00** |

Schriftart: Times New Roman, 10 Pt.

| **Seite 63: [315] hat formatiert** | **Osborn** | **31.07.2020 15:28:00** |

Schriftart: Times New Roman, 10 Pt.

| **Seite 63: [315] hat formatiert** | **Osborn** | **31.07.2020 15:28:00** |

Schriftart: Times New Roman, 10 Pt.

| **Seite 63: [316] hat formatiert** | **Osborn** | **31.07.2020 15:28:00** |

Schriftart: Times New Roman, 10 Pt.

| **Seite 63: [316] hat formatiert** | **Osborn** | **31.07.2020 15:28:00** |

Schriftart: Times New Roman, 10 Pt.

| **Seite 63: [317] hat formatiert** | **Osborn** | **31.07.2020 15:28:00** |

Schriftart: Times New Roman, 10 Pt.

| **Seite 63: [317] hat formatiert** | **Osborn** | **31.07.2020 15:28:00** |

Schriftart: Times New Roman, 10 Pt.

| Seite 63: [318] hat formatiert | Osborn | 31.07.2020 15:28:00 |
|---|---|---|

Schriftart: Times New Roman, 10 Pt.

| Seite 63: [318] hat formatiert | Osborn | 31.07.2020 15:28:00 |
|---|---|---|

Schriftart: Times New Roman, 10 Pt.

| Seite 63: [319] hat formatiert | Osborn | 31.07.2020 15:28:00 |
|---|---|---|

Schriftart: Times New Roman, 10 Pt.

| Seite 63: [319] hat formatiert | Osborn | 31.07.2020 15:28:00 |
|---|---|---|

Schriftart: Times New Roman, 10 Pt.

| Seite 63: [320] hat formatiert | Osborn | 31.07.2020 15:28:00 |
|---|---|---|

Schriftart: Times New Roman, 10 Pt.

| Seite 63: [320] hat formatiert | Osborn | 31.07.2020 15:28:00 |
|---|---|---|

Schriftart: Times New Roman, 10 Pt.

| Seite 63: [321] hat formatiert | Osborn | 31.07.2020 15:28:00 |
|---|---|---|

Schriftart: Times New Roman, 10 Pt.

| Seite 63: [321] hat formatiert | Osborn | 31.07.2020 15:28:00 |
|---|---|---|

Schriftart: Times New Roman, 10 Pt.

| Seite 63: [322] hat formatiert | Osborn | 31.07.2020 15:28:00 |
|---|---|---|

Schriftart: Fett, Nicht Kursiv

| Seite 63: [323] Formatiert | Osborn | 31.07.2020 15:28:00 |
|---|---|---|

Abstand Vor:  0 Pt.

| Seite 63: [324] hat formatiert | Osborn | 31.07.2020 15:28:00 |
|---|---|---|

Schriftart: Nicht Kursiv

| Seite 63: [325] hat formatiert | Osborn | 31.07.2020 15:28:00 |
|---|---|---|

Schriftart: 12 Pt.

| Seite 63: [326] hat formatiert | Osborn | 31.07.2020 15:28:00 |
|---|---|---|

Schriftart: Times New Roman, 11 Pt.

| Seite 63: [327] hat formatiert | Osborn | 31.07.2020 15:28:00 |
|---|---|---|

Schriftart: Times New Roman, 10 Pt.

| Seite 63: [328] Formatiert | Unknown | 31.07.2020 15:28:00 |
|---|---|---|

Zentriert

| Seite 63: [329] hat formatiert | Osborn | 31.07.2020 15:28:00 |
|---|---|---|

Schriftart: Times New Roman, 11 Pt.

| Seite 63: [330] hat formatiert | Osborn | 31.07.2020 15:28:00 |
|---|---|---|

Schriftart: Times New Roman, 10 Pt.

| Seite 63: [330] hat formatiert | Osborn | 31.07.2020 15:28:00 |
|---|---|---|

Schriftart: Times New Roman, 10 Pt.

| Seite 63: [331] hat formatiert | Osborn | 31.07.2020 15:28:00 |
|---|---|---|

Schriftart: Times New Roman, 10 Pt.

| Seite 63: [331] hat formatiert | Osborn | 31.07.2020 15:28:00 |
|---|---|---|

Schriftart: Times New Roman, 10 Pt.

| Seite 63: [332] hat formatiert | Osborn | 31.07.2020 15:28:00 |
|---|---|---|

Schriftart: Times New Roman, 10 Pt.

| Seite 63: [332] hat formatiert | Osborn | 31.07.2020 15:28:00 |
|---|---|---|

Schriftart: Times New Roman, 10 Pt.

| Seite 63: [333] hat formatiert | Osborn | 31.07.2020 15:28:00 |
|---|---|---|

Schriftart: Times New Roman, 10 Pt.

| Seite 63: [333] hat formatiert | Osborn | 31.07.2020 15:28:00 |
|---|---|---|

Schriftart: Times New Roman, 10 Pt.

| Seite 63: [334] hat formatiert | Osborn | 31.07.2020 15:28:00 |
|---|---|---|

Schriftart: Times New Roman, 10 Pt.

| Seite 63: [334] hat formatiert | Osborn | 31.07.2020 15:28:00 |
|---|---|---|

Schriftart: Times New Roman, 10 Pt.

| Seite 63: [335] hat formatiert | Osborn | 31.07.2020 15:28:00 |
|---|---|---|

Schriftart: Times New Roman, 10 Pt.

| Seite 63: [335] hat formatiert | Osborn | 31.07.2020 15:28:00 |
|---|---|---|

Schriftart: Times New Roman, 10 Pt.

| Seite 63: [336] hat formatiert | Osborn | 31.07.2020 15:28:00 |
|---|---|---|

Schriftart: Times New Roman, 10 Pt.

| Seite 63: [336] hat formatiert | Osborn | 31.07.2020 15:28:00 |
|---|---|---|

Schriftart: Times New Roman, 10 Pt.

| Seite 63: [337] hat formatiert | Osborn | 31.07.2020 15:28:00 |
|---|---|---|

Schriftart: Times New Roman, 10 Pt.

| Seite 63: [338] Formatiert | Unknown | 31.07.2020 15:28:00 |
|---|---|---|

Zentriert

| Seite 63: [339] hat formatiert | Osborn | 31.07.2020 15:28:00 |
|---|---|---|

Schriftart: Times New Roman, 11 Pt.

| Seite 63: [340] hat formatiert | Osborn | 31.07.2020 15:28:00 |
|---|---|---|

Schriftart: Times New Roman, 10 Pt.

| Seite 63: [340] hat formatiert | Osborn | 31.07.2020 15:28:00 |
|---|---|---|

Schriftart: Times New Roman, 10 Pt.

| Seite 63: [341] hat formatiert | Osborn | 31.07.2020 15:28:00 |
|---|---|---|

Schriftart: Times New Roman, 10 Pt.

| Seite 63: [341] hat formatiert | Osborn | 31.07.2020 15:28:00 |
|---|---|---|

Schriftart: Times New Roman, 10 Pt.

| Seite 63: [342] hat formatiert | Osborn | 31.07.2020 15:28:00 |
|---|---|---|

Schriftart: Times New Roman, 10 Pt.

| Seite 63: [342] hat formatiert | Osborn | 31.07.2020 15:28:00 |
|---|---|---|

Schriftart: Times New Roman, 10 Pt.

| Seite 63: [343] hat formatiert | Osborn | 31.07.2020 15:28:00 |
|---|---|---|

Schriftart: Times New Roman, 10 Pt.

| Seite 63: [343] hat formatiert | Osborn | 31.07.2020 15:28:00 |
|---|---|---|

Schriftart: Times New Roman, 10 Pt.

| Seite 63: [344] hat formatiert | Osborn | 31.07.2020 15:28:00 |
|---|---|---|

Schriftart: Times New Roman, 10 Pt.

| Seite 63: [344] hat formatiert | Osborn | 31.07.2020 15:28:00 |
|---|---|---|

Schriftart: Times New Roman, 10 Pt.

| Seite 63: [345] hat formatiert | Osborn | 31.07.2020 15:28:00 |
|---|---|---|

Schriftart: Times New Roman, 10 Pt.

| Seite 63: [345] hat formatiert | Osborn | 31.07.2020 15:28:00 |
|---|---|---|

Schriftart: Times New Roman, 10 Pt.

| Seite 63: [346] hat formatiert | Osborn | 31.07.2020 15:28:00 |
|---|---|---|

Schriftart: Times New Roman, 10 Pt.

| Seite 63: [346] hat formatiert | Osborn | 31.07.2020 15:28:00 |
|---|---|---|

Schriftart: Times New Roman, 10 Pt.

| Seite 63: [347] hat formatiert | Osborn | 31.07.2020 15:28:00 |
|---|---|---|

Schriftart: Times New Roman, 10 Pt.

| Seite 63: [348] Formatiert | Unknown | 31.07.2020 15:28:00 |
|---|---|---|

Zentriert

| Seite 63: [349] hat formatiert | Osborn | 31.07.2020 15:28:00 |
|---|---|---|

Schriftart: Times New Roman, 11 Pt.

| Seite 63: [350] hat formatiert | Osborn | 31.07.2020 15:28:00 |
|---|---|---|

Schriftart: Times New Roman, 10 Pt.

| Seite 63: [350] hat formatiert | Osborn | 31.07.2020 15:28:00 |
|---|---|---|

Schriftart: Times New Roman, 10 Pt.

| Seite 63: [351] hat formatiert | Osborn | 31.07.2020 15:28:00 |

Schriftart: Times New Roman, 10 Pt.

| Seite 63: [351] hat formatiert | Osborn | 31.07.2020 15:28:00 |

Schriftart: Times New Roman, 10 Pt.

| Seite 63: [352] hat formatiert | Osborn | 31.07.2020 15:28:00 |

Schriftart: Times New Roman, 10 Pt.

| Seite 63: [352] hat formatiert | Osborn | 31.07.2020 15:28:00 |

Schriftart: Times New Roman, 10 Pt.

| Seite 63: [353] hat formatiert | Osborn | 31.07.2020 15:28:00 |

Schriftart: Times New Roman, 10 Pt.

| Seite 63: [353] hat formatiert | Osborn | 31.07.2020 15:28:00 |

Schriftart: Times New Roman, 10 Pt.

| Seite 63: [354] hat formatiert | Osborn | 31.07.2020 15:28:00 |

Schriftart: Times New Roman, 10 Pt.

| Seite 63: [354] hat formatiert | Osborn | 31.07.2020 15:28:00 |

Schriftart: Times New Roman, 10 Pt.

| Seite 63: [355] hat formatiert | Osborn | 31.07.2020 15:28:00 |

Schriftart: Times New Roman, 10 Pt.

| Seite 63: [355] hat formatiert | Osborn | 31.07.2020 15:28:00 |

Schriftart: Times New Roman, 10 Pt.

| Seite 63: [356] hat formatiert | Osborn | 31.07.2020 15:28:00 |

Schriftart: Times New Roman, 10 Pt.

| Seite 63: [356] hat formatiert | Osborn | 31.07.2020 15:28:00 |

Schriftart: Times New Roman, 10 Pt.

| Seite 63: [357] hat formatiert | Osborn | 31.07.2020 15:28:00 |

Schriftart: Times New Roman, 10 Pt.

| Seite 63: [358] Formatiert | Unknown | 31.07.2020 15:28:00 |

Zentriert

| Seite 63: [359] hat formatiert | Osborn | 31.07.2020 15:28:00 |

Schriftart: Times New Roman, 11 Pt.

| Seite 63: [360] hat formatiert | Osborn | 31.07.2020 15:28:00 |

Schriftart: Times New Roman, 10 Pt.

| Seite 63: [360] hat formatiert | Osborn | 31.07.2020 15:28:00 |

Schriftart: Times New Roman, 10 Pt.

| Seite 63: [361] hat formatiert | Osborn | 31.07.2020 15:28:00 |
|---|---|---|

Schriftart: Times New Roman, 10 Pt.

| Seite 63: [361] hat formatiert | Osborn | 31.07.2020 15:28:00 |
|---|---|---|

Schriftart: Times New Roman, 10 Pt.

| Seite 63: [362] hat formatiert | Osborn | 31.07.2020 15:28:00 |
|---|---|---|

Schriftart: Times New Roman, 10 Pt.

| Seite 63: [362] hat formatiert | Osborn | 31.07.2020 15:28:00 |
|---|---|---|

Schriftart: Times New Roman, 10 Pt.

| Seite 63: [363] hat formatiert | Osborn | 31.07.2020 15:28:00 |
|---|---|---|

Schriftart: Times New Roman, 10 Pt.

| Seite 63: [363] hat formatiert | Osborn | 31.07.2020 15:28:00 |
|---|---|---|

Schriftart: Times New Roman, 10 Pt.

| Seite 63: [364] hat formatiert | Osborn | 31.07.2020 15:28:00 |
|---|---|---|

Schriftart: Times New Roman, 10 Pt.

| Seite 63: [364] hat formatiert | Osborn | 31.07.2020 15:28:00 |
|---|---|---|

Schriftart: Times New Roman, 10 Pt.

| Seite 63: [365] hat formatiert | Osborn | 31.07.2020 15:28:00 |
|---|---|---|

Schriftart: Times New Roman, 10 Pt.

| Seite 63: [365] hat formatiert | Osborn | 31.07.2020 15:28:00 |
|---|---|---|

Schriftart: Times New Roman, 10 Pt.

| Seite 63: [366] hat formatiert | Osborn | 31.07.2020 15:28:00 |
|---|---|---|

Schriftart: Times New Roman, 10 Pt.

| Seite 63: [366] hat formatiert | Osborn | 31.07.2020 15:28:00 |
|---|---|---|

Schriftart: Times New Roman, 10 Pt.

| Seite 63: [367] hat formatiert | Osborn | 31.07.2020 15:28:00 |
|---|---|---|

Schriftart: Times New Roman, 10 Pt.

| Seite 63: [368] Formatiert | Unknown | 31.07.2020 15:28:00 |
|---|---|---|

Zentriert

| Seite 63: [369] hat formatiert | Osborn | 31.07.2020 15:28:00 |
|---|---|---|

Schriftart: Times New Roman, 11 Pt.

| Seite 63: [370] hat formatiert | Osborn | 31.07.2020 15:28:00 |
|---|---|---|

Schriftart: Times New Roman, 10 Pt.

| Seite 63: [370] hat formatiert | Osborn | 31.07.2020 15:28:00 |
|---|---|---|

Schriftart: Times New Roman, 10 Pt.

| | | |
|---|---|---|
| **Seite 63: [371] hat formatiert** | **Osborn** | **31.07.2020 15:28:00** |

Schriftart: Times New Roman, 10 Pt.

| | | |
|---|---|---|
| **Seite 63: [371] hat formatiert** | **Osborn** | **31.07.2020 15:28:00** |

Schriftart: Times New Roman, 10 Pt.

| | | |
|---|---|---|
| **Seite 63: [372] hat formatiert** | **Osborn** | **31.07.2020 15:28:00** |

Schriftart: Times New Roman, 10 Pt.

| | | |
|---|---|---|
| **Seite 63: [372] hat formatiert** | **Osborn** | **31.07.2020 15:28:00** |

Schriftart: Times New Roman, 10 Pt.

| | | |
|---|---|---|
| **Seite 63: [373] hat formatiert** | **Osborn** | **31.07.2020 15:28:00** |

Schriftart: Times New Roman, 10 Pt.

| | | |
|---|---|---|
| **Seite 63: [373] hat formatiert** | **Osborn** | **31.07.2020 15:28:00** |

Schriftart: Times New Roman, 10 Pt.

| | | |
|---|---|---|
| **Seite 63: [374] hat formatiert** | **Osborn** | **31.07.2020 15:28:00** |

Schriftart: Times New Roman, 10 Pt.

| | | |
|---|---|---|
| **Seite 63: [374] hat formatiert** | **Osborn** | **31.07.2020 15:28:00** |

Schriftart: Times New Roman, 10 Pt.

| | | |
|---|---|---|
| **Seite 63: [375] hat formatiert** | **Osborn** | **31.07.2020 15:28:00** |

Schriftart: Times New Roman, 10 Pt.

| | | |
|---|---|---|
| **Seite 63: [375] hat formatiert** | **Osborn** | **31.07.2020 15:28:00** |

Schriftart: Times New Roman, 10 Pt.

| | | |
|---|---|---|
| **Seite 63: [376] hat formatiert** | **Osborn** | **31.07.2020 15:28:00** |

Schriftart: Times New Roman, 10 Pt.

| | | |
|---|---|---|
| **Seite 63: [376] hat formatiert** | **Osborn** | **31.07.2020 15:28:00** |

Schriftart: Times New Roman, 10 Pt.

| | | |
|---|---|---|
| **Seite 63: [377] hat formatiert** | **Osborn** | **31.07.2020 15:28:00** |

Schriftart: Times New Roman, 10 Pt.

| | | |
|---|---|---|
| **Seite 63: [378] Formatiert** | **Unknown** | **31.07.2020 15:28:00** |

Zentriert

| | | |
|---|---|---|
| **Seite 63: [379] hat formatiert** | **Osborn** | **31.07.2020 15:28:00** |

Schriftart: Times New Roman, 11 Pt.

| | | |
|---|---|---|
| **Seite 63: [380] hat formatiert** | **Osborn** | **31.07.2020 15:28:00** |

Schriftart: Times New Roman, 10 Pt.

| | | |
|---|---|---|
| **Seite 63: [380] hat formatiert** | **Osborn** | **31.07.2020 15:28:00** |

Schriftart: Times New Roman, 10 Pt.

| | | |
|---|---|---|
| **Seite 63: [381] hat formatiert** | **Osborn** | **31.07.2020 15:28:00** |

Schriftart: Times New Roman, 10 Pt.

| | | |
|---|---|---|
| **Seite 63: [381] hat formatiert** | **Osborn** | **31.07.2020 15:28:00** |

Schriftart: Times New Roman, 10 Pt.

| | | |
|---|---|---|
| **Seite 63: [382] hat formatiert** | **Osborn** | **31.07.2020 15:28:00** |

Schriftart: Times New Roman, 10 Pt.

| | | |
|---|---|---|
| **Seite 63: [382] hat formatiert** | **Osborn** | **31.07.2020 15:28:00** |

Schriftart: Times New Roman, 10 Pt.

| | | |
|---|---|---|
| **Seite 63: [383] hat formatiert** | **Osborn** | **31.07.2020 15:28:00** |

Schriftart: Times New Roman, 10 Pt.

| | | |
|---|---|---|
| **Seite 63: [383] hat formatiert** | **Osborn** | **31.07.2020 15:28:00** |

Schriftart: Times New Roman, 10 Pt.

| | | |
|---|---|---|
| **Seite 63: [384] hat formatiert** | **Osborn** | **31.07.2020 15:28:00** |

Schriftart: Times New Roman, 10 Pt.

| | | |
|---|---|---|
| **Seite 63: [384] hat formatiert** | **Osborn** | **31.07.2020 15:28:00** |

Schriftart: Times New Roman, 10 Pt.

| | | |
|---|---|---|
| **Seite 63: [385] hat formatiert** | **Osborn** | **31.07.2020 15:28:00** |

Schriftart: Times New Roman, 10 Pt.

| | | |
|---|---|---|
| **Seite 63: [385] hat formatiert** | **Osborn** | **31.07.2020 15:28:00** |

Schriftart: Times New Roman, 10 Pt.